# FINE-GRAINED MIXTURE OF EXPERTS FOR MEDICAL MULTIMODAL LEARNING

## ABSTRACT

Fine-Grained Mixture of Experts is a powerful architecture for scaling large models, yet its application in specialized domains like medicine remains underexplored. In this work, we conduct the first systematic study of expert granularity in a medical multimodal VQA context. Our findings reveal a fundamental trade-off: while increasing granularity significantly enhances out-of-distribution (OOD) generalization and robustness, it slightly degrades in-distribution (ID) fitting and, notably, amplifies expert co-occurrence and functional similarity, indicating stronger collaborative tendencies among experts. We argue that these intensified co-occurrence patterns place additional computational pressure on the routing mechanism, yet also reveal exploitable structure in how experts are jointly activated. To address this, we introduce Adaptive Expert Grouping (AEG), a novel, end-to-end learnable mechanism that leverages these collaborative patterns by dynamically clustering frequently co-activated, functionally related experts. By shifting routing decisions from the individual expert level to the group level, AEG substantially reduces computational overhead and improves model sparsity, while preserving the generalization benefits of the fine-grained architecture. We further observe similar co-occurrence phenomena beyond the medical domain, suggesting that our findings and AEG are broadly applicable. Our work offers a new path towards building more efficient and robust MoE models for specialized domains.

## 1 INTRODUCTION

The Mixture-of-Experts (MoE) paradigm has emerged as a highly effective and computationally efficient strategy for scaling large models (Lepikhin et al., 2020; Fedus et al., 2022; Dai et al., 2024). A recent advancement, Fine-Grained MoE (FGMoE), further pushes the boundaries by decomposing coarse-grained experts into a larger number of smaller, more specialized sub-experts (Krajewski et al., 2024; Dai et al., 2024). This is typically achieved by reducing an expert's internal dimensions (e.g., FFN hidden size) by a factor of $m$, while increasing the total number of experts $m$-fold. This design allows for an exponential increase in the number of potential activation pathways without altering the total parameter count or computational budget. For instance, decomposing 4 experts into 16 sub-experts ($m = 4$) and activating 4 per token boosts the number of unique expert combinations from a mere $\binom{4}{2} = 6$ to a staggering $\binom{16}{4} = 1,820$. These advantages have led FGMoE to demonstrate strong empirical performance in general-domain models, and it is gradually becoming a fundamental scaling paradigm. Motivated by this success, we introduce FGMoE into the medical multimodal domain and study its behavior in this specialized setting.

Despite this potential, the behavior and utility of FGMoE architectures in specialized vertical domains remain largely unexplored. While recent general-domain models have adopted fine-grained structures (Dai et al., 2024; Yang et al., 2024a; Li et al., 2024), the application of MoE architectures in medicine has, until now, been predominantly limited to coarser-grained setups (Jiang et al., 2024). Foundational studies have investigated the relationship between granularity and training loss (Krajewski et al., 2024), but a systematic investigation into how expert granularity impacts model performance, generalization, and internal behavior on diverse downstream tasks within a complex vertical like medicine is notably absent. This gap raises critical questions: Can the theoretical benefits of FGMoE be realized in the data-constrained, knowledge-dense medical domain? More importantly, how does the crucial design choice of expert granularity shape the model's capabilities and emergent properties?

To bridge this gap, we conduct the first systematic empirical study of FGMoE in a medical multimodal context. Our investigation reveals a fundamental dual effect of granularity. On one hand, increasing granularity brings significant benefits: it markedly enhances the model's generalization to out-of-distribution (OOD) data and improves its robustness against input perturbations. On the other hand, this comes with a trade-off in fitting capacity on in-distribution (ID) data and, more critically, leads to a significant increase in expert co-activation patterns.

We interpret the high degree of co-activation as evidence of stronger functional relatedness and partially overlapping behaviors among fine-grained experts. This discovery sheds new light on a critical challenge in scaling these models: routing pressure. As granularity increases, the number of experts ($N$) grows, inevitably increasing the computational burden on the gating network. If many experts are highly related and frequently co-activated, forcing the router to distinguish between them is an unnecessary and wasteful computation.

Building on this observation, we introduce Adaptive Expert Grouping (AEG), which exploits expert co-activation for group-level routing to achieve better computational efficiency while preserving the modeling benefits of fine-grained experts. To this end, we propose **A**daptive **E**xpert **G**rouping (AEG), a novel, end-to-end learnable mechanism. Unlike static approaches that rely on fixed heuristics, AEG dynamically learns to cluster frequently co-activated and functionally similar experts into groups during training. This transforms the complex, fine-grained routing decision at inference time into a simpler, more efficient group-level decision, effectively reducing routing pressure by exploiting the model's inherent co-occurrence structure.

Our contributions are threefold:

- We conduct the first systematic investigation of Fine-Grained MoE in the medical multimodal domain. Our findings demonstrate that increasing granularity significantly enhances model generalization and robustness, while slightly degrading fitting capacity on ID data, and comparative experiments with dense and coarse-grained MoE models further demonstrate the effectiveness of FGMoE in multimodal and knowledge-intensive domains.

- We identify and analyze the "dual effect" of granularity, providing a key insight that the performance gains from fine-graining are coupled with the emergence of strong expert co-activation and collaborative tendencies, offering a new perspective on the internal behavior of highly fine-grained MoE models.

- Based on this insight, we propose Adaptive Expert Grouping (AEG), a novel method that learns to exploit co-occurrence patterns for more efficient routing. We demonstrate that AEG substantially reduces routing overhead and improves routing sparsity, while largely preserving the performance benefits of the fine-grained architecture, and we observe similar phenomena and gains in non-medical domains, suggesting that our findings and method generalize across domains.

## 2 METHODOLOGY

### 2.1 A FRAMEWORK FOR STUDYING GRANULARITY

Our study begins with the Qwen2-VL-2B-Instruct pre-trained dense model. Following established practices (Lin et al., 2024; Jiang et al., 2024), we construct a baseline MoE architecture by replacing the Feed-Forward Network (FFN) with an MoE layer in alternating layers. This baseline layer consists of four identical copies of the original FFN, serving as coarse-grained experts. From this foundation, we generate fine-grained MoE (FGMoE) layers by introducing the concept of granularity, denoted by a factor $G$. Specifically, each of the four coarse-grained experts is sliced into $G$ smaller "sub-experts", resulting in a total of $N = 4 \times G$ fine-grained experts. To maintain a constant number of activated parameters, the number of selected experts per token ($k$) is scaled by the granularity factor $G$, a standard approach in FGMoE research (Krajewski et al., 2024). The overall process is illustrated in Figure 1.

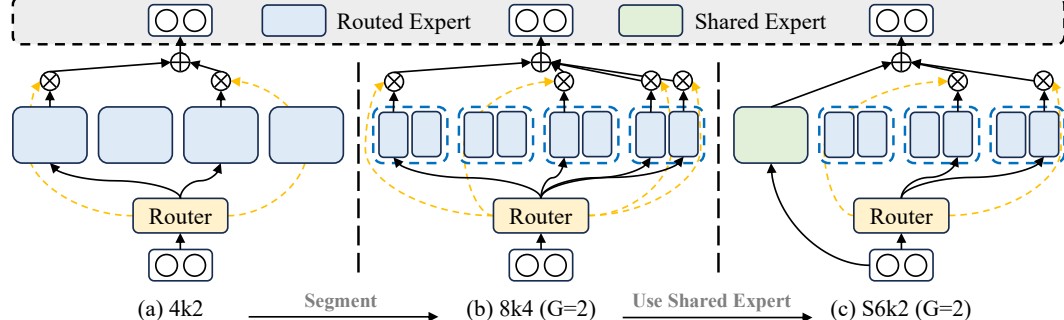

(a) 4k2 — Segment → (b) 8k4 (G=2) — Use Shared Expert → (c) S6k2 (G=2)

Figure 1: **From Coarse-Grained to Fine-Grained MoE Architectures.** **(a)** A standard coarse-grained MoE layer with 4 experts, activating the top 2. **(b)** A naive fine-grained MoE layer ($G = 2$) created by slicing each expert, doubling the number of experts and activated experts. **(c)** Our proposed FGMoE architecture ($G = 2$), which includes a shared, unsliced expert to preserve pre-trained knowledge. We use the notation `[S]<N>k<M>` to describe architectures, where `N` is the number of routable experts, `M` is the number of activated experts, and `S` denotes a shared expert.

## 2.2 FINE-GRAINED MOE WITH A SHARED EXPERT

A naive approach to building an FGMoE model is to directly slice the weights of the pre-trained FFN to initialize the sub-experts, as depicted in Figure 1(b). However, our preliminary experiments revealed that this method leads to significant performance degradation on downstream tasks, a problem that is exacerbated as granularity $G$ increases showed in Tab. 6. We hypothesize that this "direct slicing" corrupts the functional integrity of the pre-trained FFN. Consequently, the resulting sub-experts are poorly initialized, which severely hinders convergence during downstream fine-tuning.

To mitigate this issue, we propose a modified architecture that retains one complete, unsliced copy of the original FFN as a shared expert (Figure 1(c)). This shared expert is activated for every token, operating in parallel with the "top-$M$" fine-grained experts selected by the gating network, allowing the fine-grained experts to safely specialize on top of this foundation without risking model collapse. Our ablations confirmed that using a sliced sub-expert as the shared component fails to provide this stability showed in Tab. 6, underscoring the importance of preserving the full, pre-trained FFN structure.

## 2.3 DENSEMASKMOE: A PROXY FOR EFFICIENT EXPERIMENTATION

Training Mixture-of-Experts models with a large number of experts typically relies on specialized frameworks like DeepSpeed-MoE, which employ techniques such as Expert Parallelism (EP) to distribute experts across multiple devices. While powerful, these frameworks introduce significant engineering overhead and demand specific hardware configurations that can be prohibitive. Our research aims to isolate and study the algorithmic properties of fine-grained experts, which required a methodology that prioritizes rapid experimentation and minimizes infrastructural complexity. To this end, we introduce **DenseMaskMoE**, a computationally efficient proxy for a standard MoE layer that allows us to simulate a large number of experts on simpler hardware setups.

The core insight of DenseMaskMoE is to simulate the sparse routing process using dense matrix operations, thereby avoiding the communication-heavy "dispatch" and "combine" steps inherent to traditional MoE implementations. Instead of routing each token to a subset of experts, we compute the outputs of all $E$ experts for all tokens simultaneously in a fused, dense operation. The gating scores, which set the weights of inactive experts to zero for each token, are then used as a soft mask to perform a weighted element-wise sum over all expert outputs. While the underlying computation path is different, this process is mathematically equivalent to a standard MoE forward pass. A detailed, step-by-step implementation of this approach is provided in Algorithm 1 in the appendix.

The effectiveness and validity of DenseMaskMoE as a proxy are confirmed by two key results. First, it provides a substantial performance boost for our experimental setup. As shown in Figure 7,

our method achieves significantly faster training iterations (especially for granularity $G \geq 32$) and superior inference speeds compared to a standard implementation. Second, this efficiency does not compromise model performance. An ablation study detailed in Table 5 confirms that models trained with DenseMaskMoE yield nearly identical results to those trained with a conventional DeepSpeedMoE strategy. Crucially, a model trained with our proxy can be seamlessly converted to the standard expert-dispatch format for inference (labeled "DenseMaskMoE (DS-Infer)" in the table) with only minor performance changes, proving it is a faithful and interchangeable substitute for our experimental purposes. This allows us to focus squarely on the scientific questions at hand, free from complex engineering constraints.

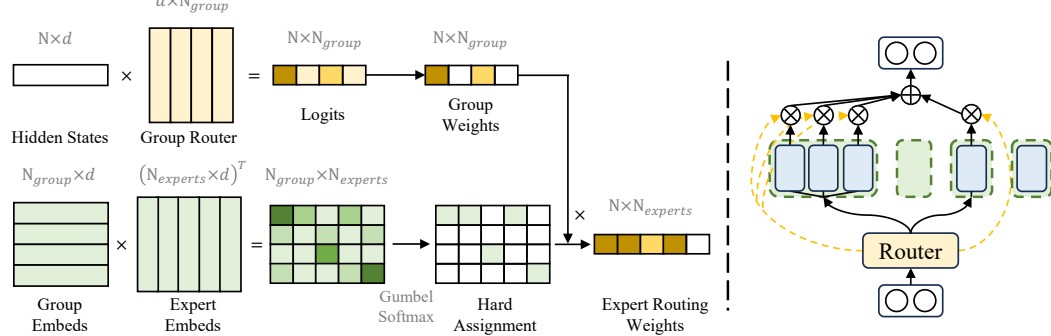

Figure 2: **The Adaptive Expert Grouping (AEG) Mechanism.** *(Left)* AEG forms an assignment matrix $\mathbf{A}$ by applying Gumbel-Softmax to an affinity matrix, which is computed from learnable group and expert embeddings. This matrix translates the router's group-level scores into the final expert-level scores used for computation. *(Right)* An illustrative example of the routing result.

## 2.4 ADAPTIVE EXPERT GROUPING (AEG)

Our empirical analysis in Section 3 reveals that increasing expert granularity leads to a significant rise in expert co-occurrence rates (see Figure 3). This observation motivates the development of a more efficient routing architecture that can exploit this redundancy to reduce routing overhead while preserving the performance gains of fine-grained specialization.

A straightforward approach would be to perform static grouping as a post-processing step, for instance, by applying a community detection algorithm like Louvain(Blondel et al., 2008) on the expert co-occurrence graph. However, we identified several critical flaws with this static approach. First, the resulting groups are highly dependent on the specific data distribution used to generate the co-occurrence statistics, potentially failing to generalize. Second, the MoE model is not trained with these groupings in mind, meaning the discovered clusters may not be optimal for the model's learned representations. And our own attempts using the Louvain algorithm often yielded unbalanced clusters consisting of one large group and many single-expert outliers. These limitations underscore the need for a dynamic grouping mechanism that is learned jointly with the primary task objectives.

To this end, we propose **A**daptive **E**xpert **G**rouping (**AEG**), a method that learns to partition experts into different groups during training. As illustrated in Figure 2, AEG introduces two learnable embeddings inside each MoE layer: a group embedding $\mathbf{E}_g \in \mathbb{R}^{N_g \times d}$ and an expert embedding $\mathbf{E}_e \in \mathbb{R}^{N_e \times d}$, where $N_g$ and $N_e$ are the number of groups and experts, respectively. An affinity score matrix is computed via the dot product between $\mathbf{E}_g$ and $\mathbf{E}_e$, and a discrete, differentiable assignment matrix $\mathbf{A} \in \{0,1\}^{N_g \times N_e}$ is obtained using the Gumbel-Softmax trick (Jang et al., 2016). This matrix $\mathbf{A}$, where $A_{ij} = 1$ if expert $j$ belongs to group $i$, is learned end-to-end within each MoE layer and is used to map group-level routing decisions from the router to the corresponding fine-grained experts.

Under AEG, the router's role is redefined: instead of selecting individual experts, it routes each token to a set of top-$k$ groups. The resulting group-level routing scores $\mathbf{S}_{\text{group}} \in \mathbb{R}^{N \times N_g}$ (for $N$ tokens) are then propagated through the assignment matrix to obtain the final expert-level scores: $\mathbf{S}_{\text{expert}} = \mathbf{S}_{\text{group}}\mathbf{A}$. Since the router now targets groups of varying sizes instead of uniform experts, its

load-balancing mechanism must be adapted. Consequently, the capacity allocated to each group and the corresponding load-balancing loss are calculated proportionally to the group's size, $s_i = \sum_j A_{ij}$.

To guide the formation of meaningful groups, we introduce a separation loss ($\mathcal{L}_{\text{sep}}$). The design objective is to enforce high intra-group cohesion and low inter-group coupling. We use the cosine similarity of expert parameter vectors ($\mathbf{w}_e$) as a proxy for functional similarity. This is based on the hypothesis that frequently co-activated experts, by virtue of processing the same tokens, are driven by similar data distributions and will thus develop comparable weight structures. The separation loss is defined as:

$$\mathcal{L}_{\text{sep}} = \mathcal{L}_{\text{intra}} + \lambda_{\text{inter}} \mathcal{L}_{\text{inter}} \tag{1}$$

where $\mathcal{L}_{\text{intra}}$ encourages cohesion by minimizing the distance between an expert's parameters and its assigned group's centroid ($\mathbf{c}_i$), and $\mathcal{L}_{\text{inter}}$ promotes separation by minimizing the similarity between the centroids of different groups. They are formulated as:

$$\mathcal{L}_{\text{intra}} = \frac{1}{|\mathcal{G}'|} \sum_{i \in \mathcal{G}'} \frac{1}{s_i} \sum_{j: A_{ij}=1} \left(1 - \text{sim}(\mathbf{w}_j, \mathbf{c}_i)\right) \tag{2}$$

$$\mathcal{L}_{\text{inter}} = \frac{1}{\binom{|\mathcal{G}''|}{2}} \sum_{i,k \in \mathcal{G}'', i<k} |\text{sim}(\mathbf{c}_i, \mathbf{c}_k)| \tag{3}$$

Here, $\mathbf{c}_i = \frac{1}{s_i} \sum_j A_{ij} \mathbf{w}_j$ is the centroid for group $i$, $\text{sim}(\cdot)$ denotes cosine similarity, $\mathcal{G}'$ is the set of groups with more than one expert, and $\mathcal{G}''$ is the set of all non-empty groups.

## 3 AN EMPIRICAL STUDY ON GRANULARITY

In this section, we conduct a series of experiments to first investigate the effects of expert granularity and then demonstrate the effectiveness of our proposed Fine-Grained MoE (FGMoE) architecture. We first investigate the impact of expert granularity on model performance and expert co-occurrence when models are fine-tuned directly on downstream tasks. We then demonstrate the effectiveness of our proposed Fine-Grained MoE (FGMoE) architecture, which is pre-trained on a large-scale medical dataset before being fine-tuned, comparing it against strong baselines.

### 3.1 EXPERIMENTAL SETUP

#### 3.1.1 DATASETS AND EVALUATION METRICS

Our experiments utilize a combination of pre-training and downstream task datasets. **Pre-training Data:** For pre-training our MoE models, we use the **Ada** dataset (Cheng et al., 2024), which contains a rich mix of synthesized visual instruction-following data and medical image-caption pairs from PubMedVision (Chen et al., 2024). **Downstream Datasets:** For fine-tuning, we use the training sets of SLAKE (Liu et al., 2021), PATH-VQA (He et al., 2020), and VQA-RAD. However, for evaluation, we only use the test sets of SLAKE and PATH-VQA as our in-distribution (ID) benchmarks. We intentionally exclude VQA-RAD from the evaluation phase to avoid any risk of data leakage from the Ada pre-training corpus. To assess generalization, we use an out-of-distribution (OOD) dataset, OMNI-MINI, which consists of 2,000 samples we curated from OmniMedVQA (Hu et al., 2024).

**Metrics:** We report accuracy for closed-ended questions and recall for open-ended ones. We define three aggregate metrics: **Avg-I** (average performance on ID datasets), **OOD** (performance on the OMNI-MINI dataset), and **Avg-A** (the average of Avg-I and OOD), which represents overall capability.

#### 3.1.2 BASELINES

Our primary comparisons are against self-constructed coarse-grained MoE models (MoE-Qwen2-VL) to ensure a fair comparison of granularity. We also include several state-of-the-art baselines: **Qwen2-VL-2B-Instruct** (Wang et al., 2024): The powerful dense base model for our MoE architecture. **AdaMLLM** (Cheng et al., 2024): A dense model based on Qwen2-VL, further pre-trained on the Ada dataset. **LLaVA-Med** (Li et al., 2023): A well-known medical multimodal model based on Llama-7B. **Med-MoE** (Jiang et al., 2024): The only open-source medical MoE model to our knowledge. Due to its weaker base model, our main comparisons focus on our own constructed baselines.

## 3.2 THE EFFECT OF GRANULARITY ON MODEL PERFORMANCE

To isolate the effect of granularity, we first construct MoE models with varying numbers of experts (from 6 to 192, corresponding to a granularity $G$ from 2 to 64) and fine-tune them directly on downstream tasks without any MoE-specific pre-training.

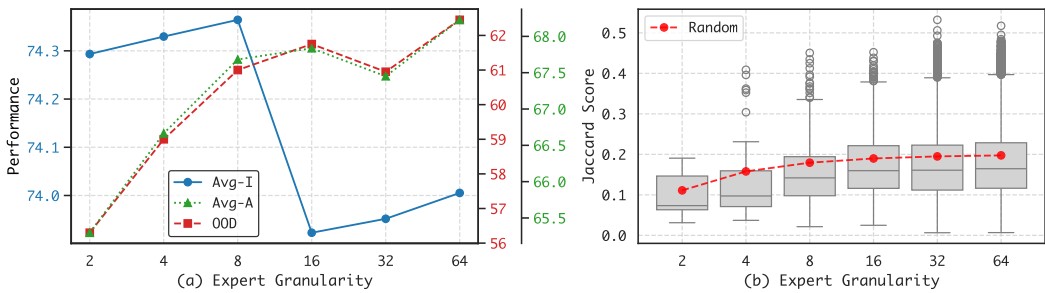

Figure 3: **Impact of Expert Granularity on Performance and Co-occurrence.** (a) Model performance on in-distribution (Avg-I), out-of-distribution (OOD), and overall (Avg-A) tasks as expert granularity increases. (b) Distribution of Jaccard similarity scores between expert pairs across different granularities, compared to a random routing baseline (red line).

As shown in Figure 3(a), increasing expert granularity reveals a clear trade-off. Performance on the OOD dataset, a proxy for generalization, shows a consistent upward trend. Conversely, in-distribution performance (Avg-I) slightly degrades, with a notable drop as granularity moves from 8 to 16. However, the overall performance (Avg-A) mirrors the OOD trend, suggesting that the significant gains in generalization outweigh the minor loss in fitting ability. It is also apparent that the benefits of granularity begin to plateau after $G = 16$, indicating diminishing returns relative to the increased computational cost.

To understand the underlying expert behavior, we analyze expert co-occurrence, the frequency with which pairs of experts are activated for the same token. We measure this using the Jaccard score, which calculates the intersection over the union of tokens processed by two experts (see Appendix E for detailed information). Figure 3(b) shows that as granularity increases, the distribution of Jaccard scores shifts upwards, far exceeding the random baseline. The emergence of high-scoring outliers signifies that certain expert pairs become strongly coupled. We observe a similar combination of improved OOD performance and stronger expert co-occurrence in our non-medical experiments as well (see Appendix F), suggesting that these behaviors reflect a general property of fine-grained experts rather than a medical-domain artifact. This growing tendency toward tightly co-activated experts at finer granularities is the primary motivation for our proposed Adaptive Expert Grouping (AEG) method, which leverages expert co-occurrence to adaptively balance finer-grained modeling effectiveness with coarser-grained computational efficiency.

## 3.3 THE EFFECTIVENESS OF FINE-GRAINED MOE

Having established the general effects of granularity, we now validate the effectiveness of our proposed Fine-Grained MoE (FGMoE) architecture. To unlock the full potential of the experts, we first pre-train the models on the Ada dataset for one epoch before fine-tuning on downstream tasks. We compare our MedFGMoE models at two different levels of granularity—a baseline level of $G = 4$ (MedFGMoE-Ada-S12k4) and a much finer level of $G = 32$ (MedFGMoE-Ada-S96k32)—against conventional coarse-grained MoE baselines (MoE-Qwen2-VL-Ada-4k2 and -S3k1) and other models.

Table 1 presents the main results. In the zero-shot setting, MedFGMoE model at granularity of $G = 4$ (S12k4, Avg-A 56.18) outperforms the standard coarse-grained MoE baseline (4k2, Avg-A 55.63), suggesting inherent benefits of the FGMoE design itself. However, when increasing the granularity within our architecture to $G = 32$, the S96k32 model shows no immediate advantage over its less-fine counterpart. This indicates that pre-training a very large number of specialized experts is more challenging and may require more extensive training to allow them to develop specializations.

Table 1: **Main Results on Medical VQA Benchmarks.** We compare our fine-grained MoE model (MedFGMoE) with dense and coarse-grained MoE baselines under both Zero-Shot and Supervised Fine-tuning settings. Best results are in **red**, second best are in blue.

| Models | In-Distribution Datasets | | | | OOD | | |
|---|---|---|---|---|---|---|---|
| Datasets | SLAKE | | PATH-VQA | | OMNI- | | |
| #Split | Open | Closed | Open | Closed | MINI | Avg-I | Avg-A |
| #Samples | 645 | 416 | 3370 | 3391 | 2000 | | |
| #Metrics | Recall | Acc | Recall | Acc | Acc | | |
| *Zero-Shot Results* | | | | | | | |
| Qwen2-VL-2B-Instruct | 44.62 | 56.97 | 13.55 | 31.05 | **65.60** | - | 42.36 |
| AdaMLLM (Qwen2-VL) | 56.43 | 78.85 | 16.00 | 56.36 | 63.40 | - | 54.20 |
| LLaVA-Med (Llama-7B) | 41.72 | 47.60 | 10.86 | 59.75 | - | - | - |
| Med-MoE (Phi2-2.7B) | 43.93 | 56.97 | 6.94 | 66.46 | - | - | - |
| MoE-Qwen2-VL-Ada-4k2 | 54.80 | **79.33** | 16.03 | 69.36 | 58.65 | - | 55.63 |
| MedFGMoE-Ada-16k8 | 53.89 | 77.64 | **16.39** | **71.66** | 52.80 | - | 54.48 |
| MedFGMoE-Ada-S12k4 | **57.48** | 79.09 | 15.24 | 68.42 | 60.70 | - | **56.18** |
| MedFGMoE-Ada-S96k32 | 55.86 | 76.92 | 16.38 | 69.57 | 57.95 | - | 55.34 |
| *Supervised Finetuning Reults* | | | | | | | |
| AdaMLLM (Qwen2-VL) | **83.93** | 87.02 | **41.72** | 92.77 | 56.40 | **76.36** | 66.38 |
| LLaVA-Med (Llama-7B) | 83.08 | 85.34 | 37.95 | 91.21 | - | 74.40 | - |
| Med-MoE (Phi2-2.7B) | 83.62 | 86.54 | 34.50 | 90.24 | 34.65 | 73.72 | 54.19 |
| MoE-Qwen2-VL-Ada-4k2 | 82.90 | **87.26** | 38.78 | 92.27 | 60.75 | 75.30 | 68.03 |
| MoE-Qwen2-VL-Ada-S3k1 | 82.96 | 87.02 | 38.47 | **92.80** | 60.95 | 75.31 | 68.13 |
| MedFGMoE-Ada-16k8 | 83.04 | 86.06 | 36.67 | 92.36 | 55.45 | 74.53 | 64.99 |
| MedFGMoE-Ada-S12k4 | 82.60 | 83.89 | 38.71 | 92.24 | 63.00 | 74.36 | 68.68 |
| MedFGMoE-Ada-S96k32 | 82.89 | 86.30 | 39.65 | 92.45 | **64.05** | 75.32 | **69.69** |

However, after supervised fine-tuning, the potential of finer granularity is fully realized. Our S96k32 ($G = 32$) emerges as the top-performing model, achieving the best overall score (Avg-A 69.69). It significantly surpasses the coarse-grained baselines (68.03 for 4k2 and 68.13 for S3k1), proving the efficacy of the fine-grained approach. Furthermore, while the dense AdaMLLM model achieves the highest in-distribution score (Avg-I 76.36), it does so by sacrificing generalization, with its OOD performance dropping sharply. In contrast, our fine-grained model maintains a much better balance, confirming its ability to both fit the training data and generalize to unseen data. Comparing within our own architecture, the finer-grained S96k32 model (69.69) now clearly outperforms its less-fine G=4 counterpart, S12k4 (68.68), demonstrating that fine-tuning successfully unlocks the performance gains latent in the finer-grained structure.

Table 2: **Robustness Evaluation under Input Corruptions.** We report the performance drop on Avg-A under three types of noise: Image Gaussian (IG), Image Rotation (IR), and Text Character Substitution (TCS). Lower drop values (closer to 0) indicate better robustness. Best results are in **red**, second best are in blue.

| Models | IG | | IR | | TCS | |
|---|---|---|---|---|---|---|
| | Avg-A | Drop | Avg-A | Drop | Avg-A | Drop |
| MoE-Qwen2-VL-Ada-4k2 | 64.54 | -3.49 | 66.77 | -1.25 | 56.80 | -11.22 |
| MoE-Qwen2-VL-Ada-S3k1 | 62.08 | -2.91 | 67.04 | **2.05** | 53.34 | -11.65 |
| MedFGMoE-Ada-16k8 | 65.28 | -2.85 | 64.05 | -4.09 | 57.12 | -11.01 |
| MedFGMoE-Ada-S12k4 | 66.11 | **-2.57** | 67.49 | -1.19 | 56.65 | -12.03 |
| MedFGMoE-Ada-S96k32 | **66.98** | -2.71 | **69.44** | -0.25 | **59.03** | **-10.66** |

Finally, we assess model robustness by measuring the performance degradation under various input corruptions. As shown in Table 2, our fine-grained models (MedFGMoE) exhibit a smaller performance drop than the coarse-grained models in most cases. Notably, within the fine-grained family, the model (MedFGMoE-Ada-S96k32) achieves the highest resilience against both Image Rotation (IR) and Text Character Substitution (TCS) when compared to its less-fine counterpart

(MedFGMoE-Ada-S12k4). This suggests that increasing expert granularity further enhances the model's resilience to input perturbations.

# 4 EXPERIMENTS AND ANALYSIS OF AEG

In Section 3.2, we observed that increasing expert granularity leads to higher expert co-occurrence, suggesting stronger collaborative tendencies and partially overlapping behaviors among experts. To address this, we introduced Adaptive Expert Grouping (AEG), a method that performs routing at the group level and leverages expert co-occurrence to adaptively balance finer-grained modeling effectiveness with coarser-grained computational efficiency. In this section, we evaluate the performance and efficiency of AEG and analyze the dynamic group structures it learns.

## 4.1 EXPERIMENTAL SETUP

The experimental setup largely mirrors that of Section 3. The key difference is the introduction of our AEG-enabled models, denoted with a "G" prefix (e.g., "MedFGMoE-Ada-G96k2"). We also establish new baselines where sparsity is achieved simply by reducing the top-$k$ value in a standard FGMoE (e.g., "S12k2", "S96k2", "S96k8"). In addition, we compare AEG with static grouping methods, including a heuristic co-occurrence–graph-based grouping method (suffixed with "C") and a $k$-means clustering method (suffixed with "K").

For $k$-means, we use scikit-learn's `KMeans` to cluster experts based on their weight parameters, setting `n_clusters` to three quarters of the total number of routed experts. For the co-occurrence–graph method, we compute the Jaccard similarity between experts and then add an edge between two experts if their similarity exceeds a threshold (0.35 for 12 experts, 0.4 for 96 experts), finally applying connected-component analysis on this graph to form expert groups.

For all AEG experiments, we set the maximum number of expert groups in each MoE layer to three quarters of thes total number of expert, encouraging the model to learn the effective number of functional clusters autonomously. We permit the router to select "empty" groups, enabling a form of dynamic sparsity: the model can choose to activate fewer than the nominal number of groups if no group is deemed relevant for a given token. At inference time, the learned group assignments are frozen, allowing for further optimization by creating fixed, fused expert pathways.

Table 3: **Performance Comparison with Adaptive Expert Grouping (AEG).** We compare AEG models (prefixed with "G") against standard FGMoE models and baselines with reduced top-$k$ values, as well as static grouping methods including a heuristic co-occurrence graph-based grouping method (suffixed with "C") and a k-means clustering method (suffixed with "K"). Best results are in **red**, and second-best results are blue.

| Models | SLAKE | | PATH-VQA | | OMNI-MINI | Avg-I | Avg-A |
|---|---|---|---|---|---|---|---|
| | Open | Closed | Open | Closed | | | |
| MedFGMoE-Ada-S12k4 | 82.60 | 83.89 | 38.71 | 92.24 | 63.00 | 74.36 | 68.68 |
| MedFGMoE-Ada-S96k32 | 82.89 | 86.30 | 39.65 | 92.45 | 64.05 | **75.32** | 69.69 |
| MedFGMoE-Ada-S12k2 | 83.54 | 85.34 | 38.55 | 92.48 | 62.95 | 74.98 | 68.96 |
| MedFGMoE-Ada-S96k2 | 84.35 | 85.10 | 38.20 | 92.33 | 61.80 | 74.99 | 68.40 |
| MedFGMoE-Ada-S12k4-C | 81.86 | 83.41 | 38.06 | 92.10 | 62.10 | 73.86 | 67.98 |
| MedFGMoE-Ada-S12k4-K | 81.50 | 83.89 | 38.08 | 92.36 | 62.35 | 73.96 | 68.15 |
| MedFGMoE-Ada-S96k32-C | 82.59 | 84.38 | 39.62 | 92.36 | 63.10 | 74.74 | 68.92 |
| MedFGMoE-Ada-S96k32-K | 82.50 | 85.10 | 39.71 | 92.39 | 63.55 | 74.93 | 69.24 |
| MedFGMoE-Ada-G12k2 | 84.27 | 87.26 | 37.79 | 91.95 | 60.85 | 75.32 | 68.08 |
| MedFGMoE-Ada-G96k2 | 83.10 | 84.86 | 39.62 | 92.39 | **65.10** | 74.99 | **70.05** |

## 4.2 PERFORMANCE AND EFFICIENCY

We first evaluate whether AEG can improve efficiency without significantly compromising performance. Table 3 compares our AEG models against the original FGMoE and baselines.

Table 4: **Comparison of Expert Grouping Strategies.** Active% denotes the percentage of groups with size $> 0$; AvgSize is the average size of groups with size $> 1$; Collab% is the percentage of groups with size $> 1$ among all active groups; MaxSize is the size of the largest group.

| Strategy | Active% | AvgSize | Collab% | SizeStd | MaxSize |
|---|---|---|---|---|---|
| Co-Occ | - | 16.18 | 18.01 | 1.8 | 17 |
| K-Means | 100 | 2.24 | 45.16 | 0.59 | 3 |
| AEG | 73.61 | 2.52 | 74.48 | 0.95 | 4.79 |

As hypothesized, the benefits of AEG are most pronounced at finer granularities where expert co-occurrence is higher. The G96k2 model achieves an Avg-A score that matches or slightly surpasses the original dense S96k32 model, while activating substantially fewer experts per token. This indicates that AEG can leverage collaborative and partially overlapping expert behaviors to preserve performance under much sparser activation. In contrast, applying AEG to a model with fewer experts (G12k2) leads to a more noticeable performance drop, which is consistent with the lower degree of expert co-activation available for grouping in the 12-expert setting.

Compared with static grouping baselines, AEG also achieves consistently better performance. Both the heuristic co-occurrence based method (C) and the $k$-means clustering method (K) provide some efficiency gains, but they lag behind AEG, highlighting the advantages of learning expert groups jointly with the router and task objective rather than fixing groups in advance.

Beyond performance metrics, AEG offers inherent efficiency benefits. By routing at the group level instead of over a large pool of individual experts, it reduces the computational overhead of dispatching and combining expert outputs. The dynamic sparsity enabled by empty groups further improves efficiency by allowing the model to skip computations on a per-token basis when no additional groups are needed.

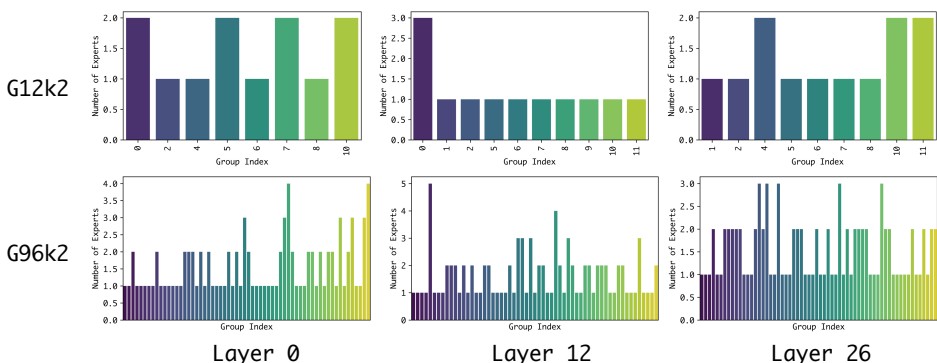

Figure 4: **Visualization of Learned Group Structures.** The bar charts show the size of non-empty groups learned by AEG for models with 12 experts (G12k2, top row) and 96 experts (G96k2, bottom row) at different layers of the network.

### 4.3 ANALYSIS OF LEARNED GROUPS

To better understand what AEG learns, we compare the group structures produced by AEG against those from the static methods and analyze their dynamics during training.

Table 4 summarizes the grouping statistics for different strategies. The heuristic co-occurrence based method tends to form a few large groups along with many singleton groups, reflecting a long-tailed distribution of group sizes. In contrast, $k$-means clustering yields more uniformly sized groups. AEG produces structures that lie between these two extremes: most groups remain small, but a subset of moderately larger groups emerges where expert co-occurrence is strong.

We also study the impact of the maximum number of groups per layer. Setting too high or too low both hurt performance, whereas values around one-half to three-quarters of the number of experts strike a better balance (see Appendix Tab. 11 and Tab. 12 for detailed results).

Figure 4 shows the final sizes of non-empty groups across layers. AEG learns distinct grouping strategies not only between models but also across different layers within the same model. The finer-grained G96k2 model develops more complex and varied group sizes, with some groups containing three or four experts, while the G12k2 model converges to simpler structures, mostly with groups of one or two experts.

Figure 18 provides insight into how these structures emerge. The model quickly prunes the maximum group size from its initial high value and converges to a much smaller and more stable configuration (Figure 18 (a)). For the empty-group selection rate (Figure 18 (c)), the router selects empty groups for roughly 30–40% of tokens, effectively choosing to activate fewer groups than top-$k$ when additional computation is not needed. This confirms that AEG enables input-dependent computational budgets, going beyond a fixed "top-$k$" sparsity policy.

Finally, we analyze the medical semantics of expert groups (Appendix Fig. 19 and 20). Across question types, we observe that one or a few small-sized groups (often singletons) handle the majority of tokens, while token distribution over groups becomes more balanced and dispersed when conditioned on imaging modality. Groups that exhibit clear modality preferences typically have size $\geq 2$ on imaging modality, indicating the emergence of collaborative patterns. These observations support our view of AEG as a mechanism that organizes experts into semantically meaningful, collaboratively functioning units.

## 5 RELATED WORK

Our research synthesizes and extends recent Mixture of Experts (MoE) advancements for the medical multimodal domain. A dominant trend is the shift towards *fine-grained experts* for efficient scaling, a concept investigated by Liu et al. (2023) and adopted in large models such as DeepSeekMoE (Dai et al., 2024), Qwen2.5-MoE (Yang et al., 2024a), and the multimodal ARIA (Li et al., 2024). While prior work has examined expert granularity and routing on general text corpora (Krajewski et al., 2024; Yang et al., 2024b), our work is, to our knowledge, the first to study its effect on generalization and robustness in the data-limited medical setting using real-world test sets. Concurrently, the use of a *shared expert* to preserve common knowledge has become standard practice (Rajbhandari et al., 2022; Costa-Jussà et al., 2022; Xue et al., 2024), often integrated alongside fine-grained experts as in DeepSeekMoE, Qwen2.5-MoE, and ARIA. We adopt this structure but uniquely repurpose the entire, unsplit FFN layer from the dense base model as the shared expert to better retain foundational knowledge. Beyond standard sparse MoE, XMoE introduces Top-$p$ routing (Yang et al., 2024b), which adaptively chooses the number of activated experts per token; this is complementary to our focus on how fine-grained experts form emergent collaborative units that can be routed as groups. Soft-MoE (Puigcerver et al., 2023) instead combines all experts through fully differentiable weighted mixtures, and has been extended to medical imaging (Wu et al., 2025) and multimodal tasks (Shen et al., 2024; Li et al., 2025), trading sparsity for stability. Related cluster- and group-oriented methods such as MoEC (Xie et al., 2023) and MoGE (Kang et al., 2025) regularize routing to encourage local expert clustering or group sparsity. In contrast, we explicitly exploit empirically observed co-activation patterns in fine-grained MoE to learn adaptive, routable expert groups, aiming to reduce redundancy while preserving the collaborative behavior that emerges in medical multimodal scenarios.

## 6 CONCLUSION AND FUTURE WORK

In this work, we demonstrate that a Fine-Grained Mixture of Experts (FGMoE) architecture significantly enhances generalization and robustness in medical multimodal learning. Our proposed Adaptive Expert Grouping (AEG) mechanism effectively mitigates the expert redundancy introduced by fine granularity, learning to group specialists to improve efficiency without performance loss. Future work will focus on refining the AEG mechanism, extending the FGMoE approach to a broader range of medical tasks, and conducting in-depth interpretability analyses to uncover the clinical functions of the learned expert groups, thereby paving the way for more trustworthy medical AI.

REPRODUCIBILITY STATEMENT

We are committed to ensuring the reproducibility of our results. To facilitate this, we provide a comprehensive description of our experimental setup, including dataset details, data preprocessing steps, and all hyperparameter configurations, in Appendix D. All datasets used in our experiments are either publicly available or will be released upon acceptance. We also intend to release our source code, along with scripts for data processing and model training, to further support reproducibility. Detailed explanations of model architectures, training procedures, and evaluation metrics are included in the main text and supplementary materials. We believe these efforts will enable other researchers to easily reproduce and build upon our work.

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

## A  LLM Usage Statement

Large Language Models (LLMs), specifically OpenAI's GPT-4, were utilized as general-purpose assistive tools during the preparation of this work. The uses of LLMs are detailed as follows:

**Code Development:** LLMs were employed for code debugging, error analysis, and offering suggestions to improve code efficiency. Additionally, LLMs assisted in generating scripts for data visualization and figure plotting, based on the authors' descriptions or requirements. **Writing Assistance:** LLMs were used to discuss the structure of the manuscript, help organize sections, and provide suggestions for improving clarity and coherence. The models were also used for language polishing, grammar checking, and rephrasing certain sentences for better readability.

No content was generated by LLMs without subsequent verification and editing by the authors. All scientific ideas, experimental designs, analyses, and conclusions were conceived and validated by the authors. The authors take full responsibility for the accuracy and integrity of the paper's content.

## B  Discussion on Expert Computation Strategies

In this work, we employ our proposed DenseMaskMoE for expert computation. This approach operates in a dense-like manner: all input tokens are processed by every expert, and subsequently, the outputs of irrelevant experts are masked to zero using router-generated weights. The procedure is detailed in Algorithm 1. For context, we also consider the DeepSpeedMoE expert computation (Algorithm 1), which utilizes an explicit token dispatch mechanism.

First, let us define the notation used in the algorithm descriptions: Let $N$ be the number of tokens, $E$ be the number of experts, $H_{in}$ be the input hidden dimension, $H_{out}$ be the output hidden dimension, and $H_{exp}$ be the expert's internal intermediate dimension. Let $K_{cap}$ be the capacity of each expert (relevant for dispatch-based MoEs).

Illustrations of these two computation strategies are presented in Fig. 6. A key advantage of the DenseMaskMoE approach is its potential for increased speed by circumventing the conventional token dispatch mechanism, as depicted in Fig. 7. Furthermore, we have empirically validated that models trained using DenseMaskMoE can be converted for inference with a DeepSpeedMoE-style framework, exhibiting minimal performance fluctuations (see Tab. 5). This suggests that the efficiency of DenseMaskMoE can be effectively leveraged during the exploratory phase to investigate various MoE configurations. The insights gained can then be directly transferred to DeepSpeedMoE-based models for practical sparse inference deployment.

However, we also observed that models trained directly with DeepSpeedMoE exhibit greater performance volatility compared to those trained with DenseMaskMoE (Tab. 5). Concurrently, the conclusions drawn from such directly trained DeepSpeedMoE models show deviations, with a more pronounced performance degradation observed at finer granularities. This discrepancy might potentially limit the extrapolation of our findings to MoE frameworks with different underlying implementations. Alternatively, this observation could suggest that the DeepSpeedMoE framework itself may be less suited for scenarios involving a large number of experts. This aspect warrants further investigation in future research.

In summary, DenseMaskMoE provides an efficient and effective strategy for researchers aiming to explore the diverse configurations and behaviors of Mixture-of-Experts models.

## C  Discussion on Fine-grained Experts with Different Structures

In this work, we utilize the original Feed-Forward Network (FFN) as the Shared Expert. We also examine two alternative configurations: one that omits the Shared Expert, and another that employs a partitioned FFN as the Shared Expert.

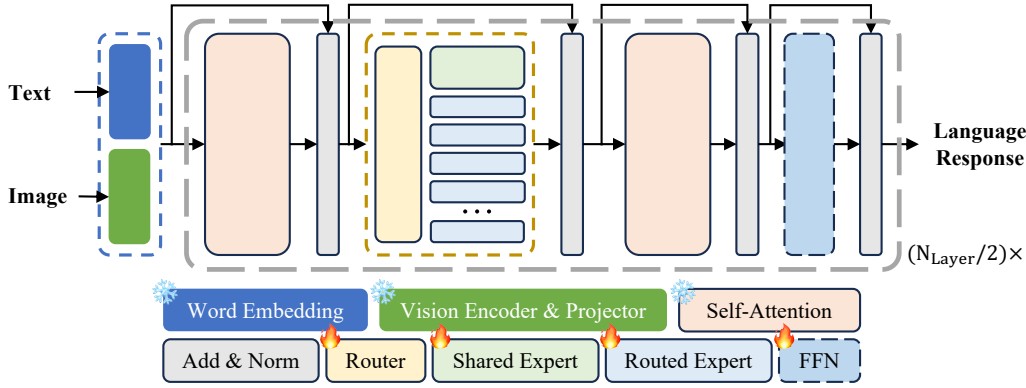

Figure 5: **Overview of the Model Architecture.**

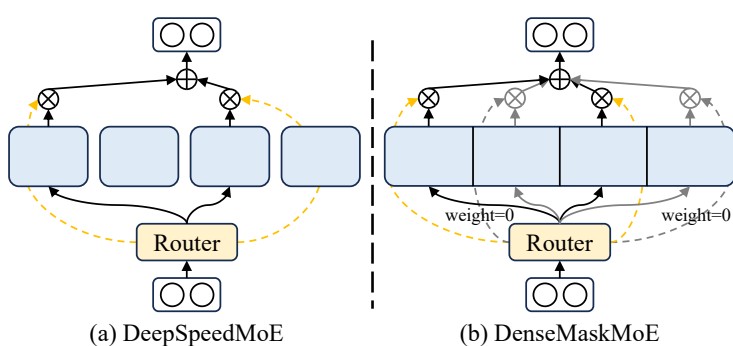

(a) DeepSpeedMoE                (b) DenseMaskMoE

Figure 6: **Different Expert Computation Strategies.** To enhance computational speed (see Fig. 7 for speed optimization details), we adopt a distinct expert computation strategy that differs from DeepSpeedMoE. Specifically, we employ a dense-like approach where tokens are processed through all experts, followed by zeroing out unnecessary expert outputs via router weights. The detailed algorithms are presented in Alg. 1 and Alg. 2. Although this increases the computational workload across experts, it provides speed advantages when expert parallelism is not utilized by avoiding token dispatch operations. We further demonstrate that models trained with DenseMaskMoE can be converted to DeepSpeedMoE for sparse computation during inference, which we term DenseMaskMoE (DS-Infer), without significant performance fluctuation, as shown in Tab. 5.

---

**Algorithm 1** DenseMaskMoE Expert Computation

1: **Input:** Tokens $\mathbf{X} \in \mathbb{R}^{N \times H_{in}}$
2: **Parameters:**
3:     Gating mechanism $\text{Gate}_{\text{dense}}(\cdot)$
4:     Down-projection weights $\mathbf{W}_{down} \in \mathbb{R}^{E \times H_{in} \times H_{exp}}$
5:     Up-projection weights $\mathbf{W}_{up} \in \mathbb{R}^{E \times H_{exp} \times H_{out}}$
        ▷ **1. Gating to obtain aggregated expert scores**
6: $(\mathbf{L}_{aux}, \text{combine\_weights}, \_, \_) \leftarrow \text{Gate}_{\text{dense}}(\mathbf{X})$     ▷ combine_weights $\in \mathbb{R}^{N \times E \times K_{cap}}$
7: $\mathbf{S}_{n,e} \leftarrow \sum_k (\text{raw\_combine\_weights})_{n,e,k}$
        ▷ **2. Fused Down-Projection**
8: Permute and reshape down-projection weights for fusion:
9:     $\mathbf{W}_{down\_fused} \leftarrow \text{reshape}(\text{permute}(\mathbf{W}_{down}, (1, 0, 2)), H_{in}, E \cdot H_{exp})$
10: Compute intermediate representations for all experts simultaneously:
11:     $\mathbf{H}_{inter\_flat} \leftarrow \mathbf{X} \cdot \mathbf{W}_{down\_fused} \quad \in \mathbb{R}^{N \times (E \cdot H_{exp})}$
12: Reshape to separate expert dimensions:
13:     $\mathbf{H}_{inter} \leftarrow \text{reshape}(\mathbf{H}_{inter\_flat}, N, E, H_{exp})$
        ▷ **3. Activation**
14: $\mathbf{A} \leftarrow \text{Activation}(\mathbf{H}_{inter}) \quad \in \mathbb{R}^{N \times E \times H_{exp}}$
        ▷ **4. Weighted Up-Projection and Combination**
15: Compute final output using weighted sum via `einsum`:
16:     $\mathbf{Y} \leftarrow \text{einsum}('ne,neb,ebh \text{-} \text{¿} nh', \mathbf{S}, \mathbf{A}, \mathbf{W}_{up}) \quad \in \mathbb{R}^{N \times H_{out}}$
17: **Output:** $\mathbf{Y}, \mathbf{L}_{aux}$

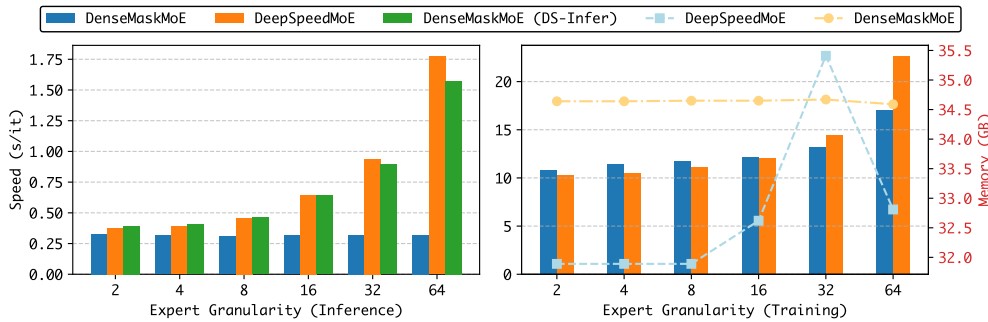

Figure 7: **Impact of Expert Computation Strategies on Speed.** We compare three different expert computation strategies for training and inference speed, with strategy definitions shown in Fig. 7. Speed is measured in seconds per iteration (s/it), where lower values indicate faster performance. For inference, we record the total evaluation time divided by the total number of steps; for training, we record the time for 300 training steps divided by the step count. Notably, for G=64, due to memory constraints, we adjust the batch size to 2 while increasing accumulation steps to 8, with detailed configurations provided in Tab. 7. Experimental results demonstrate that our proposed DenseMaskMoE achieves significant advantages in inference speed and shows superior training performance when G≥32. We also report memory consumption statistics; however, due to the gradient checkpointing strategy employed in DenseMaskMoE, memory usage measurements may not be entirely accurate.

---

**Algorithm 2** DeepSpeedMoE Expert Computation

---

1: **Input:** Tokens $\mathbf{X} \in \mathbb{R}^{N \times H_{in}}$
2: **Parameters:**
3:      Gating mechanism $\text{Gate}_{\text{dispatch}}(\cdot)$
4:      Set of $E$ expert networks $\{f^{(1)}(\cdot), \dots, f^{(E)}(\cdot)\}$, where $f^{(e)} : \mathbb{R}^{H_{in}} \to \mathbb{R}^{H_{out}}$
              ▷ **1. Gating to obtain dispatch mask and combination weights**
5: $(\mathbf{L}_{aux}, \mathbf{C}, \mathbf{D}, \_) \leftarrow \text{Gate}_{\text{dispatch}}(\mathbf{X})$
6:      (Combine weights $\mathbf{C} \in \mathbb{R}^{N \times E \times K_{cap}}$, Dispatch mask $\mathbf{D} \in \{0, 1\}^{N \times E \times K_{cap}}$)
              ▷ **2. Dispatching Tokens to Experts**
7: Dispatch tokens to expert buffers using `einsum` (or equivalent operation):
8:      $\mathbf{X}_{disp} \leftarrow \text{einsum}("nec,nh\text{-}\textgreater ech", \mathbf{D}, \mathbf{X})$    $\in \mathbb{R}^{E \times K_{cap} \times H_{in}}$
              ▷ **3. Expert Computation (Potentially Parallelized)**
9: Initialize expert outputs $\mathbf{Y}_{exp} \in \mathbb{R}^{E \times K_{cap} \times H_{out}}$
10: **for** $e = 1 \to E$ **do**               ▷ For each expert (often in parallel)
11:      $\mathbf{Y}_{exp}[e, :, :] \leftarrow f^{(e)}(\mathbf{X}_{disp}[e, :, :])$        ▷ Expert $e$ processes its $K_{cap}$ tokens
              ▷ **4. Combine Expert Outputs**
12: Combine outputs from experts using `einsum` (or equivalent operation):
13:      $\mathbf{Y} \leftarrow \text{einsum}("nec,ech\text{-}\textgreater nh", \mathbf{C}, \mathbf{Y}_{exp})$    $\in \mathbb{R}^{N \times H_{out}}$
14: **Output:** $\mathbf{Y}, \mathbf{L}_{aux}$

---

# D EXPERIMENTAL SETUP DETAILS

## D.1 PRE-TRAINING DATASETS

Our study utilizes two main pre-training datasets: Ada.

**Ada.** The Ada dataset is constructed by integrating image-caption pairs from PubMed Central (PMC) with synthetically generated visual instructions. It comprises two primary sources of image-caption data: 1. PMC[Raw]: Approximately 470,000 images with human-annotated captions.2. PMC[Refined]: Approximately 510,000 image-caption pairs where captions have been optimized using GPT-4V. These medical image-caption pairs serve dual purposes: they are directly used for image captioning tasks

Table 5: **Impact of Different Expert Computation Strategies on Performance.** This table compares the performance of the DenseMaskMoE and DeepSpeedMoE expert computation strategies when models are fine-tuned directly without pre-training. DenseMaskMoE, utilized in our experiments to enhance computational speed (see Fig. 7 for speed optimization details), differs from DeepSpeedMoE (the conventional expert computation strategy) as illustrated in Fig. 6. Cell background colors indicate the performance change of DeepSpeedMoE relative to DenseMaskMoE: redder shades denote a more significant performance degradation, while greener shades signify a greater improvement. We additionally evaluate models trained with DenseMaskMoE but converted to DeepSpeedMoE for inference, denoted as DenseMaskMoE (DS-Infer).

| Models | In-Distribution Datasets | | | | OOD | | |
| | SLAKE | | PATH-VQA | | OMNI- | Avg-I | Avg-A |
| #Split | Open | Closed | Open | Closed | MINI | | |
| *DeepSpeedMoE vs. DenseMaskMoE* | | | | | | | |
| S6k2 | 1.56 | -1.44 | -0.50 | -0.35 | -0.70 | -0.59 | -0.09 |
| S12k4 | -0.03 | 0.00 | -0.57 | -0.18 | 2.80 | -0.67 | 0.11 |
| S24k8 | -0.38 | -0.24 | 1.04 | 0.35 | 0.05 | 0.89 | 0.82 |
| S48k16 | 0.49 | 0.72 | 0.03 | -0.03 | 2.70 | 0.17 | 0.51 |
| S96k32 | -0.71 | 0.72 | 0.12 | 0.21 | 1.25 | -0.29 | -0.05 |
| S192k64 | -0.70 | -0.48 | 0.17 | 0.12 | -1.05 | 0.56 | 0.10 |
| *DenseMaskMoE (DS-Infer) vs. DenseMaskMoE* | | | | | | | |
| S6k2 | -0.50 | -0.24 | -0.24 | 0.18 | -0.95 | 0.17 | -0.05 |
| S12k4 | -0.01 | 0.24 | 0.16 | -0.09 | -0.05 | 0.00 | -0.03 |
| S24k8 | -0.05 | 0.00 | 0.02 | 0.09 | -0.30 | 0.10 | 0.01 |
| S48k16 | 0.27 | 0.48 | 0.09 | 0.03 | 0.35 | 0.33 | 0.29 |
| S96k32 | 0.16 | 0.00 | 0.14 | 0.00 | 0.15 | 0.05 | 0.07 |
| S192k64 | -0.16 | 0.24 | 0.01 | -0.12 | 0.25 | -0.19 | -0.05 |
| S12k4-Ada | 0.14 | -0.72 | 0.12 | 0.03 | 0.05 | -0.07 | -0.07 |

Table 6: **Ablation Study on the Shared Expert Architecture.** All MoE models shown were fine-tuned directly on downstream tasks without any MoE-specific pre-training. We compare performance with no shared expert, a full FFN shared expert (our method), and a segmented shared expert. We use "s" to denote the segmented expert. Cell backgrounds are color-coded, where greener indicates better performance and redder indicates worse.

| Models | In-Distribution Datasets | | | | OOD | | |
| Datasets | SLAKE | | PATH-VQA | | OMNI- | | |
| #Split | Open | Closed | Open | Closed | MINI | Avg-I | Avg-A |
| #Samples | 645 | 416 | 3370 | 3391 | 2000 | | |
| #Metrics | Recall | Acc | Recall | Acc | Acc | | |
| *Baselines* | | | | | | | |
| Qwen2-VL-2B-Instruct | 82.30 | 83.65 | 36.93 | 92.24 | 60.75 | 73.78 | 67.27 |
| MoE-Qwen2-VL-4k2 | 84.44 | 84.86 | 36.28 | 91.89 | 62.35 | 74.37 | 68.36 |
| MoE-Qwen2-VL-S3k1 | 81.55 | 85.82 | 36.47 | 91.54 | 58.10 | 73.84 | 65.97 |
| *w/o Shared Expert* | | | | | | | |
| MedFGMoE-8k4 | 82.97 | 82.69 | 35.30 | 91.71 | 53.35 | 73.17 | 63.26 |
| MedFGMoE-16k8 | 83.08 | 84.13 | 34.72 | 90.86 | 37.85 | 73.20 | 55.52 |
| *w/ Segmented Shared Expert* | | | | | | | |
| MedFGMoE-s7k3 | 82.51 | 85.58 | 35.69 | 92.21 | 58.95 | 74.00 | 66.47 |
| MedFGMoE-s15k7 | 82.76 | 83.17 | 34.98 | 91.95 | 50.65 | 73.21 | 61.93 |
| *w/ Original Shared Expert* | | | | | | | |
| MedFGMoE-S6k2 | 82.43 | 86.06 | 36.71 | 91.98 | 56.30 | 74.29 | 65.30 |
| MedFGMoE-S12k4 | 82.91 | 84.86 | 37.45 | 92.10 | 59.00 | 74.33 | 66.66 |
| MedFGMoE-S24k8 | 83.00 | 86.06 | 36.75 | 91.65 | 61.00 | 74.36 | 67.68 |
| MedFGMoE-S48k16 | 82.68 | 84.13 | 36.40 | 92.48 | 61.75 | 73.92 | 67.84 |
| MedFGMoE-S96k32 | 83.32 | 84.38 | 36.16 | 91.95 | 60.95 | 73.95 | 67.45 |
| MedFGMoE-S192k64 | 83.36 | 83.89 | 36.52 | 92.24 | 62.45 | 74.01 | 68.23 |

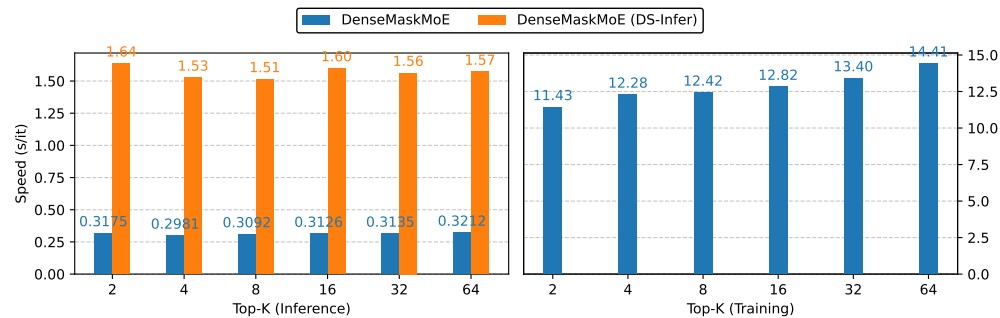

Figure 8: **Effect of Top-$k$ on Speed.** Based on the "S192k64" configuration, we evaluate how the number of activated experts affects training and inference speed. Results show that compared to the number of experts (see Tab. 7), the impact of top-$k$ is not significant.

and also act as input for a visual instruction synthesizer. This synthesizer, a fine-tuned Multimodal Large Language Model (MLLM), generates task triplets consisting of an instruction, an information-rich answer, and a precise answer. These triplets undergo a consistency-based filtering process and are structured into a chain-of-thought format. This process yielded approximately 150,000 synthetic visual instruction tasks from PMC$^{\text{Raw}}$ and 144,000 from PMC$^{\text{Refined}}$. Finally, the original medical image-captioning tasks and these synthetic medical visual instruction tasks are combined in a multi-turn dialogue format for single-stage post-training, aiming to enhance the model's capabilities in the biomedical domain.

## D.2 FINE-TUNING AND EVALUATION DATASETS

We utilize several established medical Visual Question Answering (VQA) datasets for fine-tuning and evaluation. The descriptions below are adapted from their respective publications.

- **VQA-RAD** Lau et al. (2018): Includes clinician-generated QA pairs and radiology images covering the head, chest, and abdomen. Questions are categorized into 11 types, with answers being either "OPEN" (short text) or "CLOSED" (yes/no).

- **SLAKE** Liu et al. (2021): A semantically-labeled, knowledge-enhanced medical VQA dataset featuring radiology images and diverse QA pairs annotated by physicians. It includes semantic segmentation masks, object detection bounding boxes, and covers various body parts. "CLOSED" answers are yes/no, while "OPEN" answers are one-word or short phrases. We use only the English subset.

- **PathVQA** He et al. (2020): Consists of pathology images with QA pairs focusing on aspects like location, shape, and color. Questions are categorized as "OPEN" (open-ended) or "CLOSED" (closed-ended).

- **OMNI-VQA-MED-MINI**: This is a curated subset of 2,000 samples from OmniMed-VQA Hu et al. (2024). OmniMedVQA is a large-scale benchmark comprising 118,010 images and 127,995 QA items from 73 existing medical datasets, covering 12 imaging modalities and over 20 anatomical regions. Its QA pairs were generated by converting attributes from original medical classification datasets into a multiple-choice VQA format using GPT's contextual reasoning. Our OMNI-VQA-MED-MINI subset includes 250 samples from each of 8 distinct modalities.

## D.3 IMPLEMENTATION DETAILS

The specific configurations for the Mixture-of-Experts (MoE) model are detailed in Tab. 7. The hyperparameters employed during training are presented in Tab. 8.

Table 7: **Configuration of MoE Models.** $N_{\text{act}}$ represents the number of activated experts. For count, $a + b$ denotes $a$ shared experts and $b$ routed experts. For $N_{\text{act}}$, $a + b$ denotes $a$ activated shared experts and $b$ activated routed experts. "*" indicates that the FFN structure of Phi differs from other models. "†" indicates that when using DenseMaskMoE, all parameters participate in computation, but only 2.79B parameters contribute to the actual output.

| Models | Experts | | | Hidden Size | Layers | # Params | | | | |
|---|---|---|---|---|---|---|---|---|---|---|
| | Count | $N_{act}$ | Size | | | Total | Activated | V-Encoer | Expert $(10^6)$ | Router $(10^6)$ |
| Qwen2-VL-2B-Instruct | 1+0 | 1+0 | 8960 | 1536 | 28 | 2.21B | 2.21B | 0.67B | 41.29 | - |
| LLaVA-Med (Mistral-7B) | 1+0 | 1+0 | 14336 | 4096 | 32 | 7.57B | 7.57B | 0.3B | 176.16 | - |
| Med-MoE (Phi2-2.7B) | 1+4 | 1+2 | 10240* | 2560 | 32 | 6.44B | 4.77B | 0.3B | 52.44 | 0.16 |
| MoE-Qwen2-VL-4k2 | 0+4 | 0+2 | 8960 | 1536 | 28 | 3.94B | 2.79B | 0.67B | 41.29 | 0.09 |
| MoE-Qwen2-VL-S3k1 | 1+3 | 1+1 | 8960 | 1536 | 28 | 3.94B | 2.79B | 0.67B | 41.29 | 0.06 |
| MedFGMoE-SNkM | 1+N | 1+M | 8960÷M | 1536 | 28 | 3.94B | 2.79B† | 0.67B | 41.29÷M | 0.06×M |

Table 8: **Experimental Hyperparameters.** "‡" denotes that for S192k64, the per-GPU batch size was adjusted to 2 and gradient accumulation steps to 8 due to memory constraints. For S96k32, the per-GPU batch size was set to 2 and gradient accumulation steps to 8 during pre-training due to memory limitations.

| Configuration | Pretrain | | | Finetune | | | |
|---|---|---|---|---|---|---|---|
| | Qwen2-VL | MoE-Qwen2-VL | MedFGMoE | AdaMLLM | MoE-Qwen2-VL | MedFGMoE | Med-MoE |
| Image Encoder | Qwen2-VL ViT | | | | | | CLIP-Large ViT |
| Feature Select Layer | - | | | | | | -2 |
| Image Min Pixels | $16 \times (28 \times 28)$ | | | | | | $336 \times 336$ |
| Image Max Pixels | $576 \times (28 \times 28)$ | | | | | | |
| Image Projector | 2 Linear layers with GeLU | | | | | | |
| Epoch | 1 | | | 5 | | | 9 |
| Per-GPU Batch Size | 4‡ | | | | | | - |
| Grad Accum Steps | 4‡ | | | | | | - |
| Text Max Length | 1024 | | | | | | 2048 |
| Learning Rate | 2e-5 | | | | | | |
| Scheduler Type | Cosine | | | | | | |
| Weight Decay | 0.0 | | | | | | |
| Warm-Up Ratio | 0.03 | | | | | | |
| Capacity Factor | - | 1.5 | | - | 1.5 | | |
| Aux Loss Coef | - | 0.01 | | - | 0.01 | | - |
| Train Modules | FFN | FFN + MoE (Gate + Experts) | | Full | FFN + MoE (Gate + Experts) | | FFN + MoE (Experts) |
| ¿ Training Params | 1.16B | 2.89B | | 2.21B | 2.89B | | 5.03B |
| DeepSpeed | Zero2 | | | | | | |
| Precision | BF16 | | | | | | |
| GPU | $2 \times$ NVIDIA A100-SXM4-40GB | | | | | | - |

Table 9: **Prompt Templates for the Evaluated MedVQA Datasets**

| Datasets | Instruction |
|---|---|
| VQA-RAD | {question} |
| SLAKE | {question} |
| PATH-VQA | {question} |
| VQA-Med | {question} |
| PMC-VQA | {input} Analyze the image and select the correct option. Format your response as: ``Answer: (X) – [brief one-sentence justification]'' |
| OMNI-MINI | Medical question with several options; only one is correct. Refer to the image for clues. {input} Analyze the image and select the correct option. Format your response as: ``Answer: (X) – [brief one-sentence justification]'' |

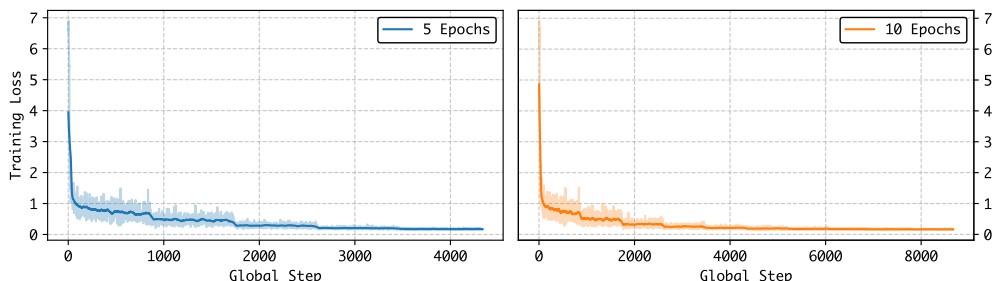

Figure 9: **Training Loss Comparison: 5 vs. 10 Epochs.** The loss curves demonstrate that training loss stabilizes around 4,000 steps (approximately 5 epochs). The latter half of 10-epoch training shows minimal loss variation, suggesting that 5 epochs represents a more efficient training configuration compared to 10 epochs.

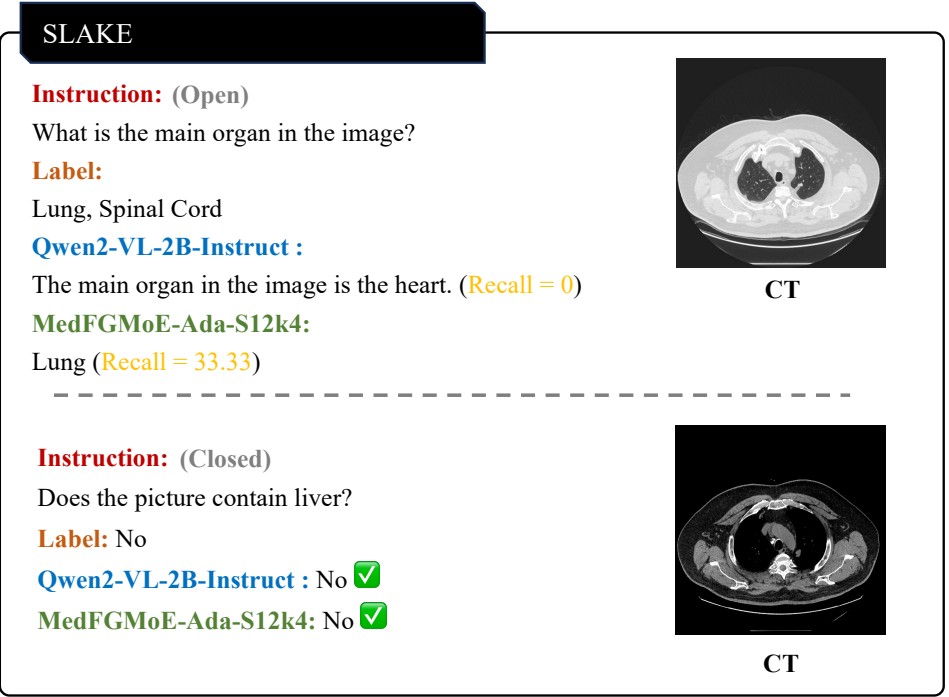

Figure 10: **Example From SLAKE Dataset With Model Outputs Comparison.**

## E    JACCARD SCORE FOR EXPERT SIMILARITY ANALYSIS

To quantitatively assess the similarity in activation patterns between pairs of experts within our Mixture-of-Experts (MoE) layer, we employ the Jaccard Score.

**Definition and Rationale**    The Jaccard Score, also known as the Jaccard Index or Intersection over Union (IoU), is a statistic used to gauge the similarity between two finite sets. For two sets, $A$ and $B$, it is defined as the size of their intersection divided by the size of their union:

$$J(A, B) = \frac{|A \cap B|}{|A \cup B|} \tag{4}$$

In our context, for any two experts, say expert $i$ and expert $j$, we consider the set of input tokens that activate expert $i$ (denoted $T_i$) and the set of tokens that activate expert $j$ (denoted $T_j$). The Jaccard Score $J(E_i, E_j)$ measures the similarity of their activation patterns.

Figure 11: **Example From Path-VQA Dataset With Model Outputs Comparison.**

While a raw co-occurrence matrix (as computed by `count_expert_cooccurrence`) provides the absolute count of tokens that simultaneously activate two experts ($|T_i \cap T_j|$), it can be skewed by experts that are individually activated very frequently. For instance, a highly active expert might show large co-occurrence counts with many other experts, even if the proportional overlap is not particularly significant. The Jaccard Score normalizes for the individual activation frequencies of each expert by considering the union of their activated token sets in the denominator. This provides a more robust measure of similarity, highlighting pairs of experts that share a significant fraction of their workload relative to their overall activity.

**Utility in MoE**   In the MoE framework, a high Jaccard Score between two experts $E_i$ and $E_j$ indicates that they tend to be activated by the same subset of input tokens. This suggests that these experts might be learning similar functions or specializing in similar types of input features. Consistently high Jaccard Scores between specific pairs could indicate potential redundancy among experts, which might inform decisions about model compression or expert merging. Conversely, low Jaccard Scores across most pairs would suggest good expert specialization, where different experts handle distinct data subsets.

**Calculation Principle**   Given the co-occurrence count for experts $i$ and $j$, denoted $\text{co\_occ}(i, j) = |T_i \cap T_j|$, and the total number of tokens activating expert $i$, denoted $\text{act}(i) = |T_i|$, and similarly for

---

**OmniMedVQA-MINI**

**Instruction:**
Medical question with several options; only one is correct. Refer to the image for clues.
The Question is: What is the specific abnormality present in this image?
The candidate Options are:
(A) Advanced glaucoma.
(B) Moderate diabetic retinopathy.
(C) Mild cataracts.
(D) Severe macular degeneration.
Analyze the image and select the correct option. Format your response as:
'Answer: (X) - [brief one-sentence justification]'
**Label:** B
**Qwen2-VL-2B-Instruct :** Answer: (B) Moderate diabetic retinopathy. ✅
**MedFGMoE-Ada-S12k4:** Answer: (B) Moderate diabetic retinopathy. ✅

**Fundus Photography**

- - - - - - - - - - - - - - - - - - - - - - - - - - - - - - -

**Instruction:**
Medical question with several options; only one is correct. Refer to the image for clues.
The Question is: What are the characteristics of the anomaly depicted in this image?
The candidate Options are:
(A) There is a dark spot in the periphery of the retina.
(B) There is a small hole in the macula, which is the central part of the retina.
(C) The abnormality is characterized by a jagged edge around the macula.
(D) The abnormality is a result of a detached retina.
Analyze the image and select the correct option. Format your response as: 'Answer: (X) - [brief one-sentence justification]'
**Label: B**
**Qwen2-VL-2B-Instruct :** Answer: (C) The abnormality is characterized by a jagged edge around the macula. ❌
**MedFGMoE-Ada-S12k4:** Answer: (B) There is a small hole in the macula, which is the central part of the retina. ✅

**OCT**

Figure 12: **Example From OMNI-VQA-Med-MINI Dataset With Model Outputs Comparison.**

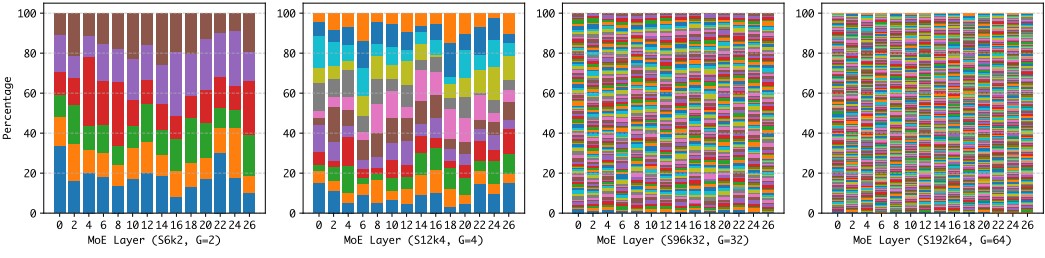

Figure 13: **Expert Load Distribution for Path-VQA Dataset.**

expert $j$, $\text{act}(j) = |T_j|$, the Jaccard Score is calculated as:

$$J(E_i, E_j) = \frac{\text{co\_occ}(i, j)}{\text{act}(i) + \text{act}(j) - \text{co\_occ}(i, j)} \tag{5}$$

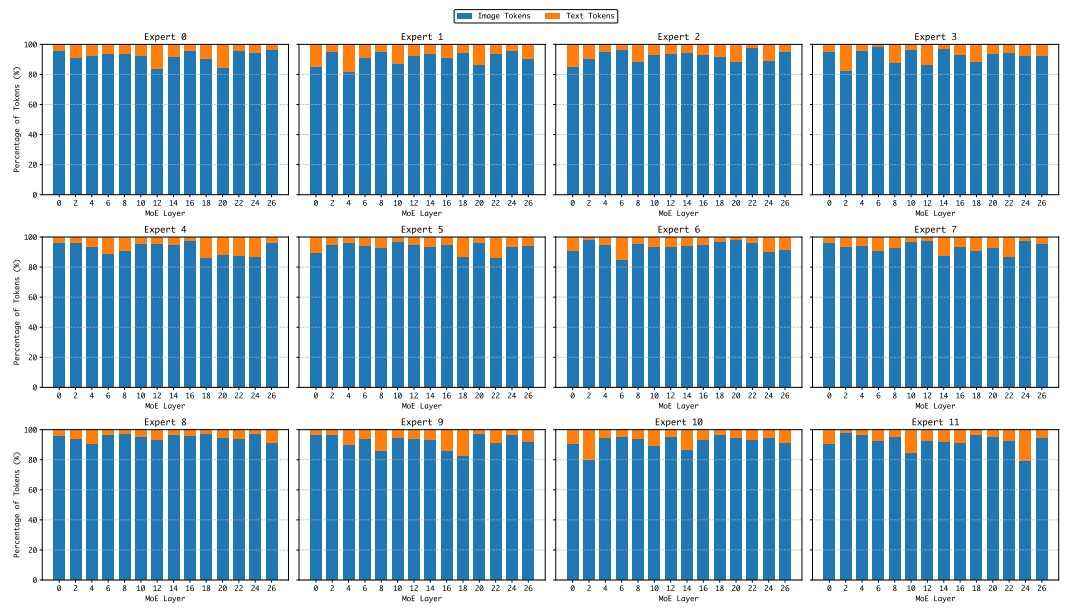

Figure 14: **Expert Specialization Across Layers in MoE Model (Path-VQA)**

This corresponds to the implementation in `compute_jaccard_scores`, where `co_occurrence[i, j]` is co_occ$(i, j)$ and `expert_activation_counts[i]` is act$(i)$.

**Expected Jaccard Score under Random Routing** To provide a baseline for interpreting observed Jaccard Scores, we can calculate the expected Jaccard Score if experts were routed to tokens randomly. Let $N_E$ be the total number of experts available and $k$ be the number of experts activated for each token. For any given token, the probability of it activating a specific expert $E_i$ is $P(E_i) = k/N_E$. The probability of a token activating both expert $E_i$ and expert $E_j$ (assuming selection of $k$ distinct experts without replacement for that token) is $P(E_i \cap E_j) = \frac{k}{N_E} \cdot \frac{k-1}{N_E-1}$. The Jaccard Score is the probability of joint activation divided by the probability of activating at least one of them:

$$J_{\text{random}}(E_i, E_j) = \frac{P(E_i \cap E_j)}{P(E_i) + P(E_j) - P(E_i \cap E_j)} \tag{6}$$

Substituting the probabilities and simplifying, we get:

$$J_{\text{random}} = \frac{\frac{k(k-1)}{N_E(N_E-1)}}{\frac{k}{N_E} + \frac{k}{N_E} - \frac{k(k-1)}{N_E(N_E-1)}} \tag{7}$$

$$= \frac{\frac{k(k-1)}{N_E-1}}{\frac{2k(N_E-1) - k(k-1)}{N_E-1}} \tag{8}$$

$$= \frac{k(k-1)}{2k(N_E-1) - k(k-1)} \tag{9}$$

$$= \frac{k-1}{2(N_E-1) - (k-1)} \tag{10}$$

$$= \frac{k-1}{2N_E - 2 - k + 1} \tag{11}$$

$$= \frac{k-1}{2N_E - k - 1} \tag{12}$$

This theoretical value $J_{\text{random}}$ (referred to as `jaccard_random` in the provided analysis script, with $N_E$ as `N_sim` and $k$ as `k_sim`) serves as a crucial reference. Observed Jaccard Scores signif-

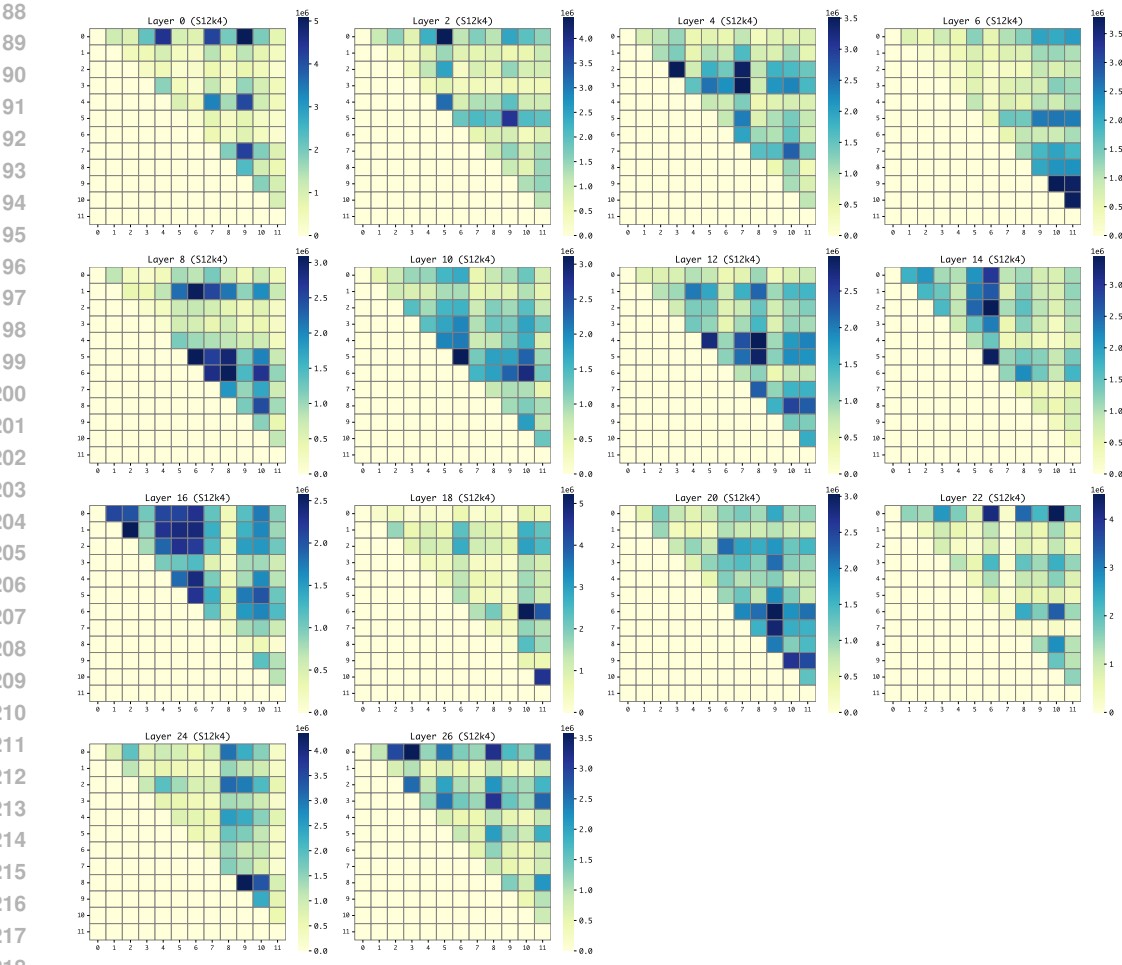

Figure 15: **Expert Co-activation Heatmaps Across Layers. (S12k4)**

icantly deviating from this random baseline can indicate meaningful learned routing strategies or specialization patterns among the experts.

## F ANALYSIS OF GRANULARITY EFFECTS AND AEG METHOD ACROSS BROADER DOMAINS

We have extended our experiments beyond medical VQA to include additional tasks and non-medical domains:

1. **Medical Report Generation (MRG):** This task requires generating clinical findings and impressions from one or more medical images, representing a significantly more complex form of multimodal reasoning than VQA. We trained our models on a subset of MIMIC-CXR and evaluated on both MIMIC-CXR (as the in-distribution/ID dataset) and IU-Xray (as the out-of-distribution/OOD dataset). We adopted RadEntityMatchExact (RaTE) as our primary metric, as it more accurately reflects the semantic correctness of clinical entities compared to traditional language generation metrics like BLEU or ROUGE.

2. **Non-medical Domain (Food VQA):** We selected the food domain as a non-medical testbed to assess cross-domain generalization. We used a subset of the Food101 dataset, formulated as a VQA task for food category classification. The dataset was split into three subsets: train,

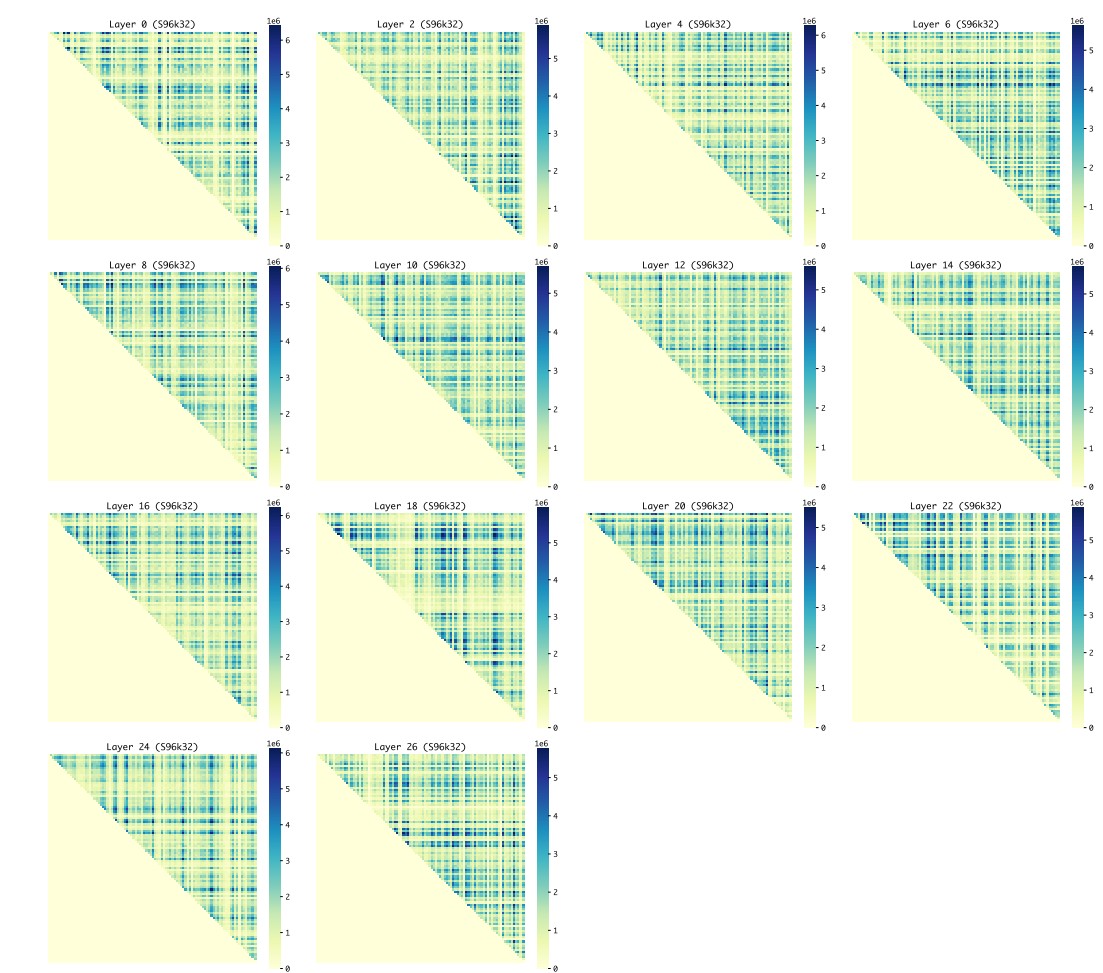

Figure 16: **Expert Co-activation Heatmaps Across Layers. (S96k32)**

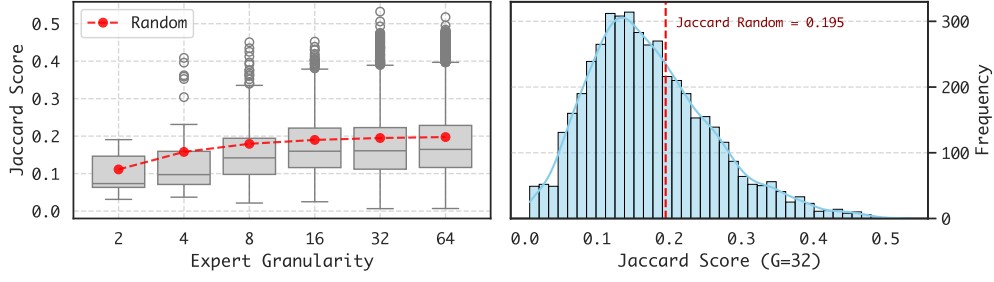

Figure 17: **Impact of Expert Granularity on Co-activation Patterns. (Layer 0)**

test (ID), and test (OOD), where the OOD set contains different food categories from those in the training and ID test sets.

**Analysis of Trade-offs Across Tasks**    The results are presented in Table 10. We first examined whether similar trade-offs—improved generalization with slightly reduced fitting capacity as granularity increases—exist in MRG and Food VQA tasks.

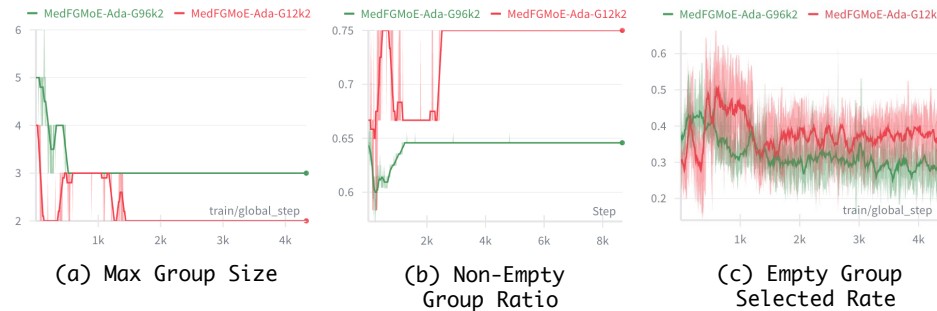

(a) Max Group Size

(b) Non-Empty
Group Ratio

(c) Empty Group
Selected Rate

Figure 18: **Training Dynamics of Adaptive Expert Grouping.** The plots show the evolution of (a) maximum group size, (b) the ratio of non-empty groups, and (c) the rate at which empty groups are selected by the router during training for G12k2 (red) and G96k2 (green) models.

Table 10: **Cross-Task and Cross-Domain Evaluation**

| Models | G | MRG | | Food VQA | |
|---|---|---|---|---|---|
| | | ID | OOD | ID | OOD |
| | | MIMC-CXR RaTE | IU-Xray RaTE | Food101 Acc. | Food101* Acc. |
| FGMoE-S6k2 | 2 | 49.88 | 54.84 | 93.70 | 38.40 |
| FGMoE-S12k4 | 4 | **52.18** | 53.95 | 94.40 | 31.70 |
| FGMoE-S24k8 | 8 | 52.05 | **56.92** | 94.50 | 36.10 |
| FGMoE-S48k16 | 16 | 49.11 | 52.71 | 94.60 | 38.10 |
| FGMoE-S96k32 | 32 | 49.02 | 52.50 | **94.70** | 39.60 |
| FGMoE-S192k64 | 64 | 49.78 | 53.01 | 94.40 | **41.00** |
| FGMoE-Ada-S12k4 | 4 | 51.74 | 53.26 | 94.70 | 24.70 |
| FGMoE-Ada-G12k2 | 4 | 51.91 | 53.74 | 94.70 | 26.60 |
| FGMoE-Ada-S96k32 | 32 | 51.65 | 53.83 | **94.80** | 28.30 |
| FGMoE-Ada-G96k2 | 32 | **52.91** | **54.73** | 94.60 | **34.10** |

For **Food VQA**, we observe improved OOD generalization as granularity increases. However, unlike medical VQA where ID performance decreases slightly, the ID performance here fluctuates rather than showing a consistent decline. We attribute this to the relatively lower semantic complexity of food classification compared to medical reasoning tasks. Meanwhile, we observe a similar increase in expert co-occurrence as in the medical domain, as shown in Figure 21.

For **Medical Report Generation**, we do not observe the same clear trade-off pattern when granularity ($G$) increases from 2 to 64. However, when examining our pre-trained models FGMoE-Ada-S12k4 and FGMoE-Ada-S96k32, we observe the characteristic trend consistent with our findings: higher granularity (S96k32) yields better OOD performance (53.83 vs. 53.26) but slightly lower ID performance (51.65 vs. 51.74). We hypothesize that this difference stems from the high semantic complexity of report generation compared to VQA tasks. The task's complexity makes it difficult to observe the expected trade-off curve in directly fine-tuned models without sufficient pre-training.

**Effectiveness of AEG Across Tasks**   We also evaluated the effectiveness of our AEG method across these tasks. Consistent with our hypothesis, S12k4 performs worse than S96k32, as finer granularity induces more pronounced expert co-activation patterns, making AEG more applicable and beneficial. Importantly, across both medical and food domains, G96k2 (our AEG variant) demonstrates better generalization than S96k32, confirming the effectiveness of our adaptive grouping approach in mitigating the redundancy introduced by fine-grained expert collaboration while maintaining the benefits of increased granularity.

Table 11: **Effect of Maximum Group Number on Model Performance**

| N_group | Avg-I | OOD | Avg-A |
|---------|-------|-------|-------|
| 24 | 75.26 | 62.50 | 69.09 |
| 48 | 74.99 | 62.75 | 69.16 |
| 72 | 74.99 | 65.10 | 69.30 |
| 96 | 74.74 | 63.10 | 68.92 |

Table 12: **Effect of Maximum Group Number on Grouping Strategies**

| #Groups | Scope | Active% | AvgSize | Collab% | SizeStd | MaxSize |
|---------|-------|---------|---------|---------|---------|---------|
| 96 | Avg (all) | 62.5 | 2.37 | 64.96 | 0.81 | 4.43 |
|  | Layer 0 | 64.58 | 2.36 | 61.46 | 0.78 | 4 |
|  | Layer 12 | 61.46 | 2.37 | 66.67 | 0.84 | 5 |
|  | Layer 26 | 64.58 | 2.17 | 65.62 | 0.64 | 3 |
| 72 | Avg (all) | 73.61 | 2.52 | 74.48 | 0.95 | 4.79 |
|  | Layer 0 | 68.06 | 2.74 | 77.08 | 1.16 | 5 |
|  | Layer 12 | 75 | 2.5 | 72.92 | 0.92 | 5 |
|  | Layer 26 | 80.56 | 2.31 | 69.79 | 0.78 | 4 |
| 48 | Avg (all) | 87.5 | 2.88 | 86.31 | 1.23 | 5.71 |
|  | Layer 0 | 83.33 | 2.87 | 89.58 | 1.26 | 6 |
|  | Layer 12 | 91.67 | 2.68 | 86.46 | 1.07 | 5 |
|  | Layer 26 | 91.67 | 2.93 | 82.29 | 1.37 | 6 |
| 24 | Avg (all) | 100 | 4.25 | 98.14 | 1.87 | 8.36 |
|  | Layer 0 | 100 | 4 | 100 | 1.38 | 7 |
|  | Layer 12 | 100 | 4.27 | 97.92 | 2.06 | 11 |
|  | Layer 26 | 100 | 4.13 | 98.96 | 1.78 | 9 |

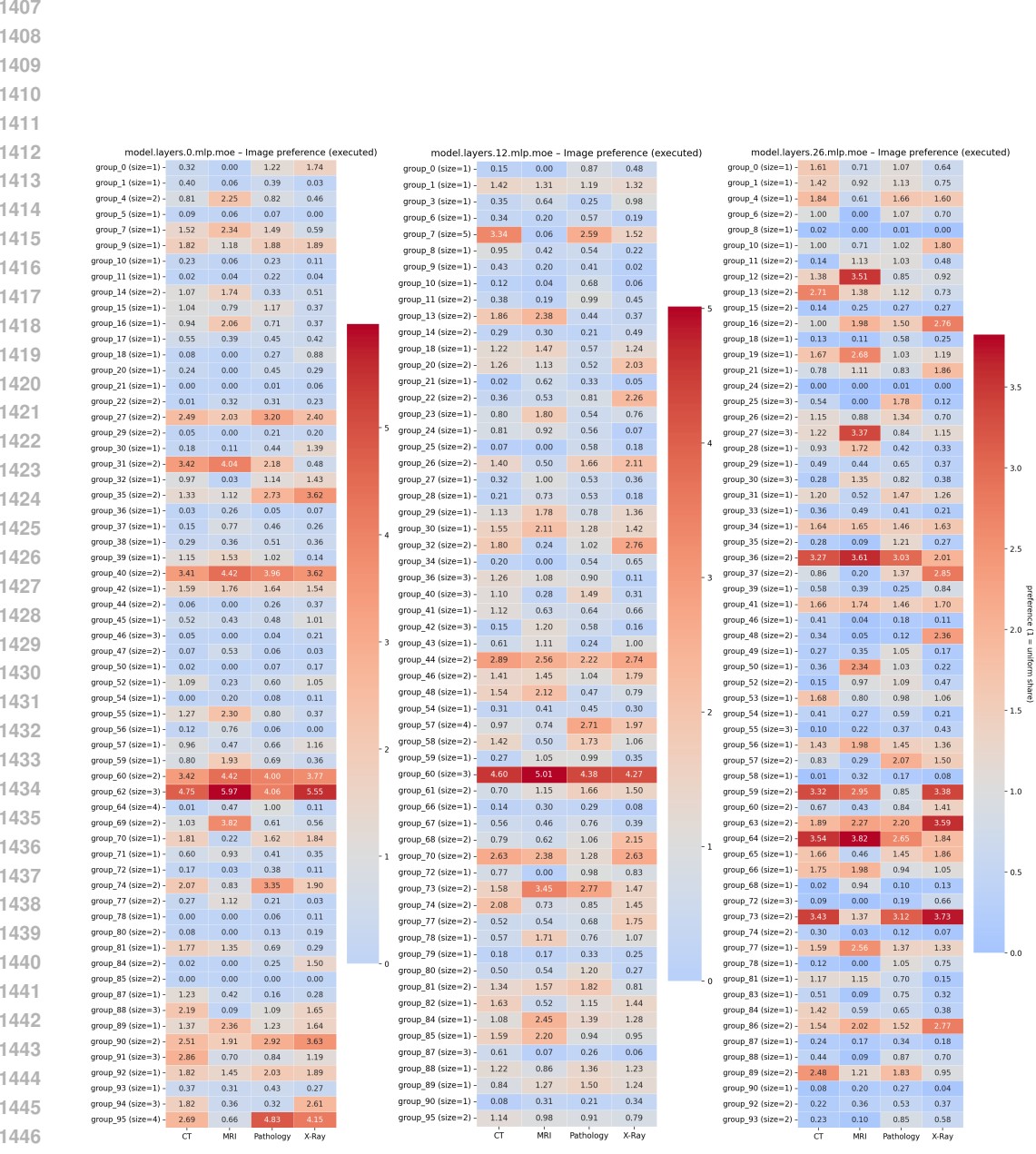

Figure 19: **Heatmap of Expert Group Preferences for Image Modalities (CT, MRI, Pathology, X-Ray) in FGMoE-Ada-G96k2**.

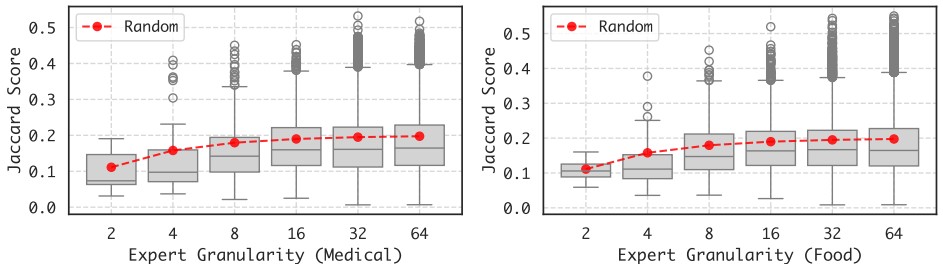

Figure 20: **Heatmap of Expert Group Preferences for Question Types in FGMoE-Ada-G96k2**.

Figure 21: **Impact of Expert Granularity on Co-occurrence Patterns Across Medical and Food Domains**.