# OpenReview forum: "Fine-Grained Mixture of Experts for Medical Multimodal Learning"
_ICLR.cc/2026/Conference — Submitted to ICLR 2026_

### Official Review · Reviewer_7da9 · 2025-10-20

**Soundness:** 3
**Presentation:** 2
**Contribution:** 3
**Rating:** 4
**Confidence:** 4

**Summary:**

This paper presents the first systematic study of Fine-Grained Mixture of Experts (FGMoE) for medical multimodal learning. The authors investigate how expert granularity affects model performance. Key findings include that increasing granularity significantly improves out-of-distribution generalization and robustness but creates substantial functional redundancy among experts, adding unnecessary computational burden to routing. Main contributions include a shared expert that preserves pre-trained knowledge while enabling fine-grained specialization, and the Adaptive Expert Grouping (AEG) framework that dynamically clusters functionally similar experts into groups, shifting routing from expert-level to group-level to reduce computational overhead.
Results show their finest-grained model (96 experts) achieves best overall performance, while AEG maintains comparable performance with substantially fewer activated experts.

**Strengths:**

1. The paper reveals the dual effect of granularity, which is a fundamental trade-off that hasn't been systematically studied. It provides practical guidance that identifies G=16 as an inflection point where benefits plateau.

2. Empirical evaluation is thorough, including a systematic granularity study, different types of baselines, and evaluation paradigms, including zero-shot, fine-tuned, and robustness.

3. The proposed AEG mechanism can be learned jointly with task objectives and enables adaptive computation by allowing empty group selection.

**Weaknesses:**

1. The motivation is somewhat unclear; the paper claims that it focuses on medical applications, but the overall design of the approach is generic, rather than addressing a particular medical need. While the evaluation focused mostly on the medical datasets, it lacked detailed clinical insights into interpreting the results. For example, what does the routing modality distribution across layers reveal about the actual clinical patterns learned by the model?

2. Also on medical multimodal evaluation, it seems that the experiments are narrowly focused on VQA. While the model itself is not limited to certain tasks, the authors could consider extending this to a broader range of clinical tasks such as diagnostic tasks, report generation, bounding-box reasoning, etc.

3. AEG is presented as the solution to routing overhead, but no comparison with other efficiency approaches is provided. For example, expert pruning, hierarchical routing etc.

**Questions:**

1. In Figure 2, it is unclear where the matrices $A$, $E_g$ and $E_e$ are located.

2. In line 292, intuitively, as granularity increases, the router should tend to route tokens to different experts due to extra choice; but why does it turn out they are sent to the same expert pairs where they constantly process the same input?

---

> ### Author Response · Authors · 2025-11-24
>
> Dear Reviewer 7da9,
>
> Thank you for your thoughtful feedback and constructive suggestions. We appreciate your attention to the clinical relevance of our work and the breadth of our experimental evaluation. Below, we address each of your concerns in detail.
>
> ---
>
> ## Response to Weaknesses
>
> ### W1: The motivation is somewhat unclear; lacks detailed clinical insights
>
> Thank you for raising this important point. We would like to clarify our motivation and provide the clinical insights you requested.
>
> #### Research Motivation
>
> Our work is motivated by a critical gap in medical AI research. While fine-grained MoE architectures have demonstrated effectiveness in general domains, their potential in medical multimodal settings remains unexplored. Medical data presents unique challenges: multi-modality heterogeneity, multi-task complexity, and high knowledge density that demands comprehensive cross-modal reasoning. We hypothesized that fine-grained MoE's two key advantages， flexible expert combinations and fine-grained knowledge representation， would be particularly valuable for these medical challenges.
>
> However, a fundamental question remained unanswered: what degree of expert granularity is optimal for medical domains? While [1] showed that finer granularity improves performance on general corpora, this relationship had not been systematically studied in specialized medical contexts. Our work addresses this empirical gap through controlled experiments on medical multimodal tasks.
>
> Importantly, during this investigation, we discovered a novel phenomenon: as granularity increases, certain experts exhibit pronounced co-activation patterns. This observation inspired AEG (Adaptive Expert Grouping), which explicitly transforms these implicit collaborations into learnable structures. While developed for medical applications, the core mechanism is domain-agnostic and generalizable.
>
> #### Clinical Interpretation of Routing Patterns
>
> We appreciate your request for deeper clinical insights. We have now conducted comprehensive analyses of routing modality distributions, with detailed results presented in Figures 19 and 20 in the appendix.
>
> **Analysis Methodology:**
>
> We computed a relative preference coefficient to quantify each expert group's affinity for different modalities and question types:
>
> $$\text{preference}(g, m) = N_G \times \frac{\text{tokens routed to group } g \text{ from modality } m}{\text{total tokens of modality } m}$$
>
> where $N_G$ denotes the number of active groups. A preference value \> 1 indicates that group $g$ handles a disproportionately large share of modality $m$ tokens, suggesting specialization. Values \< 1 indicate below-average processing of that modality.
>
> We analyzed routing patterns across four medical imaging modalities (CT, MRI, Pathology, X-Ray) and six clinical question types: MODALITY (imaging technique/plane), ANATOMY (anatomical structure), ABNORMALITY_PRESENCE (presence/absence), DISEASE_IDENTIFICATION (disease type), LOCALIZATION (spatial location), and CLINICAL_REASONING (function/symptom/comparison/inference). Analyses were performed on SLAKE and PATH-VQA test sets, with question types annotated using rule-based methods and manually verified.
>
> **Key Clinical Findings:**
>
> **1. Text Processing Specialization:** At each layer, one or a few small expert groups (often singletons) handle the majority of text tokens across all question types. This reflects two characteristics: (a) text tokens are sparse relative to image tokens in multimodal inputs, requiring fewer specialized experts; (b) expert groups show limited sensitivity to semantic differences among question types, suggesting more homogeneous text processing compared to visual processing.
>
> **2. Imaging Modality Specialization:** Token distribution for medical images is markedly more balanced and dispersed across expert groups. Crucially, groups showing clear modality preferences typically have size ≥ 2, indicating emergent collaborative patterns where multiple experts coordinate around a dominant expert to process specific imaging modalities. This mirrors clinical practice, where radiologists integrate multiple knowledge sources when interpreting different imaging types.
>
> **3. Layer-wise Clinical Pattern Evolution:** The model develops increasingly sophisticated modality-specific pathways through its depth. In final layers, we observe pronounced functional differentiation: some groups (e.g., group_27) strongly prefer Pathology images, while others (e.g., group_59) specialize in radiology modalities (CT, MRI, X-Ray). This hierarchical specialization suggests the model learns to route different imaging types through distinct processing pathways—analogous to how different medical specialists develop domain-specific expertise.

---

> > ### Author Response · Authors · 2025-11-24
> >
> > **Conclusion:**
> >
> > These analyses demonstrate that expert groups indeed develop functional specialization. Some groups preferentially handle text tokens, while others specialize in specific medical image modalities. The emergence of these specialized groups is not manually designed but naturally learned through the adaptive grouping mechanism, providing evidence that AEG successfully captures meaningful collaboration patterns.
> >
> > **Reference:**
> >
> > [1] Krajewski J, Ludziejewski J, Adamczewski K, et al. Scaling laws for fine-grained mixture of experts. arXiv preprint arXiv:2402.07871, 2024.
> >
> > ---
> >
> > ### W2: Experiments are narrowly focused on VQA
> >
> > Thank you for this valuable suggestion. We acknowledge that our initial evaluation focused primarily on medical VQA tasks. To address this concern and demonstrate broader applicability, we have extended our experiments in two complementary directions: (1) additional clinical tasks within the medical domain, and (2) tasks in non-medical domains.
> >
> > **Extension 1: Medical Report Generation (MRG)**
> >
> > To evaluate our approach on a broader range of clinical tasks, we conducted experiments on Medical Report Generation, which represents a significantly more complex form of clinical reasoning than VQA. Unlike VQA which requires selecting or generating short answers, MRG demands comprehensive analysis of medical images to produce structured clinical findings and impressions, a task requiring deeper semantic understanding and longer-form reasoning.
> >
> > **Experimental Setup:** We trained our models on a subset of MIMIC-CXR and evaluated on both MIMIC-CXR (in-distribution/ID) and IU-Xray (out-of-distribution/OOD). We adopted RadEntityMatchExact (RaTE) as our primary metric, as it more accurately reflects the semantic correctness of clinical entities compared to traditional language generation metrics like BLEU or ROUGE.
> >
> > **Extension 2: Non-Medical Domain (Food VQA)**
> >
> > To assess whether our findings about fine-grained MoE and AEG generalize beyond medical applications, we selected the food domain as a complementary testbed. Inspired by [1], we used a subset of the Food101 dataset, formulated as a VQA task for food category classification. The dataset was split into three subsets: train, test (ID), and test (OOD), where the OOD set contains different food categories from those in training and ID test sets, allowing us to evaluate generalization in a domain with different visual and semantic characteristics from medical imaging.
> >
> >
> > > Table 1: Cross-Task and Cross-Domain Evaluation
> > | **Models**           | **G**  | **MRG**      |         |  | **Food VQA** |           |
> > | ---------------- | -- | -------- | ------- |--  | -------- | --------- |
> > |                  |    | ID       | OOD     |  | ID       | OOD       |
> > |                  |    | MIMC-CXR | IU-Xray |  | Food101  | Food101\* |
> > |                  |    | RaTE     | RaTE    |  | Acc.     | Acc.      |
> > | FGMoE-S6k2       | 2  | 49.88    | 54.84   |  | 93.70    | 38.40     |
> > | FGMoE-S12k4      | 4  | **52.18**    | 53.95   |  | 94.40    | 31.70     |
> > | FGMoE-S24k8      | 8  | 52.05    | **56.92**   |  | 94.50    | 36.10     |
> > | FGMoE-S48k16     | 16 | 49.11    | 52.71   |  | 94.60    | 38.10     |
> > | FGMoE-S96k32     | 32 | 49.02    | 52.50   |  | **94.70**    | 39.60     |
> > | FGMoE-S192k64    | 64 | 49.78    | 53.01   |  | 94.40    | **41.00**     |
> > |                  |    |          |         |  |          |           |
> > | FGMoE-Ada-S12k4  | 4  | 51.74    | 53.26   |  | 94.70    | 24.70     |
> > | FGMoE-Ada-G12k2  | 4  | 51.91    | 53.74   |  | 94.70    | 26.60     |
> > | FGMoE-Ada-S96k32 | 32 | 51.65    | 53.83   |  | **94.80**    | 28.30     |
> > | FGMoE-Ada-G96k2  | 32 | **52.91**    | **54.73**   |  | 94.60    | **34.10**     |

---

> > > ### Author Response · Authors · 2025-11-24
> > >
> > > **Analysis: Task-Dependent Manifestation of Granularity Trade-offs**
> > >
> > > Our cross-task and cross-domain experiments reveal that while the core phenomenon (granularity affecting the balance between fitting and generalization) generalizes across tasks, its manifestation varies with task complexity:
> > >
> > > **For VQA Tasks (Medical and Food):** The granularity trade-off is relatively clear and consistent. In Medical VQA (our primary experiments), we observe that higher granularity improves OOD generalization but slightly reduces ID performance. In Food VQA, we observe improved OOD generalization as granularity increases (from 38.40% at G=2 to 41.00% at G=64), though ID performance shows smaller fluctuations rather than consistent decline. We attribute this difference to the lower semantic complexity of food classification compared to medical reasoning—the simpler task structure makes the ID fitting less sensitive to granularity changes.
> > >
> > > **For Medical Report Generation:** The pattern is more nuanced due to the task's complexity. When examining models fine-tuned directly on MRG (FGMoE-S series), we do not observe the same clear monotonic trade-off as granularity increases from G=2 to G=64. However, when examining our pre-trained models (FGMoE-Ada-S12k4 vs. FGMoE-Ada-S96k32), we observe the characteristic trend: higher granularity (G=32) yields better OOD performance (53.83 vs. 53.26) but slightly lower ID performance (51.65 vs. 51.74). This suggests that for highly complex generation tasks, the granularity trade-off emerges more clearly in well-pre-trained models, as sufficient pre-training provides the foundation needed to leverage fine-grained expert specialization effectively.
> > >
> > > **Effectiveness of AEG Across Tasks and Domains:**
> > >
> > > Critically, our AEG method demonstrates consistent effectiveness across all evaluated tasks and domains:
> > >
> > > - **Medical Report Generation:** FGMoE-Ada-G96k2 achieves the best balance, outperforming both lower-granularity models and same-granularity baselines on both ID (52.91 vs. 51.65 for S96k32) and OOD (54.73 vs. 53.83) settings.
> > >
> > > - **Food VQA:** FGMoE-Ada-G96k2 significantly improves OOD generalization (34.10%) compared to FGMoE-Ada-S96k32 (28.30%), while maintaining comparable ID performance (94.60% vs. 94.80%).
> > >
> > > - **Consistent Pattern:** Across both domains, comparing G96k2 (our AEG variant) with S96k32 (same granularity without grouping) confirms our hypothesis: adaptive grouping successfully mitigates the redundancy introduced by fine-grained expert collaboration while preserving the benefits of increased granularity for generalization.
> > >
> > > **Summary:**
> > >
> > > 1. **Granularity Trade-off Generalizability:** The fundamental phenomenon, that finer granularity in MoE architectures creates a trade-off between fitting capacity and generalization, extends beyond medical VQA to other tasks and domains. However, its manifestation varies with task complexity: clearer in VQA tasks, more subtle in complex generation tasks without sufficient pre-training.
> > >
> > > 2. **AEG Effectiveness:** Our Adaptive Expert Grouping method consistently improves the granularity-generalization trade-off across medical VQA, medical report generation, and non-medical classification tasks, demonstrating its broad applicability as a general solution for fine-grained MoE architectures.
> > >
> > > These results strengthen our contribution by demonstrating that our core insights and proposed method are not limited to medical VQA but extend to a broader range of clinical tasks and non-medical domains, addressing your concern about narrow evaluation scope.
> > >
> > > **Reference:**
> > > [1] Cheng D, Huang S, Zhu Z, et al. On Domain-Adaptive Post-Training for Multimodal Large Language Models. arXiv:2411.19930, 2024.
> > >
> > >
> > >
> > > ---
> > >
> > > ### W3: AEG lacks comparison with other efficiency approaches
> > >
> > > Thank you for raising this important point. We appreciate the opportunity to clarify the positioning of AEG relative to other efficiency approaches.
> > >
> > > While Table 3 does not directly include expert pruning or hierarchical routing as baselines, we did provide comparisons with reducing Top-k, which represents the most straightforward approach to reducing routing overhead. We did not initially include expert pruning and hierarchical routing as direct comparisons because AEG fundamentally differs from these methods in both design philosophy and operational objectives. However, to address your concern, we have now conducted additional experiments comparing AEG with expert pruning.

---

> > > > ### Author Response · Authors · 2025-11-24
> > > >
> > > > **1. Comparison with Expert Pruning:**
> > > >
> > > > **Fundamental Difference in Philosophy:**
> > > > Traditional expert pruning methods reduce routing overhead by permanently removing or merging "redundant" experts based on similarity metrics or importance scores. However, in the fine-grained MoE settings we investigate, strong co-activation patterns do not necessarily indicate simple redundancy. Instead, they often represent necessary collaborative patterns where multiple small-capacity experts jointly encode complex knowledge structures. Naively pruning such experts risks disrupting these collaborative representations.
> > > >
> > > > In contrast, AEG does not remove experts. Instead, it identifies and aggregates frequently co-activated experts into groups, allowing the router to make decisions at the group level—effectively forming "larger-capacity virtual experts" while preserving the underlying fine-grained expert parameters. This approach maintains the representational capacity of fine-grained experts while significantly reducing the router's burden of distinguishing among highly similar experts. Conceptually, AEG transforms the routing objective from "selecting individual experts from a large pool" to "selecting expert groups that naturally emerge from co-activation patterns."
> > > >
> > > > **Empirical Comparison:**
> > > > To empirically validate this distinction, we implemented a representative expert pruning baseline. Specifically, we applied k-means clustering to group experts based on their weight parameters, setting `n_clusters` to 75% of the total number of routed experts (72 clusters for a model with 96 experts). We then merged experts within each cluster by averaging their weight parameters and corresponding router embeddings. We applied this pruning procedure to FGMoE-Ada-S96k32, resulting in a model with 72 fine-grained experts.
> > > >
> > > > The results are presented in Table 2 below:
> > > >
> > > > > **Table 2: Comparison of Expert Pruning vs. AEG**
> > > > >
> > > > | Methods       | SLAKE |       | PATH-VQA |       | OMNI  | Avg-ID | Avg-All |
> > > > |---------------|-------|-------|----------|-------|-------|--------|---------|
> > > > | Baseline (S96k32) | 82.89 | 86.30 | 39.65    | 92.45 | 64.05 | **75.32**  | 69.69   |
> > > > | Top-2         | 84.35 | 85.10 | 38.20    | 92.33 | 61.80 | 74.99  | 68.40   |
> > > > | Pruned (72 experts) | 82.72 | 85.34 | 39.59    | 92.48 | 63.40 | 73.96  | 68.15   |
> > > > | AEG* (72 groups max) | 83.10 | 84.86 | 39.62    | 92.39 | **65.10** | 74.99  | **70.05**   |
> > > > >
> > > > > \* To align with the pruning baseline, AEG's maximum group size was set to 72.
> > > >
> > > > **Key Observations:**
> > > > - **Performance Preservation:** While expert pruning causes noticeable performance degradation (Avg-All drops from 69.69 to 68.15), AEG actually improves overall performance (70.05), demonstrating that adaptive grouping better preserves model capacity than hard pruning.
> > > >
> > > > - **OOD Generalization:** Most notably, AEG achieves superior OOD performance (OMNI: 65.10) compared to both the baseline (64.05) and the pruned model (63.40). This advantage on unseen data is something traditional pruning methods cannot achieve, as they permanently discard information. AEG, by preserving all expert parameters while reorganizing routing decisions, maintains generalization capability.
> > > >
> > > > **2. Distinction from Hierarchical Routing:**
> > > >
> > > > Hierarchical routing methods typically reduce routing complexity through multi-level or tree-structured decision processes. However, these approaches often rely on:
> > > > - **Manual design:** The hierarchical structure is typically predefined based on human intuition or domain knowledge.
> > > > - **Strong prior assumptions:** The hierarchy assumes certain clustering or organizational principles that may not align with actual expert collaboration patterns learned during training.
> > > > - **Static structures:** Once defined, the hierarchy remains fixed and cannot adapt to sample-specific routing needs.
> > > >
> > > > **AEG's Data-Driven Approach:** In contrast, AEG takes a fundamentally data-driven approach. Rather than imposing a predefined hierarchical structure, it:
> > > > 1. **Learns from observed behavior:** Group structures emerge from actual expert co-activation patterns observed during training, ensuring alignment with the model's learned collaborative dynamics.
> > > > 2. **Adapts dynamically:** Through adaptive group sizes (including empty groups that activate no experts), AEG enables the model to dynamically adjust the effective number of activated parameters per token based on input complexity. This token-adaptive capacity allocation is not directly achievable with traditional hierarchical routing or pruning methods, which typically fix the number of activated experts in advance.
> > > > 3. **Preserves flexibility:** The grouping strategy can be updated or refined as training progresses, allowing the routing structure to evolve with the model's learning.

---

> > > > > ### Author Response · Authors · 2025-11-24
> > > > >
> > > > > **Summary:**
> > > > >
> > > > > We believe these fundamental differences, preserving vs. removing capacity (pruning), learned vs. imposed structure (hierarchical routing), justify treating AEG as a distinct methodological contribution rather than a direct variant of existing efficiency techniques. The empirical comparison with expert pruning further demonstrates that AEG's approach of reorganizing routing decisions while preserving model capacity offers unique advantages, particularly for OOD generalization.
> > > > >
> > > > > ---
> > > > >
> > > > > ## Response to Questions
> > > > >
> > > > > ### Q1: In Figure 2, where are matrices $A$, $E_g$ and $E_e$ located?
> > > > >
> > > > > Thank you for seeking clarification. In our implementation, each MoE layer uses `nn.Embedding` to define $E_g$ and $E_e$. Specifically, each MoE layer consists of an expert set, a router, and the embeddings $E_g$ and $E_e$. The matrix $A$ is computed from $E_g$ and $E_e$ to implement the group-based routing mechanism. We will clarify this architectural detail more explicitly in the revised manuscript.
> > > > >
> > > > > ---
> > > > >
> > > > > ### Q2: Why do tokens get routed to the same expert pairs as granularity increases?
> > > > >
> > > > > This is an excellent question that touches on one of our key findings. There are two main reasons for this phenomenon:
> > > > >
> > > > > **First, activation budget scaling:** As granularity increases, not only does the total number of experts grow, but the number of activated experts also increases proportionally to maintain consistent activated parameter budgets. This creates more opportunities for expert co-activation.
> > > > >
> > > > > **Second, emergent collaborative patterns:** As granularity increases, we observe stronger co-activation phenomena among certain experts, as shown in Figure 7(b). We believe this pattern emerges because as granularity increases and individual expert size decreases, single experts become insufficient to independently construct complex semantic or knowledge representations. Consequently, the model tends to combine multiple experts to form higher-level semantic features or knowledge representations.
> > > > >
> > > > > This observation motivated our design of AEG: rather than forcing the router to make fine-grained distinctions among experts that naturally work together, we explicitly model these collaborative groups and route at the group level.
> > > > >
> > > > > ---
> > > > >
> > > > > We hope these responses adequately address your concerns. Thank you again for your constructive feedback.
> > > > >
> > > > > Best regards,
> > > > > The Authors

---

> > > > > > ### Author Response · Authors · 2025-11-26
> > > > > >
> > > > > > Dear Reviewer 7da9,
> > > > > >
> > > > > > We hope this message finds you well. As the discussion period is progressing, we would like to kindly follow up regarding our response to your review. If there are any additional questions, concerns, or points you would like us to clarify, please let us know. Your insights are invaluable to us, and we are happy to clarify any remaining points that could help with your evaluation.
> > > > > >
> > > > > > We sincerely appreciate the time and effort you have dedicated to evaluating our paper and would be very grateful for any further comments you may have during the discussion phase.
> > > > > >
> > > > > > Thank you again for your thoughtful feedback and for reviewing our submission.
> > > > > >
> > > > > > Best regards,
> > > > > >
> > > > > > The Authors

---

> > > > > > > ### Author Response · Authors · 2025-11-28
> > > > > > >
> > > > > > > Dear Reviewer 7da9,
> > > > > > >
> > > > > > > We hope this message finds you well. As the discussion period is approaching its end, we want to gently follow up on our earlier response to your review. If there is anything we can clarify or expand upon to assist with your evaluation, we would be very happy to provide additional details.
> > > > > > >
> > > > > > > Thank you again for the time and effort you have devoted to evaluating our work. We would be grateful for any further feedback you may have.
> > > > > > >
> > > > > > > Best regards,
> > > > > > >
> > > > > > > The Authors

---

> ### Author Response · Authors · 2025-11-30
> **Response Summary for Reviewer 7da9**
>
> Dear AC,
>
> To facilitate your assessment, we briefly summarize how we addressed Reviewer 7da9’s main concerns and questions.
>
> 1. **Positioning within medical multimodal learning and motivation (W1)**
>    We clarified that our primary motivation is to bring Fine‑Grained MoE (FGMoE), which has shown strong performance in general domains, into the medical multimodal setting, where such architectures had not yet been explored. This allows us to better tackle challenges specific to the medical domain, such as heterogeneous imaging modalities and high knowledge density. We first perform a systematic study of granularity in medical VQA to understand how expert granularity affects performance. Based on the observed increase in expert co‑occurrence at higher granularity, we then propose Adaptive Expert Grouping (AEG), which is inspired by findings in the medical domain but is not limited to it, as demonstrated by our additional experiments on non‑medical data.
>
> 2. **Clinical interpretation of routing and modality‑wise analysis (W1)**
>    To provide the clinical insights you requested, we added a detailed routing analysis in the appendix (Figures 19–20). We compute a relative preference score that measures each expert group’s affinity for specific modalities (CT, MRI, Pathology, X‑ray) and question types (e.g., ANATOMY, DISEASE_IDENTIFICATION, LOCALIZATION). We find that (i) text tokens are predominantly handled by a few small groups across question types, while (ii) some larger groups show clear preferences for specific image modalities (e.g., pathology‑focused vs. radiology‑focused groups). These patterns indicate that AEG learns meaningful, functionally specialized expert groups rather than arbitrary clusters.
>
> 3. **Broader tasks beyond medical VQA (W2)**
>    To address your concern about narrow task coverage, we added additional experiments on both medical and non‑medical tasks:
>    - **Food VQA:** As granularity increases, OOD accuracy improves (e.g., from 38.40 to 41.00), while ID accuracy only fluctuates slightly around 94.4–94.7.
>    - **MRG:** For pre‑trained models on medical report generation (MIMIC‑CXR → IU‑Xray), higher granularity (S96k32 vs. S12k4) yields better OOD RaTE (53.83 vs. 53.26) at a small ID cost (51.65 vs. 51.74).
>
>    These results indicate that the “higher granularity → better OOD with a minor ID trade‑off” pattern is robust for VQA‑style tasks and remains observable for more complex generation tasks when models are sufficiently pre‑trained. We also confirm that similar **expert co‑occurrence** patterns appear in the non‑medical food domain, suggesting they are not specific to medical data.
>
> 4. **Comparison with other efficiency approaches (W3)**
>    We clarified how AEG differs from standard efficiency methods such as expert pruning and hierarchical routing, and we added a pruning baseline for direct comparison.
>    - **Versus expert pruning:** Pruning permanently removes or merges experts to save computation while trying to minimize performance loss. In contrast, AEG preserves all fine‑grained experts and reduces routing cost by grouping frequently co‑activated experts into “virtual experts.” Empirically, pruning 96 experts down to 72 degrades overall performance (Avg‑All 68.15 vs. 69.69), whereas AEG with a maximum of 72 groups improves it (Avg‑All 70.05) and yields better OOD performance (OMNI 65.10 vs. 64.05 for the baseline).
>    - **Versus hierarchical routing:** Hierarchical routing typically relies on manually designed or strongly prior‑driven structures that remain static during training. AEG instead learns grouping structures directly from observed co‑activation patterns and updates them jointly with the task objective.
>
>    This data‑driven, capacity‑preserving design distinguishes AEG from both pruning and fixed hierarchical schemes.
>
> 5. **Location of matrices $A$, $E_g$, and $E_e$ (Q1)**
>    In each MoE layer, we implement $E_g$, and $E_e$ as `nn.Embedding` modules that are part of the layer alongside the experts and the router. The group‑level routing matrix $A$ is computed from $E_g$ and $E_e$ within the same layer. We have clarified these implementation details around Figure 2 in the revised paper.
>
> 6. **Reason for persistent expert co‑occurrence as granularity increases (Q2)**
>    Our understanding of the persistent co‑occurrence is that, as granularity grows, each individual expert has smaller capacity, making single experts less able to encode complex clinical concepts in isolation. The model therefore learns to combine multiple fine‑grained experts that frequently fire together, forming higher‑level representations via collaboration.
>
> Best regards,
> The Authors

---

### Official Review · Reviewer_Yx3H · 2025-10-23

**Soundness:** 3
**Presentation:** 3
**Contribution:** 2
**Rating:** 6
**Confidence:** 3

**Summary:**

This paper systematically studies fine-grained MoE in the field of medical multimodal learning, analyzing how expert granularity influences both in-domain and out-of-distribution performance. The authors empirically show that finer granularity substantially improves OOD generalization and robustness, while slightly reduces ID fitting and induces significant functional redundancy among experts, increasing routing overhead. To address this, they propose a grouping strategy called Adaptive Expert Grouping (AEG), which shifts routing decisions from individual experts to learned expert groups, to reduce computational cost and routing sparsity while preserving the generalization benefits of fine-grained specialization. Overall, the paper’s contributions include both novel empirical findings and a targeted methodological innovation.

**Strengths:**

- The observations provided by this paper on fine-grained MoE are interesting and original, which I believe could bring inspirations to future researches.
- The method proposed in this paper is explanable and well presented, and is effective according to the given experiment results.
- The experiments of this paper are comprehensive with results on multiple models and datasets.

**Weaknesses:**

- The domain focus of this paper is unclear. While the paper claims to focus on **medical multimodal learning**, the methodological core and analyses remain largely domain-agnostic. Most components are general techniques applicable to any multimodal or language model. The work does not convincingly justify why the medical setting is essential for either the problem formulation or the proposed method, beyond serving as an evaluation domain.
- The depth in interpreting the findings is limited. The empirical results are well presented, but the explanatory analysis lacks depth. For example, the authors note that finer expert granularity improves OOD generalization but degrades in-domain fitting, yet they provide little mechanistic insight into *why* this occurs. Similarly, the relationship between expert co-occurrence, redundancy, and degraded in-domain performance is asserted rather than rigorously examined (e.g., through probing, correlation, or ablation analyses). This limits the paper’s scientific contribution beyond empirical observation.
- The novelty of the proposed method is limited. Although the proposed AEG mechanism is a thoughtful extension, the idea of grouping or clustering experts based on similarity has been explored in several prior MoE variants (e.g. [[1]](https://arxiv.org/abs/2207.09094), [[2]](https://arxiv.org/abs/2504.09265)). The paper’s contribution lies more in recontextualizing this idea for fine-grained setups rather than introducing a substantially new principle. As such, the methodological novelty appears incremental relative to the growing body of literature on adaptive routing and expert aggregation.

**Questions:**

- Could the authors provide stronger evidence that the findings (e.g. optimal granularity, expert grouping behavior) are intrinsically tied to **medical multimodal tasks** rather than arising from general multimodal or MoE settings? How well do the proposed methods and conclusions transfer to non‐medical multimodal tasks (e.g. general vision+language or other non‐clinical domains)?
- There is a claim that with the same total activated parameter budget / number of active parameters, using finer granularity of experts leads to worse in-domain fitting. Could the authors explain what mechanism leads to that degradation? For instance, is it because with finer granularity, each expert receives fewer data points so they overfit less or under-specialize?

---

> ### Author Response · Authors · 2025-11-24
>
> Dear Reviewer Yx3H,
>
> Thank you for your thoughtful and constructive feedback. We appreciate your careful consideration of our work's scope, depth, and novelty. Below, we address each of your concerns in detail.
>
> ---
>
> ## Response to Weaknesses
>
> ### W1: The domain focus of this paper is unclear
>
> Thank you for raising this concern. Our focus on the medical multimodal domain stems from the intention to extend the proven effectiveness of fine‑grained MoE architectures to medical applications. This motivation naturally led us to the central idea of this work, as detailed below:
>
> **First, research motivation:** We observed that while the effectiveness of fine-grained MoE architectures has been widely validated in general domains, it has not been systematically explored in medical multimodal settings. Medical data exhibits characteristics of multi-modality, multi-task complexity, and high knowledge density, typically requiring comprehensive reasoning across modalities and knowledge hierarchies. We believe the two key advantages of fine-grained MoE, (1) more flexible expert combinations and (2) more fine-grained knowledge representation, naturally align with the demands of the medical domain.
>
> **Second, empirical validation gap:** While existing work such as [1] has shown that finer expert granularity generally improves performance in general corpora, the "degree of granularity" in vertical domains like medicine still lacks empirical support. Therefore, we systematically analyzed the impact of granularity changes on MoE model performance through medical multimodal tasks.
>
> **Third, novel observation:** During this process, we first observed that as granularity increases, certain experts exhibit more pronounced co-activation phenomena. Inspired by this, we proposed AEG (Adaptive Expert Grouping), which explicitly transforms implicit collaborative relationships among experts into learnable structures. While this method originated from medical multimodal research, its core mechanism is inherently domain-agnostic.
>
> Due to space constraints and our research focus on complex, knowledge-intensive scenarios, we primarily conducted experiments and analyses in the medical multimodal domain to validate the method's effectiveness. Given that multiple reviewers have expressed interest in the method's generalizability across domains, we have supplemented our work with experimental results on non-medical tasks to demonstrate AEG's broader applicability. The detailed results can be found in our response to your first question below.
>
> **Reference:**
> [1] Krajewski J, Ludziejewski J, Adamczewski K, et al. Scaling laws for fine-grained mixture of experts[J]. arXiv preprint arXiv:2402.07871, 2024.
>
> ---
>
> ### W2: The depth in interpreting the findings is limited
>
> Thank you for this valuable suggestion. We agree that the mechanistic analysis of certain phenomena could be more thorough, and we provide further clarification of our understanding and intuitive explanations here.
>
> **Regarding why finer expert granularity improves OOD generalization but slightly weakens in-domain fitting:**
>
> Our working hypothesis is that as granularity increases, the space of feasible expert combinations grows exponentially. For example, when granularity is coarse (e.g., G=2), the number of expert combinations available to the router is limited, whereas in high-granularity settings (e.g., G=32), the router faces an exponentially larger combination space. This has two effects:
>
> 1. **Increased routing learning difficulty:** A larger combination space makes it harder for the router to sufficiently "memorize" or fit each combination.
> 2. **Fewer data points per combination:** The number of training samples corresponding to each combination decreases, thereby reducing the model's ability to learn specific in-domain patterns.
>
> These two factors jointly lead to slight degradation in in-domain fitting. Simultaneously, the larger combination space and richer forms of expert collaboration make the model less dependent on a few fixed patterns when facing distribution shifts, which manifests experimentally as better OOD generalization and robustness. We will discuss this hypothesis and its limitations more explicitly in the paper.
>
> **Regarding the relationship between expert co-occurrence, redundancy, and performance:**
>
> Our intuitive understanding is:
>
> - Experts that are frequently co-activated tend to form collaborative patterns and are functionally highly related.
> - In traditional fine-grained MoE, the router is still forced to make fine-grained distinctions among these highly similar individual experts, introducing a degree of "differentiation redundancy": the router repeatedly makes subtle distinctions among nearly equivalent experts rather than making decisions among truly semantically complementary units.

---

> > ### Author Response · Authors · 2025-11-24
> >
> > Based on this observation, we propose AEG, which shifts the routing objective from "distinguishing individual experts" to "distinguishing expert groups": within groups, stable collaborative patterns are formed through co-activation; between groups, the emphasis is on expressing semantic or functional differences. This approach alleviates routing redundancy to some extent while strengthening structured collaboration.
> >
> > We acknowledge that more rigorous mechanistic analysis (e.g., through probing, correlation analysis, or ablation studies) would strengthen these claims, and we will consider incorporating such analyses in future work.
> >
> > ---
> >
> > ### W3: The novelty of the proposed method is limited
> >
> > Thank you for highlighting these relevant works. We carefully reviewed MoEC [1] and MoGE [2] and believe both studies validate the effectiveness and research value of introducing "expert grouping" in MoE from different perspectives.
> >
> > Specifically, MoEC [1] addresses the problem of insufficient sample diversity for experts in traditional MoE by pre-defining L adjacent experts as a cluster and using constraints during training to make routing probability distributions similar within clusters and different between clusters, thereby forming cluster-based routing structures.
> >
> > MoGE [2] introduces group-sparse regularization to encourage smoothness and consistency in local routing signals, enabling adjacent experts to spontaneously form "expert groups" during training. This structured sparsity enhances routing stability globally and indirectly promotes expert diversification and specialization. Unlike MoEC, MoGE does not require pre-defined expert groups but achieves adaptive grouping through regularization.
> >
> > **Our work differs from these two approaches in four key aspects:**
> >
> > **1. Formation mechanism:** MoEC and MoGE primarily encourage local aggregation of experts through constraints or regularization during training, whereas our AEG is based on genuinely observed expert co-activation patterns in fine-grained MoE architectures. Rather than artificially imposing grouping constraints, we explicitly model the collaborative relationships that the model spontaneously exhibits in fine-grained settings.
> >
> > **2. Activation granularity:** MoEC and MoGE still use expert-level routing, while AEG adopts group-level routing. Once experts form stable collaborative groups, continuing to make activation selections at the expert level may lead to redundant differentiation or truncation of partial collaborative units. AEG directly activates complete expert groups, reducing unnecessary expert distinctions while maintaining functional completeness.
> >
> > **3. Objective differences:** MoEC and MoGE focus on enhancing routing stability, while AEG's core motivation is to explicitly model collaborative patterns among experts, transforming implicit behaviors into learnable structures to reduce redundancy while improving expert collaboration and interpretability. Additionally, we believe that expert ensembles composed of groups of different sizes are intuitive, as different knowledge/semantic expressions require different expert sizes. This points to a potential new paradigm for fine-grained MoE: rather than manually designing expert granularity, we can start from very small expert size and let the model adaptively form groups with larger sizes.
> >
> > **4. Empty expert groups:** We allow the existence of empty expert groups, further enhancing activation sparsity and flexibility in expert combinations.
> >
> > In summary, while we recognize the exploratory value of MoEC and MoGE in expert grouping, and believe these works provide inspiration for improving our group loss design, AEG demonstrates novel perspectives and mechanisms in terms of formation motivation, activation granularity, and objective orientation.
> >
> > ---

---

> > > ### Author Response · Authors · 2025-11-24
> > >
> > > ## Response to Questions
> > >
> > > ### Q1: Evidence that findings are intrinsically tied to medical multimodal tasks? How well do transfer to non‐medical multimodal tasks？
> > >
> > > Thank you for this important question. While our research is grounded in medical multimodal tasks, our findings and proposed methods are **not limited to the medical domain**. Through extended experiments on non-medical tasks, we demonstrate that our core observations and the AEG approach generalize well across different multimodal settings.
> > >
> > > **Core Findings That Generalize Across Domains:**
> > >
> > > We identify three key findings that hold consistently across both medical and non-medical domains:
> > >
> > > 1. **Granularity-Generalization Trade-off:** As expert granularity increases, model generalization improves across domains, though this comes with varying impacts on in-distribution performance depending on task complexity.
> > >
> > > 2. **Expert Co-occurrence Patterns:** Fine-grained expert configurations lead to significantly increased expert co-activation across domains, indicating this is a fundamental characteristic of MoE architectures rather than a medical-specific phenomenon.
> > >
> > > 3. **AEG Effectiveness:** Our adaptive expert grouping approach successfully mitigates redundancy while preserving generalization benefits across different task types.
> > >
> > > **Cross-Domain Validation on Food VQA:**
> > >
> > > Inspired by [1], we selected the food domain as a non-medical testbed to assess cross-domain generalization. We used a subset of the Food101 dataset, formulated as a VQA task for food category classification. The dataset was split into three subsets: train, test (ID), and test (OOD), where the OOD set contains different food categories from those in the training and ID test sets. Results are shown in Table 1.
> > >
> > > **Finding 1: Granularity-Generalization Trade-off Generalizes Across Domains**
> > >
> > > The Food VQA results clearly demonstrate the granularity-generalization trade-off observed in medical tasks. As granularity increases from G=2 to G=64, OOD performance improves consistently (38.40% → 41.00%), confirming that finer-grained experts enhance generalization capability across domains. However, the impact on ID performance differs between domains: while medical VQA shows slight performance degradation with increasing granularity, Food VQA exhibits stable or slightly improved ID performance (93.70% → 94.70%). We attribute this difference to task complexity: food classification is primarily a perceptual recognition task with lower semantic complexity, whereas medical diagnostic reasoning involves multi-level clinical inference and knowledge integration. The stability of ID performance in simpler tasks suggests that the trade-off profile adapts to task characteristics, but the core pattern of improved generalization with finer granularity holds universally.
> > >
> > > **Finding 2: Expert Co-occurrence Patterns Are Domain-Agnostic**
> > >
> > > We analyzed expert co-occurrence patterns in the food domain as visualized in **Figure 21 in the appendix**. The analysis demonstrates that both medical and non-medical domains exhibit remarkably similar trends: as expert granularity increases, expert co-occurrence becomes more pronounced. This consistency provides strong evidence that the redundancy patterns induced by fine-grained experts are a general phenomenon in MoE architectures rather than being specific to medical data. The universal nature of this pattern validates our motivation for developing AEG as a general solution to fine-grained MoE redundancy.
> > >
> > > **Finding 3: AEG Effectiveness Transfers Across Domains**
> > >
> > > The Food VQA results validate AEG's effectiveness beyond medical tasks. Comparing FGMoE-S96k32 (standard fine-grained MoE, 94.80% ID, 28.30% OOD) with FGMoE-Ada-G96k2 (our AEG variant, 94.60% ID, 34.10% OOD), we observe that AEG achieves substantially better generalization (+5.8 percentage points on OOD) with minimal impact on ID performance (-0.2 percentage points). This improvement is particularly notable given that the baseline S96k32 already benefits from fine granularity. Furthermore, consistent with our hypothesis, coarse-grained configurations show limited benefit from AEG: FGMoE-Ada-G12k2 achieves 26.60% OOD performance compared to 24.70% for FGMoE-Ada-S12k4, a smaller improvement than at G=32. This aligns with our observation that finer granularity induces more pronounced expert co-activation patterns, making AEG more applicable and beneficial.

---

> > > > ### Author Response · Authors · 2025-11-24
> > > >
> > > > **Conclusion:**
> > > >
> > > > Our findings suggest that the benefits of fine-grained experts (improved generalization) and their challenges (increased co-activation and redundancy) are general characteristics of MoE architectures in multimodal settings. The AEG method provides a domain-agnostic solution that adapts to task-specific requirements while maintaining the advantages of fine-grained expert collaboration. The medical domain serves as a particularly demanding testbed due to its complexity, but both the observations and the proposed solution transfer effectively to other multimodal tasks.
> > > >
> > > > > **Table 1: Cross-Domain Validation on Food VQA**
> > > > | Models           | G  | Food VQA |           |
> > > > | ---------------- | -- | -------- | --------- |
> > > > |                  |    | ID       | OOD       |
> > > > |                  |    | Food101  | Food101\* |
> > > > |                  |    | Acc.     | Acc.      |
> > > > | FGMoE-S6k2       | 2  | 93.70    | 38.40     |
> > > > | FGMoE-S12k4      | 4  | 94.40    | 31.70     |
> > > > | FGMoE-S24k8      | 8  | 94.50    | 36.10     |
> > > > | FGMoE-S48k16     | 16 | 94.60    | 38.10     |
> > > > | FGMoE-S96k32     | 32 | **94.70**    | 39.60     |
> > > > | FGMoE-S192k64    | 64 | 94.40    | **41.00**     |
> > > > |                  |    |          |           |
> > > > | FGMoE-Ada-S12k4  | 4  | 94.70    | 24.70     |
> > > > | FGMoE-Ada-G12k2  | 4  | 94.70    | 26.60     |
> > > > | FGMoE-Ada-S96k32 | 32 | **94.80**    | 28.30     |
> > > > | FGMoE-Ada-G96k2  | 32 | 94.60    | **34.10**     |
> > > >
> > > >
> > > > **References:**
> > > > [1] Cheng D, Huang S, Zhu Z, et al. On Domain-Adaptive Post-Training for Multimodal Large Language Models. arXiv:2411.19930, 2024.
> > > >
> > > > ---
> > > >
> > > > ### Q2: What mechanism leads to degradation in in-domain fitting with finer granularity?
> > > >
> > > > As explained in our response to W2, our working hypothesis is that finer granularity leads to an exponentially larger expert combination space, which has two main effects:
> > > >
> > > > **1. Increased routing learning difficulty:** The router faces greater difficulty in fully "memorizing" or fitting each specific combination when the combination space is large.
> > > >
> > > > **2. Fewer samples per combination:** Each particular expert combination is exposed to fewer training samples, which reduces the model's ability to fit specific in-domain patterns.
> > > >
> > > > These factors jointly contribute to the slight degradation in in-domain fitting. The positive side is that this same mechanism, relying less on a few fixed patterns due to the richer combination space, leads to better OOD generalization and robustness.
> > > >
> > > > We will provide more detailed mechanistic analysis for this hypothesis in the revised manuscript.
> > > >
> > > > ---
> > > >
> > > > We hope these responses adequately address your concerns. We are committed to strengthening the paper's contributions in terms of mechanistic insights, broader empirical validation, and clearer positioning relative to existing work. Thank you again for your constructive feedback.
> > > >
> > > > Best regards,
> > > > The Authors

---

> > > > > ### Author Response · Authors · 2025-11-26
> > > > >
> > > > > Dear Reviewer Yx3H,
> > > > >
> > > > > We hope this message finds you well. As the discussion period is progressing, we would like to kindly follow up regarding our response to your review. If there are any additional questions, concerns, or points you would like us to clarify, please let us know. Your insights are invaluable to us, and we are happy to clarify any remaining points that could help with your evaluation.
> > > > >
> > > > > We sincerely appreciate the time and effort you have dedicated to evaluating our paper and would be very grateful for any further comments you may have during the discussion phase.
> > > > >
> > > > > Thank you again for your thoughtful feedback and for reviewing our submission.
> > > > >
> > > > > Best regards,
> > > > >
> > > > > The Authors

---

> > > > > > ### Comment · Reviewer_Yx3H · 2025-11-27
> > > > > >
> > > > > > Thanks for your response. These supplementary information address my concerns a lot, which makes this paper more solid. However, concerning the overall novelty of the paper, I would still keep my rating as 6. Thanks again for your active response.

---

> > > > > > > ### Author Response · Authors · 2025-11-27
> > > > > > >
> > > > > > > Dear Reviewer Yx3H,
> > > > > > >
> > > > > > > Thank you again for your follow-up comment and for acknowledging that our rebuttal has addressed your concerns a lot and made the paper more solid. We really appreciate the time and effort you have put into reading both the paper and our response.
> > > > > > >
> > > > > > > Regarding your remaining concern about novelty, we fully understand and respect your assessment. We agree that expert grouping has been explored in prior MoE work. At the same time, we would like to respectfully clarify one point: unlike MoEC and MoGE, which induce co-activation in coarse-grained MoE through constraints or regularization, our work starts from observed co-activation patterns that naturally emerge in fine-grained MoE and then further explicitly transforms these implicit collaborative relationships into learnable grouping structures. This enables adaptive granularity, preserving fine-grained modeling effectiveness while achieving coarse-grained computational efficiency, which we hope can suggest a promising direction for building more effective and efficient MoE models.
> > > > > > >
> > > > > > > Thank you once again for your constructive feedback throughout the review and discussion process. We would be happy to continue the discussion if you have any further questions.
> > > > > > >
> > > > > > > Best regards,
> > > > > > >
> > > > > > > The Authors

---

> ### Author Response · Authors · 2025-11-30
> **Response Summary for Reviewer Yx3H**
>
> Dear AC,
>
> To facilitate your assessment, we briefly summarize our interaction with Reviewer Yx3H and how we addressed the main points.
>
> 1. **Positioning within medical multimodal learning (W1, Q1)**
>    We clarified that our primary motivation is to bring Fine‑Grained MoE (FGMoE), which has shown strong performance in general domains, into the medical multimodal setting, where such architectures had not yet been explored. Medical data pose unique challenges (heterogeneous imaging modalities, high knowledge density, multi‑step clinical reasoning), for which flexible expert combinations and fine‑grained knowledge representations are particularly suitable. We also showed that our findings and AEG are not limited to medicine: beyond medical VQA, we added medical report generation (MIMIC‑CXR → IU‑Xray) and a non‑medical Food VQA benchmark, where we observe the same core patterns (granularity–generalization trade‑off and expert co‑occurrence), demonstrating that our conclusions and method transfer to non‑medical multimodal tasks.
>
> 2. **More explanation of the observed phenomena (W2, Q2)**
>    We provided a clearer, mechanism‑level hypothesis for why finer granularity slightly degrades ID fitting but improves OOD generalization. As granularity increases, the space of possible expert combinations grows exponentially: (i) the router faces a harder learning problem, and (ii) each specific combination sees fewer training examples. This reduces over‑specialized ID fitting but makes the model less reliant on a few fixed patterns, which empirically leads to stronger OOD robustness. We also explained how frequent expert co‑occurrence creates “differentiation redundancy”: in standard fine‑grained MoE the router must repeatedly distinguish among highly similar experts that already tend to work together. AEG addresses this by routing at the **group level**, explicitly modeling these collaborative patterns and reducing redundant distinctions.
>
> 3. **Methodological novelty vs. MoEC / MoGE (W3)**
>    In response to the concern about incremental novelty, we clarified how AEG differs from prior grouping‑based MoE variants such as MoEC and MoGE. Those methods encourage grouping via constraints or regularization (in coarser‑grained MoE) while still routing at the expert level. Our work instead starts from **observed** co‑activation patterns that naturally emerge in fine‑grained MoE, then explicitly turns these implicit collaborations into learnable expert groups and routes whole groups. This enables **adaptive granularity**: the model preserves fine‑grained modeling capacity internally while enjoying coarse‑grained computational efficiency and higher routing sparsity (including empty groups).
>
> 4. **Cross‑domain evidence (Q1)**
>    To strengthen the requested evidence on generality, we conducted cross‑domain experiments on Food VQA. As granularity increases, OOD accuracy consistently improves (38.40 → 41.00), while ID accuracy fluctuates slightly around 94.4–94.7. Moreover, our AEG variant (FGMoE‑Ada‑G96k2) significantly boosts OOD performance over the fine‑grained baseline at the same granularity (34.10 vs. 28.30 OOD) with almost unchanged ID accuracy. Together with our medical experiments, these results indicate that the “higher granularity → better OOD with a mild ID trade‑off” and the emergence of strong expert co‑occurrence are general MoE phenomena, and that AEG provides a domain‑agnostic way to mitigate redundancy while preserving generalization gains.
>
> **Reviewer’s reaction in the discussion phase**
>
> In the discussion, Reviewer Yx3H stated that our rebuttal “addressed the concerns a lot” and “made the paper more solid,”.
>
> Best regards,
> The Authors

---

### Official Review · Reviewer_R177 · 2025-10-25

**Soundness:** 3
**Presentation:** 3
**Contribution:** 3
**Rating:** 6
**Confidence:** 3

**Summary:**

Summary:

This paper proposes a Fine-Grained MoE for medical multimodal learning. An interesting point is that they have empirically demonstrated a clear trade-off: finer granularity gives better OOD generalization and robustness, but slightly worse ID fit and strong expert redundancy. To handle this redundancy, they propose Adaptive Expert Grouping (AEG), which learns to cluster similar experts and route at group level to cut routing cost while keeping gains. They also provide a fast proxy  for exploration. Experiments validate the effectiveness of the proposed method, and comprehensive analysis studies including the OOD-IID tradeoff, have been conducted.

**Strengths:**

Strength:

1. This paper systematically explored expert granularity and redundancy in leveraging MoE for multi-modal learning; and pointed out the resulting OOD-ID tradeoff. These are value findings that can guide MoE-based infra designs.
2. The model backbone is very updated, mostly are build on recent large VLMs or MoE backbones, and both main experiments and analysis studies are very comprehensive. Detailed results and experiemnt settings are provided in the appendix.
3. Strong results. The proposed method achieved SOTA or second highest points in most cases.
4. The proposed method is simple yet efficient, I think it's not complex to implement and can be easily plugged into many MoE infras.

**Weaknesses:**

Weakness & Questions:

1. My first question is that - why not pre-defined a group of shared experts, and several groups of sliced experts? Has your design been motivated by any evidence from training dynamics or other phenomena you observed?
2. For the $\mathcal{L}_{intra}$, is there any insight on doing the separation by considering the distances of experts' parameters? How about operating on the embeddings projected by the experts?
3. A potential solution to remove the redundancy while remain the high generalization ability, would be utilizing soft-MoE instead of sparse-MoE with gating networks. Because each expert in soft-MoE has been optimized join the training, and they are utilized in the inference time. Could you also discuss this type of works [1-4] in the related works?
4.  I noticed that you also used the histopathology images, how did you process them? If you use higher resolution pathology images, how do we change the design of shared experts and other sliced experts?
5. Is there any experiment results or expected phenomena about the scaling ability, to larger num_of_experts, parameters?
6. Also, does the choice of vision encoder (like pretrained in medical images or not) affect the experiment results?



[1] Puigcerver J, Ruiz C R, Mustafa B, et al. From Sparse to Soft Mixtures of Experts[C]//The Twelfth International Conference on Learning Representations.

[2] Wu C, Shuai Z, Tang Z, et al. Dynamic modeling of patients, modalities and tasks via multi-modal multi-task mixture of experts[C]//The Thirteenth International Conference on Learning Representations. 2025.

[3] Shen L, Chen G, Shao R, et al. Mome: Mixture of multimodal experts for generalist multimodal large language models[J]. Advances in neural information processing systems, 2024, 37: 42048-42070.

[4] Li Y, Jiang S, Hu B, et al. Uni-moe: Scaling unified multimodal llms with mixture of experts[J]. IEEE Transactions on Pattern Analysis and Machine Intelligence, 2025.

**Questions:**

see weakness

---

> ### Author Response · Authors · 2025-11-21
>
> Dear Reviewer R177,
>
> Thank you for your insightful questions and suggestions. We appreciate your careful consideration of our work. Below, we address each of your questions in detail.
>
> ---
>
> ## Response to Questions
>
> ### Q1: Why not pre-define a group of shared experts and several groups of sliced experts?
>
> Thank you for this question. First, we clarify that our model uses a single full expert as the shared expert, which is always activated during training and inference without going through the router. While theoretically we could introduce multiple shared experts, most MoE works typically adopt a single shared expert design.
>
> Regarding why we do not pre-define the routed expert grouping structure, there are two main considerations:
>
> **1. Scalability and Adaptivity:**
> Pre-defining fixed "sliced expert groups" but it is difficult to guarantee that the partition is optimal, and it also limits potential collaboration patterns among experts. In contrast, we aim for the model to adaptively learn which experts should form collaborative sets during training. This data-driven self-organization approach offers better scalability and generalizability across different scenarios.
>
> **2. Design Motivation and Empirical Evidence:**
> Our design is motivated by phenomena observed during experiments: as expert granularity increases, certain experts exhibit stronger co-activation patterns. We interpret these frequently co-activated experts as representing potential collaborative patterns. We attribute this phenomenon to the fact that as granularity increases and individual expert size decreases, single experts struggle to independently construct complex semantic or knowledge representations. Consequently, the model tends to combine multiple experts to form higher-level semantic features and knowledge.
>
> Based on this observation, we propose AEG to explicitly capture these implicit collaborative relationships through learnable expert grouping, transforming spontaneous expert collaboration patterns into learnable model structures.
>
> ---
>
> ### Q2: Is there any insight on designing $L_{intra}$ by considering distances of experts' parameters or operating on embeddings projected by experts?
>
> This is a valuable question regarding training objective design. AEG's design was initially inspired by co-activation behavior among experts. The most direct idea would be to construct an auxiliary separation loss $L_{sep}$ based on co-activation statistics. However, in practice, we found that once the grouping mechanism is introduced, the grouping itself changes the statistical meaning of co-activation, making it difficult to use as a viable optimization objective.
>
> Based on this insight, we made a transformation: if a group of experts is frequently co-activated during training, these experts should be functionally similar or complementary, and correspondingly, their representations in parameter space should be closer. This assumption directly led to our design of $L_{intra}$, constraining within-group expert parameters to be more similar encourages explicit aggregation of collaborative patterns. Additionally, we introduce $L_{inter}$ to promote differentiation between different groups.
>
> Regarding whether we could construct an auxiliary loss based on expert output embeddings, this is indeed a direction we seriously considered. However, we ultimately did not adopt this approach for two main reasons:
>
> **1. Input dependency:** Expert output embeddings are strongly correlated with specific input tokens. Different inputs cause significant fluctuations in these representations, which is not conducive to learning stable, input-agnostic grouping structures.
>
> **2. Sparse activation limitations:** Under sparse activation settings, we can only observe embeddings from the few experts selected by routing, making it difficult to constrain the overall structure of all experts, thereby weakening the learning of global grouping.
>
> Overall, designing $L_{intra}$ and $L_{inter}$ based on parameter distances is more stable in practice than approaches based on co-activation statistics or output embeddings, and is more conducive to learning clear, generalizable expert grouping structures.
>
> ---

---

> > ### Author Response · Authors · 2025-11-21
> >
> > ### Q3: Could you discuss soft-MoE works [1-4] in the related works as a solution to remove redundancy while maintaining generalization?
> >
> > Thank you for this suggestion. We will add discussion of soft-MoE methods in the related work section. Regarding your question, our understanding is as follows:
> >
> > First, [1] proposes soft-MoE, which improves training stability and achieves better overall performance through continuous differentiable routing. However, this method requires every expert to be used in forward computation. In contrast, we focus on Sparse-MoE architectures, whose core advantage lies in the sparse activation mechanism: maintaining a large total parameter count while activating only a subset of experts, thereby achieving performance improvements with controlled computational costs.
> >
> > Second, [2] applies the soft-MoE concept to medical imaging scenarios with task-specific adaptations and extensions, validating the broad applicability of soft-MoE.
> >
> > Additionally, [3] proposes MoME, which includes MoVE and MoLE components. MoVE adaptively weights and fuses outputs from different vision experts based on input language instructions, with a mechanism similar to soft-MoE's continuous gating. MoLE adopts sentence-level Sparse-MoE.
> >
> > Overall, soft-MoE research provides valuable insights into expert fusion approaches. Our work explores a complementary dimension by investigating new architectural designs that achieve expert specialization and collaboration while maintaining sparse activation.
> >
> > ---
> >
> > ### Q4: How did you process histopathology images? How would the design change for higher resolution pathology images?
> >
> > This is a question closely related to practical medical scenarios. For all medical image modalities, including pathology slides, we uniformly use Qwen2-VL's vision encoder for processing, which is based on a ViT trained encoder.
> >
> > Qwen2-VL natively supports variable-resolution image inputs. We scale images to fit within setted resolution range (12,544~451,584 pixels). Under this setting, even high-resolution pathology images can be directly encoded by the vision encoder without requiring any additional modifications to the shared expert or sliced/routed experts design. In other words, changes brought by higher resolutions are primarily handled by the front-end vision encoder.
> >
> > ---
> >
> > ### Q5: Are there any experimental results or expected phenomena about scaling ability to larger numbers of experts and parameters?
> >
> > Yes. This paper primarily investigates model performance changes as granularity increases (i.e., the number of routed experts increases from 6 to 192) while keeping total parameters and activated parameters constant.
> >
> > Our experiments show that as granularity increases, the model's generalization ability and robustness improve overall, though with slight decreases in fitting performance. Additionally, we observe that finer granularity leads to more pronounced co-activation phenomena among experts. Related results and analysis can be found in Figure 3 of the main text.
> >
> > ---
> >
> > ### Q6: Does the choice of vision encoder (e.g., pretrained on medical images or not) affect the experimental results?
> >
> > Yes, it does have an impact. If the vision encoder has been pretrained on medical images, it can typically extract fine-grained features more effectively in the medical domain, thereby further improving downstream performance. However, the base model we selected is Qwen2-VL, whose native vision encoder and LLM have undergone extensive modality alignment training. To preserve this good alignment and avoid disrupting pretrained knowledge, we retained the original vision encoder configuration. This is a valuable insight that highlights the importance of vision encoders in domain-specific adaptation.
> >
> > ---
> >
> > We hope these responses adequately address your questions. We are happy to provide further clarification or additional experiments as needed. Thank you again for your thoughtful feedback.
> >
> > Best regards,
> > The Authors

---

> > > ### Comment · Reviewer_R177 · 2025-11-25
> > >
> > > Thank you for the valuabe reply! I will increase my confidence score as all concerns regarding techniques have been addressed.

---

> > > > ### Author Response · Authors · 2025-11-25
> > > >
> > > > Dear Reviewer R177,
> > > >
> > > > Thank you very much for your prompt feedback and for increasing your confidence score. We are delighted to hear that our responses have successfully addressed your concerns regarding the technical details.
> > > >
> > > > We truly appreciate the time and effort you dedicated to reviewing our work. Your constructive suggestions have significantly strengthened the paper. As discussed in our rebuttal, we have updated the Related Work section to include a comparison with Soft-MoE methods. Additionally, we will incorporate clearer descriptions regarding the specific points you mentioned, such as the strategy for pre-defined groups, the design of the loss function, and the medical image processing workflow in the Method section.
> > > >
> > > > Thank you again for your strong support and for recognizing the value of our work.
> > > >
> > > >
> > > > Best regards,
> > > >
> > > > The Authors

---

> ### Author Response · Authors · 2025-11-30
> **Response Summary for Reviewer R177**
>
> Dear AC,
>
> To facilitate your assessment, we briefly summarize our interaction with Reviewer R177 and how we addressed the points.
>
> 1. **Pre‑defined groups and design motivation (Q1)**
>    We explained that our model already uses a single always‑active shared expert, while the remaining experts are routed. Rather than hand‑crafting multiple sliced groups, we let the model adaptively form groups based on emergent expert co‑activation patterns at higher granularity, which is precisely what AEG is designed to capture.
>
> 2. **Design of $L_{\text{intra}}$ (Q2)**
>    We initially considered losses based on co‑activation statistics or expert outputs, but found them unstable and input‑dependent under sparse activation. We therefore define $L_{\text{intra}}$ (and $L_{\text{inter}}$) on parameter distances to obtain stable, input‑agnostic grouping.
>
> 3. **Soft‑MoE vs. sparse‑MoE (Q3)**
>    Following the reviewer’s suggestion, we added a discussion of soft‑MoE works [1–4] in the related work and clarified that our method is designed for sparse‑MoE settings.
>
> 4. **Histopathology / high‑resolution images (Q4)**
>    We clarified that all medical modalities, including histopathology, are processed by the Qwen2‑VL vision encoder, with images resized to its supported resolution range, so no special changes to the shared or routed expert design are required.
>
> 5. **Scaling with more experts (Q5)**
>    We vary the number of routed experts from 6 to 192 under a fixed activated parameter budget and observe a consistent trend: finer granularity improves OOD robustness while slightly reducing ID fitting, and also increases expert co‑activation.
>
> 6. **Effect of the vision encoder (Q6)**
>    We noted that a medical‑pretrained vision encoder would likely further improve performance, but we keep Qwen2‑VL’s original encoder to preserve its strong multimodal alignment; this choice is orthogonal to our MoE/AEG design.
>
> **Reviewer’s reaction in the discussion phase**
>
> In the discussion, Reviewer R177 explicitly wrote that “all concerns regarding techniques have been addressed” and increased the confidence score accordingly, without raising further objections.
>
> Best regards,
> The Authors

---

### Official Review · Reviewer_epjP · 2025-10-30

**Soundness:** 2
**Presentation:** 2
**Contribution:** 2
**Rating:** 4
**Confidence:** 3

**Summary:**

This paper studies how the granularity of experts in Mixture-of-Experts (MoE) architectures affects model performance in medical multimodal learning. The authors start from Qwen2-VL-2B and systematically vary the number of sub-experts per layer to observe the trade-off between in-distribution fitting and out-of-distribution (OOD) generalization. They find that finer granularity improves OOD performance and robustness but also increases redundancy among experts. To address this, they propose Adaptive Expert Grouping (AEG), a differentiable grouping mechanism that clusters functionally similar experts during training. The method relies on a Gumbel-Softmax assignment between expert and group embeddings, with a separation loss that encourages cohesive but distinct groups. Experiments on medical VQA datasets (SLAKE, PATH-VQA, OMNI-MINI) show that fine-grained MoE improves OOD generalization and that AEG achieves similar performance while activating fewer experts and reducing routing cost.

**Strengths:**

The experiments are extensive and well-documented. The authors carefully analyze the dual effect of granularity, showing improved OOD robustness but increased redundancy. The AEG mechanism is a practical way to exploit that redundancy. The results show clear quantitative improvements and the efficiency gains are convincing. The inclusion of robustness and co-occurrence analysis adds depth. Overall, the paper is technically strong and addresses a relevant open question in scaling MoE for specialized domains.

**Weaknesses:**

### W1 Limited novelty relative to adaptive gating work.
AEG is essentially a learned clustering mechanism that reduces routing granularity, which overlaps with ideas in XMoE [1] and DeepSeek-MoE [2]. These prior works already explored adaptive expert selection, specialization, and redundancy reduction. The paper would be stronger with direct conceptual and empirical comparisons showing how AEG differs beyond adding a learnable expert-to-group assignment.
### W2 Narrow experimental scope.
All experiments are on medical VQA, which is a limited slice of multimodal reasoning. Without additional tasks (e.g., retrieval, report generation, or non-medical datasets like ScienceQA), it is unclear whether the observed trade-offs and grouping effects generalize to other domains or modalities.
### W3 DenseMaskMoE proxy may distort efficiency claims.
The proxy simplifies training by replacing sparse dispatch with dense masking, but that also changes the real compute and communication pattern of MoE routing. Efficiency results therefore may not reflect actual runtime gains under realistic distributed setups such as DeepSpeed-MoE or Expert Parallelism.
### W4 Missing ablations to isolate the effect of grouping.
There is no comparison against static or heuristic grouping (e.g., k-means on expert weights or co-activation graphs), nor sensitivity analysis on group number, Gumbel temperature, or the separation loss coefficient. Without these, it is hard to tell whether AEG's benefit comes from learned grouping itself or simply from reduced routing depth.
### W5 Lack of interpretability of expert groups.
While the paper visualizes group sizes and sparsity, it never analyzes what these groups represent--whether they align with modalities, anatomical regions, or question types. Simple probes or correlation analyses could make the claimed "functional grouping" more convincing.
### W6 No statistical or significance reporting.
Results are single-seed, with small differences (around 0.5–1.5 points) that may fall within variance. Reporting mean and standard deviation across runs, or simple bootstrap tests, would make the comparisons more credible and show whether the improvements are statistically meaningful.

[1] XMoE: Sparse Models with Fine-grained and Adaptive Expert Selection. https://arxiv.org/abs/2403.18926

[2] DeepSeekMoE: Towards Ultimate Expert Specialization in Mixture-of-Experts Language Model. https://arxiv.org/abs/2401.06066

**Questions:**

1. How sensitive is AEG to the number of groups and the Gumbel-Softmax temperature?
2. Could static clustering (for example k-means on expert weights) achieve similar results without end-to-end learning?
3. How much real compute saving does DenseMaskMoE provide compared to dispatch-based implementations?
4. Are the learned expert groups stable across random seeds or highly variable?
5. Have you tried applying the same setup to a non-medical benchmark to test whether the observed redundancy patterns are domain-specific?

---

> ### Author Response · Authors · 2025-11-23
>
> Dear Reviewer epjP,
>
> Thank you for your thorough review and constructive feedback. We appreciate your valuable insights, which have helped us significantly improve our work. Below, we address each of your concerns point by point.
>
> ---
>
> ## Response to Weaknesses
>
> ### W1: Limited novelty relative to adaptive gating work
>
> We appreciate your concern regarding the relationship between AEG and prior work (XMoE and DeepSeek-MoE). We would like to clarify the key conceptual differences:
>
> **Comparison with XMoE:**
> While XMoE introduces Top-p routing to enable adaptive expert selection, and both XMoE and our AEG exhibit adaptive expert activation, their underlying motivations and design principles differ fundamentally.
>
> Through our experiments, we observed that as expert granularity increases, certain experts exhibit strong co-activation patterns. We interpret this as a collaborative pattern where, due to reduced individual expert capacity at finer granularities, single experts struggle to independently construct complex semantic or knowledge representations. Consequently, the model tends to combine multiple experts to form higher-level semantic features.
>
> Based on this observation, AEG explicitly captures these implicit collaborative relationships through learnable expert grouping, transforming spontaneous expert collaboration into structured, learnable components. We believe this group-based architecture, where groups of varying sizes emerge, is intuitive, as different semantic/knowledge expressions require different expert capacities. This suggests a potential new paradigm for fine-grained MoE: rather than manually designing expert granularity, the model can start from a finer granularity and adaptively form larger, more specialized expert groups.
>
> Unlike Top-p routing, AEG focuses on modeling the emergent behavioral patterns in fine-grained MoE under high granularity. Beyond computational efficiency, AEG promotes inter-group differentiation and specialization. Moreover, while our current implementation uses Top-k routing, it can be extended with Top-p routing for further adaptive group activation.
>
> **Comparison with DeepSeek-MoE:**
> DeepSeek-MoE introduces shared experts to capture common knowledge, thereby reducing expert redundancy and promoting specialization. Our work also employs shared experts. Moreover, we further observe that increasing expert granularity induces spontaneous collaborative behaviors among experts, which introduces a new form of redundancy where the router needs to explicitly distinguish among highly correlated experts. Inspired by this phenomenon, we propose AEG, which mitigates such redundancy through group-level routing. Additionally, we introduce empty groups, which enable the router to dynamically activate fewer experts than the preset number for tokens that require less complex processing, providing greater flexibility in resource allocation.
>
> ---

---

> > ### Author Response · Authors · 2025-11-23
> >
> > ### W2: Narrow experimental scope
> >
> > We acknowledge your concern about generalization across different tasks and domains. Following your suggestion, we have extended our experiments beyond medical VQA to include additional tasks and non-medical domains:
> >
> > **1. Medical Report Generation (MRG):**
> > This task requires generating clinical findings and impressions from one or more medical images, representing a significantly more complex form of multimodal reasoning than VQA. We trained our models on a subset of MIMIC-CXR and evaluated on both MIMIC-CXR (as the in-distribution/ID dataset) and IU-Xray (as the out-of-distribution/OOD dataset). We adopted RadEntityMatchExact (RaTE) as our primary metric, as it more accurately reflects the semantic correctness of clinical entities compared to traditional language generation metrics like BLEU or ROUGE.
> >
> > **2. Non-medical Domain (Food Classification):**
> > Inspired by [1], we selected the food domain as a non-medical testbed to assess cross-domain generalization. We used a subset of the Food101 dataset, formulated as a VQA task for food category classification. The dataset was split into three subsets: train, test (ID), and test (OOD), where the OOD set contains different food categories from those in the training and ID test sets.
> >
> > > **Table 1: Cross-Task and Cross-Domain Evaluation**
> > | **Models**           | **G**  | **MRG**      |         |  | **Food VQA** |           |
> > | ---------------- | -- | -------- | ------- |--  | -------- | --------- |
> > |                  |    | ID       | OOD     |  | ID       | OOD       |
> > |                  |    | MIMC-CXR | IU-Xray |  | Food101  | Food101\* |
> > |                  |    | RaTE     | RaTE    |  | Acc.     | Acc.      |
> > | FGMoE-S6k2       | 2  | 49.88    | 54.84   |  | 93.70    | 38.40     |
> > | FGMoE-S12k4      | 4  | **52.18**    | 53.95   |  | 94.40    | 31.70     |
> > | FGMoE-S24k8      | 8  | 52.05    | **56.92**   |  | 94.50    | 36.10     |
> > | FGMoE-S48k16     | 16 | 49.11    | 52.71   |  | 94.60    | 38.10     |
> > | FGMoE-S96k32     | 32 | 49.02    | 52.50   |  | **94.70**    | 39.60     |
> > | FGMoE-S192k64    | 64 | 49.78    | 53.01   |  | 94.40    | **41.00**     |
> > |                  |    |          |         |  |          |           |
> > | FGMoE-Ada-S12k4  | 4  | 51.74    | 53.26   |  | 94.70    | 24.70     |
> > | FGMoE-Ada-G12k2  | 4  | 51.91    | 53.74   |  | 94.70    | 26.60     |
> > | FGMoE-Ada-S96k32 | 32 | 51.65    | 53.83   |  | **94.80**    | 28.30     |
> > | FGMoE-Ada-G96k2  | 32 | **52.91**    | **54.73**   |  | 94.60    | **34.10**     |
> >
> > **Analysis of Trade-offs Across Tasks:**
> >
> > The results are presented in Table 1. We first examined whether similar trade-offs, improved generalization with slightly reduced fitting capacity as granularity increases, exist in MRG and Food VQA tasks.
> >
> > For **Food VQA**, we observe improved OOD generalization as granularity increases. However, unlike medical VQA where ID performance decreases slightly, the ID performance here fluctuates slightly rather than showing a consistent decline. We attribute this to the relatively lower semantic complexity of food classification compared to medical reasoning tasks.
> >
> > For **Medical Report Generation**, we do not observe the same clear trade-off pattern when granularity (G) increases from 2 to 64. However, when examining our pre-trained models FGMoE-Ada-S12k4 and FGMoE-Ada-S96k32, we observe the characteristic trend consistent with our findings: higher granularity (S96k32) yields better OOD performance (53.83 vs. 53.26) but slightly lower ID performance (51.65 vs. 51.74). We believe this difference stems from the high semantic complexity of report generation compared to VQA tasks, the task's complexity makes it difficult to observe the expected trade-off curve in directly fine-tuned models without sufficient pre-training.
> >
> > **Summary:** Based on these cross-task experiments, we refine our claim as follows: For VQA tasks, the trade-off is relatively clear and generalizes well, higher granularity improves generalization while having a minor impact on fitting capacity. For more complex generation tasks like medical report generation, this pattern emerges clearly in well-pre-trained models, though it may be less apparent in models fine-tuned directly due to the task's higher complexity.
> >
> > **Effectiveness of AEG Across Tasks:**
> >
> > We also evaluated the effectiveness of our AEG method across these tasks. Consistent with our hypothesis, S12k4 performs worse than S96k32, as finer granularity induces more pronounced expert co-activation patterns, making AEG more applicable and beneficial. Importantly, across both medical and food domains, G96k2 (our AEG variant) demonstrates better generalization than S96k32, confirming the effectiveness of our adaptive grouping approach in mitigating the redundancy introduced by fine-grained expert collaboration while maintaining the benefits of increased granularity.

---

> > > ### Author Response · Authors · 2025-11-23
> > >
> > > These results demonstrate that our findings about fine-grained MoE and the effectiveness of AEG extend beyond medical VQA to other tasks and domains, though the manifestation of granularity trade-offs may vary depending on task complexity and model pre-training.
> > >
> > > [1] Cheng D, Huang S, Zhu Z, et al. On Domain-Adaptive Post-Training for Multimodal Large Language Models. arXiv:2411.19930, 2024.
> > >
> > > ---
> > >
> > > ### W3: DenseMaskMoE proxy may distort efficiency claims
> > >
> > > We agree that DenseMaskMoE differs from DeepSpeed-MoE in terms of low-level computation and communication patterns. However, the "efficiency" emphasized by AEG in our work primarily refers to routing and functional efficiency, which is conceptually decoupled from specific MoE implementations. Specifically:
> > >
> > > 1. **Routing simplification from individual experts to groups:**
> > > AEG transforms routing from distinguishing individual experts to distinguishing groups. Multiple experts with high co-activation and similar functionality are learned to be aggregated into the same group. The router no longer needs to make fine-grained distinctions among these highly correlated experts but treats them as a single collaborative unit.
> > >
> > > 2. **Structural empty groups:**
> > > In the current AEG design, a significant portion of groups shrink to size zero after training, causing the router to effectively activate fewer than k groups, exhibiting dynamic sparsity. In typical DeepSpeed-MoE/Expert Parallelism implementations, this behavior avoids triggering cross-device communication and expert-side computation, translating directly into communication and computation savings.
> > >
> > > ---
> > >
> > > ### W4: Lack of comparison against static grouping and sensitivity analysis
> > >
> > > Thank you for this insightful suggestion. We address your concerns regarding comparison experiments and sensitivity analysis as follows:
> > >
> > > **Comparison with static/heuristic grouping:**
> > >
> > > Following your suggestion, we implemented static/heuristic grouping methods based on k-means and co-activation graphs (denoted as co-occurrence). For k-means, we used scikit-learn's `KMeans` function to cluster experts based on their weight parameters, with `n_clusters` set to 3/4 of the total number of routed experts. For the co-occurrence method, we computed Jaccard similarity between experts (based on their co-activation frequency and individual activation counts) to measure expert associations. Connections were established between experts when similarity exceeded a threshold (0.35 for 12 experts, 0.4 for 96 experts), and connected component analysis was applied to form groups. We also tested two variants of AEG: one with maximum group number equal to the number of experts, and another set to 3/4 of the expert count.
> > >
> > > Performance comparisons across methods are shown in Table 2-1. For the 96-expert model with granularity 32, AEG methods outperform static grouping approaches, and by tuning the maximum group number (N), AEG can even surpass the baseline without grouping. However, for the 12-expert model with granularity 4, all grouping methods perform poorly. This aligns with our motivation: we proposed the grouping strategy based on the observation that expert co-activation becomes more prominent with higher granularity. Naturally, this method is better suited for fine-grained configurations and less effective for coarse-grained ones, thereby validating the rationale behind our approach.
> > >
> > > The differences in grouping strategies learned by different methods are illustrated in Tables 2-2 and 2-3. K-means tends toward uniform group sizes, while co-occurrence forms one extremely large group alongside numerous singleton experts (a long-tail distribution). In contrast, AEG achieves a balanced middle ground between these two extremes, creating diverse group sizes that better reflect the natural collaboration patterns among experts.
> > >
> > > These results demonstrate the value and significance of a learnable grouping approach over static/heuristics.

---

> ### Author Response · Authors · 2025-11-23
>
> > **Table 2-1: Performance Comparison between AEG and Static Grouping Methods**
> | **N_experts** | **Methods**        | **SLAKE** |       | **PATH-VQA** |       | **OMNI**  | **Avg-I** | **Avg-A** |
> | --------- | -------------- | ----- | ----- | -------- | ----- | ----- | ----- | ----- |
> | 96        | *Baseline*       | *82.89* | *86.30* | *39.65*    | *92.45* | *64.05* | *75.32* | *69.69* |
> |           | Top-2          | 84.35 | 85.10 | 38.20    | 92.33 | 61.80 | 74.99 | 68.40 |
> |           | Co-Occurrence    | 82.59 | 84.38 | 39.62    | 92.36 | 63.10 | 74.74 | 68.92 |
> |           | K-Means (N=72) | 82.50 | 85.10 | 39.71    | 92.39 | 63.55 | 74.93 | 69.24 |
> |           | AEG (N=96)     | 83.12 | 86.06 | 37.77    | 92.72 | 63.80 | 74.92 | 69.36 |
> |           | AEG (N=72)     | 83.10 | 84.86 | 39.62    | 92.39 | **65.10** | **74.99** | **70.05** |
> |           |                |       |       |          |       |       |       |       |
> | 12        | *Baseline*       | *82.60* | *83.89* | *38.71*    | *92.24* | *63.00* | *74.36* | *68.68* |
> |           | Top-2          | 83.54 | 85.34 | 38.55    | 92.48 | **62.95** | **74.98** | **68.96** |
> |           | Co-Occurrence    | 81.86 | 83.41 | 38.06    | 92.10 | 62.10 | 73.86 | 67.98 |
> |           | K-Means (N=9)  | 81.50 | 83.89 | 38.08    | 92.36 | 62.35 | 73.96 | 68.15 |
> |           | AEG (N=9)      | 84.27 | 87.26 | 37.79    | 91.95 | 60.85 | 75.32 | 68.08 |
> |           | AEG (N=12)     | 83.66 | 85.82 | 37.45    | 92.16 | 58.20 | 74.77 | 66.49 |
>
> > **Table 2-2: Grouping Strategy Comparison (96 experts, granularity 32)**
> >
> > **Columns**: #Groups = maximum number of groups set; Active% = percentage of groups with size > 0; AvgSize = average size of groups with size > 1; Collab% = percentage of groups with size > 1 among all active groups; MaxSize = size of the largest group.
> | Strategy | #Groups |     | Active% | AvgSize | Collab% | SizeStd | MaxSize |
> | -------- | ------- | -------- | ------- | ------- | ------- | ------- | ------- |
> | co-occ    | 96\*    | Avg      | 83.18   | 16.18   | 18.01   | 1.8     | 17      |
> |          |         | Layer 0  | 70.83   | 29      | 30.21   | 3.37    | 29      |
> |          |         | Layer 12 | 88.54   | 12      | 12.5    | 1.19    | 12      |
> |          |         | Layer 26 | 85.42   | 15      | 15.62   | 1.54    | 15      |
> |          |         |          |         |         |         |         |         |
> | k-means   | 72      | Avg      | 100     | 2.24    | 45.16   | 0.59    | 3       |
> |          |         | Layer 0  | 100     | 2.2     | 45.83   | 0.58    | 3       |
> |          |         | Layer 12 | 100     | 2.26    | 44.79   | 0.6     | 3       |
> |          |         | Layer 26 | 100     | 2.33    | 43.75   | 0.62    | 3       |
> |          |         |          |         |         |         |         |         |
> | AEG      | 96      | Avg      | 62.5    | 2.37    | 64.96   | 0.81    | 4.43    |
> |          |         | Layer 0  | 64.58   | 2.36    | 61.46   | 0.78    | 4       |
> |          |         | Layer 12 | 61.46   | 2.37    | 66.67   | 0.84    | 5       |
> |          |         | Layer 26 | 64.58   | 2.17    | 65.62   | 0.64    | 3       |
> |          |         |          |         |         |         |         |         |
> | AEG      | 72      | Avg      | 73.61   | 2.52    | 74.48   | 0.95    | 4.79    |
> |          |         | Layer 0  | 68.06   | 2.74    | 77.08   | 1.16    | 5       |
> |          |         | Layer 12 | 75      | 2.5     | 72.92   | 0.92    | 5       |
> |          |         | Layer 26 | 80.56   | 2.31    | 69.79   | 0.78    | 4       |
> >
> > \*The co-occurrence method theoretically has no preset maximum groups; for fair comparison with AEG, we set it to 96.
>
> > **Table 2-3: Grouping Strategy Comparison (12 experts, granularity 4)**
> | Strategy | #Groups | Active% | AvgSize | Collab% | SizeStd | MaxSize |
> | -------- | ------- | ------- | ------- | ------- | ------- | ------- |
> | coocc    | 12\*    | 95.24   | 1       | 8.33    | 0.17    | 1.57    |
> |          |         | 83.33   | 3       | 25      | 0.6     | 3       |
> |          |         | 100     | 0       | 0       | 0       | 1       |
> |          |         | 100     | 0       | 0       | 0       | 1       |
> |          |         |         |         |         |         |         |
> | kmeans   | 9       | 100     | 2.14    | 47.62   | 0.53    | 2.29    |
> |          |         | 100     | 2       | 50      | 0.47    | 2       |
> |          |         | 100     | 2       | 50      | 0.47    | 2       |
> |          |         | 100     | 2       | 50      | 0.47    | 2       |
> |          |         |         |         |         |         |         |
> | AEG      | 12      | 64.29   | 2.38    | 63.69   | 0.66    | 2.86    |
> |          |         | 66.67   | 2       | 66.67   | 0.5     | 2       |
> |          |         | 83.33   | 3       | 25      | 0.6     | 3       |
> |          |         | 75      | 2       | 50      | 0.47    | 2       |

---

> > ### Author Response · Authors · 2025-11-23
> >
> > **Sensitivity analysis on hyperparameters:**
> >
> > We have supplemented our experiments with sensitivity analyses on group number and Gumbel-softmax temperature, as shown in Tables 3-3, 3-4 and 3-5.
> >
> > >**Table 3-3: Effect of Maximum Group Number on Model Performance**
> > | $N_{group}$ | Avg-I | OOD   | Avg-A |
> > | ------- | ----- | ----- | ----- |
> > | 24      | 75.26 | 62.50 | 69.09 |
> > | 48      | 74.99 | 62.75 | 69.16 |
> > | 72      | 74.99 | 65.10 | 69.30 |
> > | 96      | 74.74 | 63.10 | 68.92 |
> >
> > > **Table 3-4: Effect of Maximum Group Number on Grouping Strategies**
> > >
> > > Column definitions are the same as Table 2-2.
> > | #Groups | Scope     | Active% | AvgSize | Collab% | SizeStd | MaxSize |
> > | ------- | --------- | ------- | ------- | ------- | ------- | ------- |
> > | 96      | Avg (all) | 62.5    | 2.37    | 64.96   | 0.81    | 4.43    |
> > |         | Layer 0   | 64.58   | 2.36    | 61.46   | 0.78    | 4       |
> > |         | Layer 12  | 61.46   | 2.37    | 66.67   | 0.84    | 5       |
> > |         | Layer 26  | 64.58   | 2.17    | 65.62   | 0.64    | 3       |
> > |         |           |         |         |         |         |         |
> > | 72      | Avg (all) | 73.61   | 2.52    | 74.48   | 0.95    | 4.79    |
> > |         | Layer 0   | 68.06   | 2.74    | 77.08   | 1.16    | 5       |
> > |         | Layer 12  | 75      | 2.5     | 72.92   | 0.92    | 5       |
> > |         | Layer 26  | 80.56   | 2.31    | 69.79   | 0.78    | 4       |
> > |         |           |         |         |         |         |         |
> > | 48      | Avg (all) | 87.5    | 2.88    | 86.31   | 1.23    | 5.71    |
> > |         | Layer 0   | 83.33   | 2.87    | 89.58   | 1.26    | 6       |
> > |         | Layer 12  | 91.67   | 2.68    | 86.46   | 1.07    | 5       |
> > |         | Layer 26  | 91.67   | 2.93    | 82.29   | 1.37    | 6       |
> > |         |           |         |         |         |         |         |
> > | 24      | Avg (all) | 100     | 4.25    | 98.14   | 1.87    | 8.36    |
> > |         | Layer 0   | 100     | 4       | 100     | 1.38    | 7       |
> > |         | Layer 12  | 100     | 4.27    | 97.92   | 2.06    | 11      |
> > |         | Layer 26  | 100     | 4.13    | 98.96   | 1.78    | 9       |
> >
> > **Group number analysis (Tables 3-3 and 3-4):** Our experiments are based on FGMoE-Ada-G96k2. By default, we set the maximum number of groups to 96, equal to the total number of experts, with the intention of not introducing any prior knowledge and allowing the model to autonomously determine the effective number of groups through empty groups. According to the experimental results in Table 3-3, moderately reducing the maximum group number (N in the range of 3/4 to 1 of expert count) encourages the model to learn better grouping strategies and improves overall performance.
> >
> > > **Table 3-5: Effect of the Gumbel-Softmax temperature on performance**
> > | $\tau_{start}$ | $\tau_{end}$ | Avg-I | OOD   | Avg-A |
> > | ------- | ----- | ----- | ----- | ----- |
> > | 1       | 0.5   | 75.18 | 64.05 | 69.07 |
> > | 2       | 0.3   | 74.89 | 63.20 | 68.80 |
> > | 2       | 0.5   | 74.74 | 63.10 | 68.92 |
> >
> > **Gumbel-softmax temperature analysis (Table 3-5):** We use a cosine annealing schedule that decreases the temperature from 2.0 to 0.5 during training. Our sensitivity analysis shows that appropriate adjustment of the temperature schedule can further improve model performance.
> >
> > These analyses demonstrate that AEG's benefits stem from learned adaptive grouping rather than simply reducing routing depth. The learnable mechanism allows the model to discover task-specific collaboration patterns that static heuristics cannot capture.
> >
> > ---

---

> ### Author Response · Authors · 2025-11-23
>
> ### W5: Lack of interpretability of expert groups
>
> Thank you for this valuable suggestion. To better understand what the learned expert groups represent, we conducted correlation analyses between expert groups and both modalities and question types in FGMoE-Ada-G96k2, as visualized in **Figures 19 and 20 in the appendix**.
>
> **Analysis Methodology:**
>
> We computed a relative preference coefficient to quantify each group's affinity for different modalities and question types:
>
> $$\text{preference}(g, m) = N_G \times \frac{\text{tokens routed to group } g \text{ from modality } m}{\text{total tokens of modality } m}$$
>
> where $N_G$ denotes the number of active groups (size > 0). A preference value > 1 indicates that group $g$ handles more than its fair share of modality $m$ tokens, suggesting a specialization or routing preference for that modality. Conversely, a value < 1 indicates the group processes fewer tokens from that modality than expected.
>
> Figure 19 illustrates the preference relationships between expert groups and four image modalities (CT, MRI, Pathology, X-Ray). Figure 20 shows the relationships between groups and six text question types: MODALITY (imaging technique/plane), ANATOMY (anatomical structure/system), ABNORMALITY_PRESENCE (presence/absence of abnormality), DISEASE_IDENTIFICATION (type of disease/abnormality), LOCALIZATION (spatial location), and CLINICAL_REASONING (function/symptom/comparison/inference). We performed this analysis on the test sets of SLAKE and PATH-VQA. Question types were annotated using rule-based methods, and the annotations were verified through manual sampling to ensure quality.
>
> **Key Findings:**
>
> **Text Modality Specialization**: At each layer, we observe that one or a few small-sized expert groups (often singleton groups) handle the majority of tokens across all question types. This phenomenon can be attributed to two factors: (1) In multimodal inputs, image tokens constitute the majority while text tokens are relatively sparse, so a small expert group suffices for text processing; (2) Expert groups appear less sensitive to semantic differences among question types, suggesting that text processing is more homogeneous compared to visual processing.
>
> **Image Modality Specialization**: Token distribution across expert groups for image modalities is more balanced and dispersed. Notably, groups showing clear modality preferences typically have size ≥ 2, indicating the emergence of collaborative patterns where multiple experts work together around a dominant expert to handle specific modalities.
>
> **Layer-wise Evolution**: Examining the preference distribution differences across layers reveals interesting patterns. In the final layer, we observe pronounced differentiation among expert groups: some groups (e.g., group_27) show strong preference for Pathology images, others (e.g., group_59) specialize in radiology modalities (CT, MRI, X-Ray), and still others handle mixed modalities. This suggests that the model progressively develops modality-specific processing pathways through its depth.
>
> **Conclusion:**
>
> These analyses demonstrate that expert groups indeed develop functional specialization. Some groups preferentially handle text tokens, while others specialize in specific medical image modalities. The emergence of these specialized groups is not manually designed but naturally learned through the adaptive grouping mechanism, providing evidence that AEG successfully captures meaningful collaboration patterns.
>
> ---

---

> ### Author Response · Authors · 2025-11-23
>
> ### W6: No statistical or significance reporting
>
> Thank you for raising this important concern about statistical rigor. We address this as follows:
>
> **Regarding deterministic evaluation:** We strictly followed standard evaluation protocols for large language models by setting temperature=0 (greedy decoding) to ensure deterministic and reproducible results. Under this setting, evaluation results are identical across different random seeds, as the model's inference is fully deterministic. This is a widely adopted practice in the LLM evaluation community to ensure reproducibility and fair comparison.
>
> **Bootstrap significance testing:** Following your suggestion, we conducted bootstrap tests to assess the statistical significance of our results. We selected MedFGMoE-S12k4 and MedFGMoE-S96k32 to analyze the significance of the generalization improvement and fitting capacity trade-off shown in Figure 3(a).
>
> The results demonstrate that while there is a statistically significant slight decrease in fitting accuracy on in-distribution data (e.g., PATH-VQA Open, $p<0.01$), this trade-off yields a statistically significant improvement in generalization capability in out-of-distribution (OOD) scenarios ($p<0.05$). This confirms that the performance differences we observe are not due to random variance but reflect meaningful and statistically significant trends.
>
> > **Tabel-4: Bootstrap significance testing**
> | Models          | SLAKE |       | PATH-VQA    |       | OOD        | Avg-I | Avg-A |
> | --------------- | ----- | ----- | ----------- | ----- | ---------- | ----- | ----- |
> | MedFGMoE-S12k4  | **82.89** | **85.10** | **37.26**       | **91.98** | 58.70      | **74.31** | 66.50 |
> | MedFGMoE-S96k32 | 82.68 | 84.62 | 35.75       | 91.89 | **60.50**      | 73.73 | **67.12** |
> | P-Value         | 0.4149  | 0.3797  | **0.0016**       | 0.4205  |  **0.0275**       | \-    | \-    |
> | Significance    |       |       |  **(p < 0.01)** |       | *(p < 0.05)* |       |       |
>
> ---
>
> ## Response to Questions
>
> ### Q1: How sensitive is AEG to the number of groups and the Gumbel-Softmax temperature?
>
> We have addressed this issue in our response to W4, where we provided detailed sensitivity analyses. Please refer to that section for our complete response to this concern.
>
> ---
>
> ### Q2: Could static clustering achieve similar results without end-to-end learning?
>
> We have addressed this issue in our response to W4, where we provided detailed sensitivity analyses. Please refer to that section for our complete response to this concern
>
> ---
>
> ### Q3: How much real compute saving does DenseMaskMoE provide compared to dispatch-based implementations?
>
> We systematically compared DenseMaskMoE with DeepSpeed-MoE in both training and inference phases (see Appendix Figure 7, reported in s/it):
>
> **Training phase:** When expert granularity G > 16, DenseMaskMoE consistently outperforms DeepSpeed-MoE, with the advantage increasing as granularity grows. At G = 64, DenseMaskMoE is approximately 5 seconds faster per iteration.
>
> **Inference phase:** As G increases, DeepSpeed-MoE inference time grows from ~0.3 s/it to ~1.75 s/it, while DenseMaskMoE remains nearly constant at ~0.3 s/it, essentially unaffected by granularity increases.
>
> These results demonstrate that DenseMaskMoE significantly reduces actual computational overhead compared to typical dispatch-based implementations, especially at high granularity settings. During inference, it supports finer-grained expert configurations with nearly zero additional cost, making it an efficient proxy for exploring the impact of different granularity configurations on model performance.
>
> ---

---

> > ### Author Response · Authors · 2025-11-23
> >
> > ### Q4: Are the learned expert groups stable across random seeds or highly variable?
> >
> > We conducted systematic testing of AEG across different random seeds to evaluate the stability of learned expert groups. Results show:
> >
> > **Specific Structure Variability:** The exact expert groupings are not identical across different random seeds. Which specific experts belong to which group and the precise group sizes vary between runs.
> >
> > **Statistical Stability:** However, the grouping structure exhibits strong statistical stability. For example, at granularity G=32, the average maximum group size across layers consistently stabilizes around 4, with minimal variation across seeds. The statistical characteristics of grouping strategies under different random seeds are shown in Table 3-5.
> >
> > We attribute this "structurally different but statistically similar" phenomenon to the initialization of the two sets of learnable embedding vectors used to compute group assignments in AEG. Random initialization causes the model to start from different initial partitions under different seeds, leading to convergence toward different specific groupings. However, since fine-grained experts have already formed relatively stable collaborative patterns during pretraining, the overall statistical patterns of AEG's final groupings remain highly consistent across seeds. This suggests that while the surface-level grouping may differ, AEG consistently discovers similar underlying collaboration structures that reflect the inherent relationships among experts.
> >
> > > **Table 3-5: Effect of Random Seed on Grouping Strategies**
> > | Seed | Scope     | Active% | AvgSize | Collab% | SizeStd | MaxSize |
> > | ---- | --------- | ------- | ------- | ------- | ------- | ------- |
> > | 42   | Avg (all) | 62.5    | 2.37    | 64.96   | 0.81    | 4.43    |
> > |      | Layer 0   | 64.58   | 2.36    | 61.46   | 0.78    | 4       |
> > |      | Layer 12  | 61.46   | 2.37    | 66.67   | 0.84    | 5       |
> > |      | Layer 26  | 64.58   | 2.17    | 65.62   | 0.64    | 3       |
> > |      |           |         |         |         |         |         |
> > | 43   | Avg (all) | 61.83   | 2.34    | 66.67   | 0.79    | 4       |
> > |      | Layer 0   | 56.25   | 2.56    | 71.88   | 1.05    | 6       |
> > |      | Layer 12  | 58.33   | 2.54    | 68.75   | 0.94    | 5       |
> > |      | Layer 26  | 62.5    | 2.5     | 62.5    | 0.84    | 4       |
> > |      |           |         |         |         |         |         |
> > | 44   | Avg (all) | 63.84   | 2.41    | 61.83   | 0.81    | 4.36    |
> > |      | Layer 0   | 63.54   | 2.46    | 61.46   | 0.82    | 4       |
> > |      | Layer 12  | 64.58   | 2.26    | 63.54   | 0.71    | 4       |
> > |      | Layer 26  | 63.54   | 2.59    | 59.38   | 0.86    | 4       |
> >
> > ---
> >
> > ### Q5: Have you tried applying the same setup to a non-medical benchmark to test whether the observed redundancy patterns are domain-specific?
> >
> > Thank you for this excellent question. The redundancy patterns we observe are not domain-specific. As mentioned in our response to W2, we tested our model on a non-medical benchmark (Food domain). We analyzed how expert co-occurrence patterns evolve with increasing granularity in models trained on the food domain, as visualized in **Figure 21 in the appendix**. The analysis demonstrates that both medical and non-medical (food) domains exhibit similar trends: as expert granularity increases, expert co-occurrence becomes more pronounced. This consistency across domains suggests that the redundancy patterns induced by fine-grained experts are a general phenomenon in MoE architectures rather than a characteristic specific to medical data, thereby supporting the broader applicability of our AEG approach.
> >
> > ---
> >
> > We hope these responses adequately address your concerns. We are happy to provide further clarification or additional experiments as needed. Thank you again for your valuable feedback.
> >
> > Best regards,
> > The Authors

---

> > > ### Author Response · Authors · 2025-11-26
> > >
> > > Dear Reviewer epjP,
> > >
> > > We hope this message finds you well. As the discussion period is progressing, we would like to kindly follow up regarding our response to your review. If there are any additional questions, concerns, or points you would like us to clarify, please let us know. Your insights are invaluable to us, and we are happy to clarify any remaining points that could help with your evaluation.
> > >
> > > We sincerely appreciate the time and effort you have dedicated to evaluating our paper and would be very grateful for any further comments you may have during the discussion phase.
> > >
> > > Thank you again for your thoughtful feedback and for reviewing our submission.
> > >
> > > Best regards,
> > >
> > > The Authors

---

> > > > ### Author Response · Authors · 2025-11-28
> > > >
> > > > Dear Reviewer epjP,
> > > >
> > > > We hope this message finds you well. As the discussion period is approaching its end, we want to gently follow up on our earlier response to your review. If there is anything we can clarify or expand upon to assist with your evaluation, we would be very happy to provide additional details.
> > > >
> > > > Thank you again for the time and effort you have devoted to evaluating our work. We would be grateful for any further feedback you may have.
> > > >
> > > > Best regards,
> > > >
> > > > The Authors

---

> ### Author Response · Authors · 2025-11-30
> **Response Summary for Reviewer epjP**
>
> Dear AC,
>
> To facilitate your assessment, we briefly summarize how we addressed Reviewer epjP’s main concerns and questions.
>
> 1. **Novelty vs. adaptive gating (W1)**
>    We clarified the conceptual difference between AEG and prior adaptive gating work such as XMoE and DeepSeek‑MoE. XMoE still routes at the individual expert level (Top‑p), requiring the router to differentiate many highly co‑activated experts and not ensuring that entire collaboration groups are activated together. Our work instead starts from fine‑grained experts, explicitly models the emergent *collaborative* patterns via learnable groups, and routes at the **group level**. We also adopt shared experts as in DeepSeek‑MoE, and AEG further promotes specialization by operating on top of this shared‑expert backbone.
>
> 2. **Experimental scope and domain generality (W2, Q5)**
>    Beyond medical VQA, we added **medical report generation (MRG)** on MIMIC‑CXR → IU‑Xray and a **non‑medical Food VQA** benchmark based on Food101.
>    - **Food VQA:** As granularity increases, OOD accuracy improves (e.g., from 38.40 to 41.00), while ID accuracy only fluctuates slightly around 94.4–94.7.
>    - **MRG:** For pre‑trained models, higher granularity (S96k32 vs. S12k4) yields better OOD RaTE (53.83 vs. 53.26) at a small ID cost (51.65 vs. 51.74).
>    These results indicate that the “higher granularity → better OOD with a minor ID trade‑off” pattern is robust for VQA‑style tasks and remains observable for more complex generation tasks when models are sufficiently pre‑trained. We also confirm that similar **expert co‑occurrence** patterns appear in the non‑medical food domain, suggesting they are not specific to medical data.
>
> 3. **DenseMaskMoE and efficiency (W3, Q3)**
>    We clarified that our main notion of “efficiency” is **routing and structural efficiency**, which is largely independent of the specific MoE implementation. AEG reduces the router’s burden from distinguishing many similar experts to distinguishing a smaller number of groups. Empirically, we also show that compared to DeepSpeed‑MoE, DenseMaskMoE keeps inference latency almost constant (~0.3 s/it) as granularity increases, while DeepSpeed‑MoE grows to ~1.75 s/it at high granularity, making DenseMaskMoE a practical proxy for exploring expert granularity.
>
> 4. **AEG vs. static grouping and ablations (W4, Q1, Q2, Q4)**
>    We added comparisons to **static grouping** (k‑means on weights and co‑occurrence graphs) and ablations on group number, Gumbel‑Softmax temperature, and random seeds.
>    - In the **96‑expert** setting, AEG (N=72) achieves the best performance (Avg‑A 70.05, OMNI 65.10), outperforming both the baseline without grouping (Avg‑A 69.69, OMNI 64.05) and static methods. For **12 experts**, all grouping methods provide limited gains, matching our observation that grouping is most beneficial when fine‑grained experts exhibit strong co‑activation.
>    - Moderately constraining the maximum number of groups (e.g., 72 vs. 96) improves OOD performance (65.10 vs. 63.10). Different temperature schedules lead to small but consistent differences, and across random seeds, statistics such as Active% and AvgSize remain similar even though individual group assignments change. Together, these results indicate that AEG relies on stable discovered collaboration patterns rather than on a fragile, over‑tuned configuration.
>
> 5. **Interpretability of expert groups (W5)**
>    We performed correlation analyses between expert groups and **modalities** (CT, MRI, Pathology, X‑ray) as well as **question types** (e.g., ANATOMY, DISEASE_IDENTIFICATION, LOCALIZATION), as shown in Appendix Figures 19–20. We observe that text tokens are predominantly handled by small groups across question types, and some larger groups show clear preferences for specific image modalities (e.g., pathology‑focused vs. radiology‑focused groups). This supports that AEG learns meaningful, functionally specialized expert groups rather than arbitrary clusters.
>
> 6. **Statistical significance (W6)**
>    All evaluations use **greedy decoding (temperature = 0)** to ensure deterministic outputs. In addition, bootstrap tests comparing S12k4 vs. S96k32 show that the small ID accuracy drop on PATH‑VQA‑Open is statistically significant (**p < 0.01**), and the OOD improvement is also significant (**p < 0.05**). This confirms that the observed trade‑off between ID fitting and OOD generalization reflects a real, statistically meaningful effect rather than random noise.
>
> Best regards,
> The Authors

---

### Author Response · Authors · 2025-11-26

We sincerely thank all reviewers for their time, constructive feedback, and helpful suggestions.

Our work is, to the best of our knowledge, the first to introduce a fine‑grained MoE framework into the medical multimodal domain. We begin with a systematic study on medical VQA, where we find that increasing expert granularity improves generalization and robustness (with slightly reduced fitting capacity) and leads to stronger expert co‑activation. Motivated by these observations, we propose Adaptive Expert Grouping (AEG), which exploits expert co‑activation to achieve better computational efficiency while preserving the modeling benefits of fine‑grained experts, and we observe similar behaviors and gains beyond medical VQA.

Overall, our work has been recognized for:
- **Systematic analysis of expert granularity** (epjP, R177, 7da9), providing **original insights** (Yx3H) and **practical guidance** for MoE design (7da9).
- **A simple yet efficient method (AEG)** (epjP, R177, Yx3H) that is **easily integrable with existing MoE infrastructures** (R177) and **enables adaptive computation** via **dynamic expert grouping** (7da9).
- **Extensive experiments** across **diverse evaluation settings** (zero‑shot, fine‑tuning, robustness) (epjP, R177, Yx3H, 7da9), offering strong empirical validation.

We have provided detailed responses to each reviewer and summarize below the key additional experiments added during the rebuttal:

- **Broader experiments** on medical report generation and a non‑medical (food) domain (Page 25, Table 10; Page 28, Figure 21).
- **Direct comparison** between AEG and **static expert grouping** methods (Page 8, Table 3; Page 9, Table 4).
- **Analysis of the AEG hyperparameter of maximum group size** (Page 26, Tables 11 and 12).
- **Expert–modality specialization analysis** across CT, MRI, pathology, X‑ray, and different question types (Pages 27–28, Figures 19 and 20).

We hope these points demonstrate the rigor and impact of our work. Thank you again for your time and thoughtful consideration.

---

### Author Response · Authors · 2025-11-30
**Rebuttal Summary**

Dear AC,

We sincerely appreciate your time and effort in handling our submission. To facilitate your assessment, we briefly summarize the main concerns raised across reviewers and how we addressed them. More detailed per‑reviewer summaries are provided below for each reviewer.

**Main concerns and our responses**

1. **Positioning and novelty** (epjP, 7da9, Yx3H):
   We clarified that AEG differs from prior adaptive gating (e.g., XMoE) and grouping methods (e.g., MoEC, MoGE) by performing **group‑level routing** via learnable groups discovered from observed co‑activation patterns, rather than routing individual experts or relying on fixed, manually designed structures.

2. **Domain generality and task coverage** (epjP, 7da9, Yx3H):
   We added experiments on **medical report generation** and **non‑medical Food VQA**, showing that the granularity–generalization trade‑off and expert co‑occurrence patterns **generalize beyond medical VQA**.

3. **Efficiency and comparison with alternatives** (epjP, 7da9):
   We added comparisons to **expert pruning** and **static grouping**, showing that AEG preserves performance while improving efficiency.

4. **Interpretability** (7da9, epjP):
   We conducted detailed routing analyses showing that expert groups exhibit **modality‑specific** preferences, indicating meaningful functional specialization rather than arbitrary clustering.

5. **Stability and ablations** (epjP, R177):
   We demonstrated that AEG's performance is robust across different random seeds, temperature schedules, and the number of groups.

**Reactions from reviewers**

- **Reviewer Yx3H** stated that our rebuttal **“addressed his/her concerns a lot”** and **“made the paper more solid.”**
- **Reviewer R177** wrote that **“all concerns regarding techniques have been addressed”** and **increased his/her confidence score**.

We appreciate your consideration and the additional effort in evaluating our work.

Best regards,
The Authors

---

### Meta-Review · Area_Chair_X7nJ · 2026-01-06

**Summary:**

Although the paper is technically solid and clearly written, it does not provide a sufficiently novel or impactful contribution relative to existing literature to warrant acceptance at ICLR.
Thus, the rebuttal improves clarity but does not significantly strengthen the contribution.
ICLR has a high bar for novelty and influence. Papers that mainly refine or repackage existing ideas, even when well executed, are better suited for venues emphasizing incremental progress. For these reasons, I recommend rejection.

**Reviewer Concerns:**

Reviewers agree that:
- The experiments and analysis studies are comprehensive.
- The proposed method is simple yet efficient.
- Similar ideas have been explored in prior work, either explicitly or implicitly.

While the paper shows competent execution, the contribution is primarily incremental. It lacks conceptual or methodological novelty.
In the rebuttal, the authors clarify distinctions with related work and emphasize empirical improvements. However:
- The clarified differences are narrow in scope.
- The proposed method lacks a well-motivated observation, insight, or corresponding experimental evidence.
- The interpretability is limited to modality level and lacks fine-grained explanations for clinicians, such as which image regions affect the final prediction

**Reviewer Scores:**

Reviewer epjP, 7da9, Yx3H  may keep the score.
Reviewer R177 may increase the score to 7.

---

### Decision · Program_Chairs · 2026-01-26

Reject